# Improving the representation of cropland sites in the Community Land Model (CLM) version 5.0

Theresa Boas[1,2], Heye Bogena[1,2], Thomas Grünwald[3], Bernard Heinesch[4], Dongryeol Ryu[5], Marius Schmidt[1], Harry Vereecken[1,2], Andrew Western[5], Harrie-Jan Hendricks Franssen[1,2]

[1]Research Centre Jülich, Institute of Bio- and Geosciences: Agrosphere (IBG-3), 52425 Jülich, Germany
[2]Centre for High-Performance Scientific Computing in Terrestrial Systems: HPSC TerrSys, Geoverbund ABC/J, 52425 Jülich, Germany.
[3]Technische Universität Dresden (TU Dresden): Institute of Hydrology and Meteorology, 01062 Dresden, Germany
[4]University of Liège: Gembloux Agro-Bio Tech (GxABT), 5030 Gembloux, Belgium
[5]University of Melbourne: Department of Infrastructure Engineering, Parkville VIC 3010, Australia

*Correspondence to:* Theresa Boas (t.boas@fz-juelich.de)

**Abstract.** The incorporation of a comprehensive crop module in land surface models offers the possibility to study the effect of agricultural land use and land management changes on the terrestrial water, energy and biogeochemical cycles. It may help to improve the simulation of biogeophysical and biogeochemical processes on regional and global scales in the framework of climate and land use change. In this study, the performance of the crop module of the Community Land Model version 5 (CLM5) was evaluated at point scale with site specific field data focussing on the simulation of seasonal and inter-annual variations in crop growth, planting and harvesting cycles, and crop yields as well as water, energy and carbon fluxes. In order to better represent agricultural sites, the model was modified by (1) implementing the winter wheat subroutines after Lu et al. (2017) in CLM5; (2) implementing plant specific parameters for sugar beet, potatoes and winter wheat, thereby adding the two crop functional types (CFT) for sugar beet and potatoes to the list of actively managed crops in CLM5; (3) introducing a cover cropping subroutine that allows multiple crop types on the same column within one year. The latter modification allows the simulation of cropping during winter months before usual cash crop planting begins in spring, which is an agricultural management technique with a long history that is regaining popularity to reduce erosion and improve soil health and carbon storage and is commonly used in the regions evaluated in this study. We compared simulation results with field data and found that both the new crop specific parameterization, as well as the winter wheat subroutines, led to a significant simulation improvement in terms of energy fluxes (RMSE reduction for latent and sensible heat by up to 57 % and 59 %, respectively), leaf area index (LAI), net ecosystem exchange and crop yield (up to 87 % improvement in winter wheat yield prediction) compared with default model results. The cover cropping subroutine yielded a substantial improvement in representation of field conditions after harvest of the main cash crop (winter season) in terms of LAI magnitudes and seasonal cycle of LAI, and latent heat flux (reduction of winter time RMSE for latent heat flux by 42 %). Our modifications significantly improved model simulations and should therefore be applied in future studies with CLM5 to improve regional yield predictions and to better understand large-scale impacts of agricultural management on carbon, water and energy fluxes.

## 1    Introduction

Global climate change is widely believed to have an important impact on future agriculture and consequently food security under changing climate is an important research topic (Lobell et al., 2011; Aaheim et al., 2012; Ma et al.,

40 2012; Gosling, 2013; Rosenzweig et al., 2014). With a trend of declining crop yield and increasing uncertainty in yields in many parts of the world (Urban et al., 2012; Challinor et al., 2014; Deryng et al., 2014; Rosenzweig et al., 2014; Tai et al., 2014; Levis et al., 2018), understanding the impact of climate change on crop production and improving its prediction at local to global scales is a research topic of great importance to society. Also, agricultural expansion and management practices exert strong influences on physical and biogeochemical properties of

45 terrestrial ecosystems that need to be considered in model simulations of the terrestrial system. Thus, the evaluation and improvement of integrated modelling approaches, including through incorporation of improved crop phenology, to simulate realistic land management and crop yield in response to climate conditions are the focus of many studies (Stehfest et al., 2007; Olesen et al., 2011; Van den Hoof et al., 2011; Rosenzweig et al., 2014). Nevertheless, the sophisticated representation of agricultural land cover in Earth system models (ESMs) remains

50 an ongoing challenge due to the complexity of agricultural management decisions and the variety of different crop types and their respective phenologies. In many land surface models (LSMs) and land components of ESMs, the representation of crops is limited to simplistic schemes lacking the representation of management (e.g. irrigation and fertilization) or to surrogate representation by natural grassland (Betts, 2005; Elliott et al., 2015; McDermid et al., 2017). In recent studies there is a trend towards the incorporation of a comprehensive crop module in LSMs.

55 These modules offer improved potential to study changes in water and energy cycles and crop production in response to climate, environmental, land use, and land management changes. This may help to improve the simulation of biogeophysical and biogeochemical processes on regional and global scales (Kucharik and Brye, 2003; Lobell et al., 2011; Lokupitiya et al., 2009; Levis et al., 2012; Osborne et al., 2015; McDermid et al., 2017; Lawrence et al., 2018; Lombardozzi et al., 2020). For example, the Simple Biophere model (SiB) incorporated a

60 crop module to represent a number of temperate crop varieties which resulted in improved simulated LAI and net ecosystem exchange (NEE) (Lokupitiya et al., 2009). Also, the Joint UK Land Environment Simulator (JULES) was extended to a global representation of crops which improved simulated LAI and gross primary production (GPP) (Osborne et al., 2015).

 Recent versions of CLM (i.e. 4.0, 4.5 and 5.0) have adopted the prognostic crop module from the Agro-Ecosystem

65 Integrated Biosphere Simulator (Agro-IBIS) (Kucharik and Brye, 2003), which has the ability to simulate the soil-vegetation-atmosphere system including crop yields, and has been evaluated in multiple studies (e.g. Twine and Kucharik, 2009; Webler et al., 2012; Xu et al., 2016). Even the simplified version of the Agro-IBIS crop scheme that was implemented in CLM4 led to improved simulation of climate-crop interactions and more comprehensive ecosystem balances than previous CLM versions (Levis et al., 2012). Evaluation studies of CLM4 by Levis et al.

70 (2012) and Chen et al. (2015) revealed significant sensitivities of energy and carbon fluxes to biases in crop phenology, especially for the seasonality of the NEE for managed crop sites where the flux is governed by planting and harvest times. In its latest version, CLM (CLM5) has been extended with an interactive crop module that represents crop management. It includes eight actively managed crop types (temperate soybean, tropical soybean, temperate corn, tropical corn, spring wheat, cotton, rice, and sugarcane), as well as irrigated and non-irrigated

75 unmanaged crops (Lombardozzi et al., 2020). CLM5 is to date the only land surface model that includes time-varying spatial distributions of major crop types and their management (Lombardozzi et al., 2020). Despite these improvements over earlier versions of CLM, the few studies that evaluated CLM5 at point and regional scale suggest inaccurate phenology and crop yield estimates for specific crops (Chen et al., 2018; Sheng et al., 2018). In summary, current crop modules in LSMs are limited by their ability to represent many different crop types and

80 important management practices such as cover cropping, flexible fertilizer application types and amounts, etc. The

main challenges are related to the complex parameterization of simulated crop varieties due to their distinct phenology in combination with information scarcity, as well as the complexity of human interaction through management decisions and biogeochemical processes. In addition to irrigation and fertilizer application, crop rotations and cover cropping are important management practices and their consideration is a crucial factor to accurately represent energy fluxes and crop phenology of agricultural sites (or areas) over longer time scales.

In Western Europe, a large proportion of arable land is cultivated with rotations of different non-perennial cash crops (Kollas et al., 2015; Eurostat, 2018). The most important cash crops grown in the European Union (EU) are cereals such as wheat (mostly winter wheat varieties in Western Europe), barley and maize, root crops such as sugar beet and potatoes, and oilseed crops such as rape, turnip rape, and sunflower (Eurostat, 2018). Cereals account for the majority of all crop production in the EU, contributing up to 12 % to global cereal grain production (Eurostat, 2018). The EU production of sugar beet accounts for about half of the global production (Eurostat, 2018). The use of cover crops is a common agricultural management practice to reduce soil erosion, soil compaction, and nitrogen leaching as well as to increase agricultural productivity by nitrogen fixation (Sainju et al., 2003; Lobell et al., 2006; Basche et al., 2014; Plaza-Bonilla et al., 2015; Tiemann et al., 2015; Kaye and Quemada, 2017). The biogeochemical effects and benefits of cover crops as well as their potential to mitigate climate change are the focus of many studies (e.g. Sainju et al., 2003; Lobell et al., 2006; Groff, 2015; Plaza-Bonilla et al., 2015; Basche et al., 2016; Carrer et al., 2018; Lombardozzi et al., 2018; Hunter et al., 2019). Despite recent development efforts, the representation of these management practices has not yet been included in CLM5. Furthermore, in a previous study by Lu et al. (2017) the default representation of winter cereals performed poorly in simulating the phenology of winter wheat.

In this study, we evaluate and enhance the performance of the crop module of CLM5 focusing on the representation of seasonal and inter annual variations in crop growth, planting and harvesting cycles, and crop yields as well as energy and carbon fluxes. First, we have transferred the modified vernalization and cold tolerance routine by Lu et al. (2017) to the CLM5 code to simulate winter cereal in a more meaningful way. Secondly, new crop specific parameter sets for winter wheat, sugar beet and potatoes that were gathered from the literature and from observation data were added to the default parameter scheme. Finally, we extended CLM5 by adding a new crop rotation and cover cropping subroutine that models the growth of winter cover crops and the rotation from a summer to a winter crop within the same year. All modification were tested at point scale at four cropland reference sites of the ICOS (Integrated Carbon Observation System) and TERENO (Terrestrial Environmental Observatory) networks in central Europe.

## 2    Materials and Methods

### 2.1    Community Land Model

Land surface models such as CLM5 are broadly applied in scientific studies to simulate water, energy and nutrient fluxes in the terrestrial ecosystem (Niu et al., 2011; Han et al., 2014; Lawrence et al., 2018; Naz et al., 2019). CLM5 represents the latest version of the land component in the Community Earth System Model (CESM) (Lawrence et al., 2018; 2019). In CLM5, simulated land surface fluxes such as latent and sensible heat are driven by atmospheric/meteorological input variables in combination with soil and vegetation states (e.g. soil moisture and LAI) and parameters (e.g. hydraulic conductivity, land cover) (Oleson et al., 2010; Lawrence et al., 2011; Lawrence et al., 2018). The new biogeochemistry and crop module of CLM5 (BGC-Crop) adopted the prognostic

crop module from the Agro-Ecosystem Integrated Biosphere Simulator (Agro-IBIS) (Kucharik and Brye, 2003). This incorporation of agriculturally managed land cover may help to improve the general representation of biogeochemical processes on the global scale to better address challenges from land use changes and agriculture practices (e.g. Lobell, Bala, and Duffy, 2006). The CLM5 crop module includes new crop functional types, updated fertilization rates and irrigation triggers, a transient crop management option as well as some adjustments to

phenological parameters. Also extensive modifications have been made to the grain C and N pool, e.g. C for annual crop seeding comes from the grain C pool and initial seed C for planting is increased from 1 to 3 gCm$^{-2}$ (Lawrence et al., 2018, 2019; Lombardozzi et al., 2020).

Vegetated land units are separated into natural vegetation and crop land units, with only one crop functional type (CFT) on each soil column, including irrigation as a CFT specific land management technique ( Lawrence et al.,

2018; Lombardozzi et al., 2020). A total of 78 plant and crop functional types are included in CLM5 including an irrigated and unirrigated unmanaged C3 crop, eight actively managed crop types - spring wheat, temperate and tropical corn, temperate and tropical soybean, cotton, rice and sugarcane and 23 crop types without specific crop parameters associated that are merged to the most closely related and parameterised CFTs (Lombardozzi et al., 2020). For the simulation of those inactive crop types, the specific crop parameters of the spatially closest and

most similar out of the eight active crop types are used. Irrigation is simulated dynamically for defined irrigated CFTs in response to soil moisture conditions and is partly based on the implementation of Ozdogan et al. (2010) (Leng et al., 2013; Lawrence et al., 2018).

Besides water availability from irrigation and precipitation, crop yield and food productivity greatly depends on fertilization. In CLM5-BGC-Crop, fertilization is represented by adding nitrogen directly to the soil mineral pool

(Lawrence et al., 2018). Fertilization dynamics and annual fertilizer amounts depend on the crop functional types and vary spatially and yearly based on the land use and land cover change time series derived from the Land Use Model Intercomparison Project (Lawrence et al., 2019). In CLM5, land fractions with natural vegetation are not influenced by fertilizer application. In cropping units, mineral fertilizer application starts during the leaf emergence phase of crop growth and continues for 20 days. Manure nitrogen is applied at slower rates (0.002 kg N m$^{-2}$ per

year by default) to prevent rapid denitrification rates that were observed in earlier CLM versions so that more uptake by the plant is achieved (Lawrence et al., 2018).

CLM5-BGC-Crop is fully prognostic with regards to carbon and nitrogen in the soil, vegetation and litter at each time step. The crop phenology as well as the carbon and nitrogen cycling processes follow three phenology phases: phase (1) from planting to leaf emergence, phase (2) from leaf emergence to beginning of grain fill and phase (3)

from beginning of grain fill to maturity and harvest. These phenology phases are governed by temperature thresholds and the percentage of Growing Degree Days (GDD) required for maturity of the crop with harvest occurring when maturity is reached (Lombardozzi et al., 2020).

The first phenology stage, planting, starts when crop specific 10-day mean temperature thresholds (of both the daily 2-m air temperature $T_{10d}$ and the daily minimum 2-m air temperature $T_{min,10d}$) are met. The transition from

planting to leaf emergence (phase 2) begins when the growing degree-days of soil temperature at 0.05 m depth ($GDD_{Tsoi}$) reaches 1 - 5 % of the GDD required for maturity ($GDD_{mat}$), depending on a crop specific base temperature for the $GDD_{Tsoi}$. Grain fill (phase 3) starts with either the simulated 2-m air temperature ($GDD_{T2m}$) reaching a heat unit threshold (h) of 40 – 65 % of $GDD_{mat}$ or when the maximum leaf area index ($L_{max}$) is reached. The crop is harvested in one time step when 100 % $GDD_{mat}$ is reached or when the crop specific maximum number

of days past planting is exceeded. The LAI is dependent on the specified specific leaf area (SLA) and the calculated leaf C. The SLA as well as the maximum LAI are specified for each crop in the parameter file (Table A2).

The allocation of carbon and nitrogen also follows the phenology phases. During the leaf emergence phase, carbon from the seed carbon pool is transferred to the leaf carbon pool. Nitrogen is supplied through the soil mineral nitrogen pool. During the grain fill phases, nitrogen from the leaf and stem of the plant is translocated to the grain

pool. Allocation ends upon harvest of the crop where grain carbon and nitrogen are transferred from the grain pool to the grain product pool and, a small amount of 3g C m$^{-2}$, to the seed carbon pool for the next planting (Lawrence et al., 2018;  Lombardozzi et al., 2020).

The total amount of assimilated carbon and nitrogen is regulated by availability of soil nitrogen, among other resources, and also depends on crop specific target C/N ratios in the plant tissue (varying for roots, stem, leaves,

reproductive pools) (Lawrence et al., 2018;  Lombardozzi et al., 2020). For a detailed technical description of the model and all its features, the reader is referred to the technical documentation and description of new features in CLM5 ( Lawrence et al., 2018,  2019; Lombardozzi et al., 2020).

### 2.2      Model modifications

In the course of this study, three main limitation of CLM5 for the intended simulation of agricultural sites in

Western Europe at point scale were identified: (1) the default CLM5-BGC-Crop code and parameterization yielded a very poor representation of crop growth of winter wheat and other winter crops, (2) the default plant parameter data set lacks specific parameterization for several important cash crops (here especially sugar beet and potatoes), and (3) CLM5-BGC-Crop does not allow a second crop growth onset or a second CFT to be grown on the same field within one year. These limitations were met by modifications to the code structure and parameterization of

the CLM5-BGC-Crop module described below.

### 2.2.1      Winter cereal representation

Winter wheat is an important crop for global food production and covers a significant fraction of the European croplands. (Chakraborty and Newton, 2011; Vermeulen et al., 2012). In general, winter wheat is exposed to a different range of environmental stresses compared to summer crops such low temperatures. In regions with

sufficiently cold winters, the main processes that allow a successful cultivation of winter wheat during the colder months are vernalization and cold tolerance (Barlow et al., 2015; Chouard, 1960). Vernalization represents the process that an exposure to a period of non-lethal low temperatures is required to enter the flowering stage for winter crops. In general, the vernalization process ensures that the reproductive development of plants growing over winter (winter crops and also natural vegetation) does not start in late summer or fall but rather in late winter

or spring. The other process, cold tolerance, ensures that the crop can acclimate to low temperatures and thus survive cold temperatures and even freeze-thaw cycles. However, cold damage to the crop can occur when the crop is exposed to low temperatures at a certain development stage. These damages have been documented to have significant impacts in crop yield (Lu et al., 2017). Lu et al. (2017) introduced a new vernalization, as well as a cold tolerance and frost damage subroutine in CLM4.5 to better simulate the phenology of winter cereal.  For this, they

adapted the winter wheat vernalization model from Streck et al. (2003). Streck et al. (2003) evaluated their vernalization algorithm for a wide range of winter wheat cultivars for the purpose of being used in crop model approaches. Furthermore, Lu  et al. (2017)  implemented  a cold tolerance scheme including frost damage

representation using the approaches after Bergjord et al. (2008) and Vico et al. (2014). In this study, their modifications were ported to the newer version of the model, CLM5, and tested for several study sites.

Vernalization and cold tolerance are cumulative processes that operate in a certain optimum temperature ranges (that can be different for different crop types and cultivars). The vernalization process starts after leaf emergence and ends before flowering (Streck et al., 2003) and is dependent on the crown temperature ($T_{crown}$) (see Eq. A1). The crown is the connecting tissue between the roots and the shoots at the base of the plant. For winter wheat, the crown node is located at about 3 – 5 cm soil depth (Aase and Siddoway, 1979). The daily vernalization dependence

is calculated based on $T_{crown}$ and the optimum vernalization temperature ($T_{opt}$), limited to times when the crown temperature lies within the minimum to maximum vernalization temperature ($T_{min}$ and $T_{max}$) range:

$$vd = \sum fvn(T_{crown}) \tag{1}$$

$$fvn(T_{crown}) = \frac{2(T_{crown}-T_{min})^{\alpha}(T_{opt}-T_{min})^{\alpha}-(T_{crown}-T_{min})^{2\alpha}}{(T_{opt}-T_{min})^{2\alpha}} \tag{2}$$

$$\alpha = \frac{ln2}{ln[(T_{max}-T_{min})/(T_{opt}-T_{min})]} \tag{3}$$

$$vf = \frac{vd^5}{22.5^5+vd^5} \tag{4}$$

where $vd$ [-] is the sum of the sequential vernalization days, $fvn$ [-] is the daily vernalization rate, $vf$ [-] is the vernalization factor, $T_{crown}$ [K] is the crown temperature, $T_{opt}$ [K], $T_{max}$ [K] and $T_{min}$ [K] are the optimum, maximum and minimum vernalization temperatures respectively.

The vernalization factor can range between 0 (not vernalized) and 1 (fully vernalized). It is multiplied with the

GDD during the phenology phase after planting and the grain carbon allocation coefficient which leads to a reduced growth rate in the beginning of the phenology cycle until the plant is fully vernalized. The vernalization factor is further used in the cold tolerance subroutine to assess the cumulative cold hardening of the plant and the dehardening process when exposed to higher temperatures (see below). Lu et al. (2017) introduced a scheme to quantify the impacts of frost damage based on the approaches after Bergjord et al. (2008) and Vico et al. (2014).

The damage from low temperatures is quantified by three main variables: the temperature at which 50 % of the plant is damaged ($LT_{50}$), the survival probability ($f_{surv}$) and winter killing degree days (WDD) (Bergjord et al., 2008; Lu et al., 2017; Vico et al., 2014). A detailed description of these approaches can be found in Bergjord et al. (2008) and Vico et al. (2014).

The temperature at which 50 % of the plant is damaged ($LT_{50}$) is calculated interactively at each time step ($LT_{50,t}$)

depending on the previous time step ($LT_{50,t-1}$) and on several accumulative parameters. These parameters are the exposure to near-lethal temperatures ($rate_s$), the stress due to respiration under snow ($rate_r$), the cold hardening or low temperature acclimation (contribution of hardening – $rate_h$) and the loss of hardening due to the exposure to a period of higher temperatures (dehardening – $rate_d$) that are each functions of the crown temperature (Lu et al., 2017 and references therein) (see Eq. A2-A11).

The survival rate ($f_{surv}$) is then calculated as a function of $LT_{50}$ and the crown temperature. The probability of survival is a function of $T_{crown}$ in time (t). It increases once $T_{crown}$ is higher than $LT_{50}$ or decreases when it is lower (Vico et al., 2014):

$$f_{surv}(T_{crown}, t) = 2^{-\frac{T_{crown}^{\alpha_{surv}}}{LT_{50}}} \tag{5}$$

where $\alpha_{surv}$ is a shape parameter of 4.

The winter killing degree day (WDD) is calculated as a function of crown temperature and survival probability, where the maximum function limits the integration to the potentially damaging periods, when the air temperature ($T$) is lower than the base temperature ($T_{base}$) of 0°C (Vico et al., 2014):

$$WDD = \int_{winter} max[(T_{base} - T_{crown}),0]\ [1 - f_{surv}(T_{crown}, t)]dt \qquad (6)$$

Lower $LT_{50}$ indicate a higher frost tolerance and would result in higher survival rates, smaller WDD and less cold
damage to the plant. Thus, when the survival probability and crown temperature are low, the WDD will be high (Vico et al., 2014).

Lu et al. (2017) also implemented a relationship between frost damage described above and the subsequent growth or carbon allocation of the plant. Whenever the survival factor is less than 1, a small amount of leaf carbon (5 g C m$^{-2}$ per model time step) as well as a small amount of leaf nitrogen (scaled by the prescribed C/N target ratios,
Table 1 and Table A2) are transferred to the soil carbon and nitrogen litter pool thus simulating a reduction in growth and/or damage of small/young leaves and seedlings. Additionally, in order to simulate more drastic and instantaneous damage or death of the plant due to a longer duration of lethal temperatures (most likely to occur in spring when the plant has emerged and is close to or already fully vernalized), a second frost damage function is implemented. When WDD > 1° days the frost damage function is triggered, leading to crop damage by transferring
leaf carbon (amount scaled by the survival probability (1 -$f_{surv}$)) to the soil carbon litter pool.

A more detailed description of these routines can be found in the source literature Lu et al. (2017) and references therein.

### 2.2.2    Crop specific parameterization

Table 1: CFT specific phenology and CN allocation parameters.

| Parameter | CLM variable name | Units |
|---|---|---|
| *Phenology* | | |
| Minimum planting date for the Northern Hemisphere | min_NH_planting_date | MMDD |
| Maximum planting date for the Northern Hemisphere | max_NH_planting_date | MMDD |
| Average 5 day daily temperature needed for planting | planting_temp | K |
| Average 5 day daily minimum temperature needed for planting | min_planting_temp | K |
| Minimum growing degree days | gddmin | °days |
| Maximum number of days to maturity | mxmat | Days |
| Growing Degree Days for maturity | hygdd | °days |
| Base Temperature for GDD | baset | °C |
| Maximum Temperature for GDD | mxtmp | °C |
| Percentage of GDD for maturity to enter phase 3 | lfemerg | % GDDmat |
| Percentage of GDD for maturity to enter phase 4 | grnfill | % GDDmat |
| Canopy top coefficient | ztopmax | M |
| Maximum Leaf Area Index | laimx | m$^2$/m$^2$ |
| Specific Leaf Area | slatop | m$^2$/gC |
| *CN ratios and allocation* | | |
| Leaf C/N | leafcn | gC/gN |
| Minimum leaf C/N | leafcn_min | gC/gN |
| Maximum leaf C/N | leafcn_max | gC/gN |
| Fine root C/N | frootcn | gC/gN |
| Grain C/N | graincn | gC/gN |

| Fraction of leaf N in Rubisco | | flnr | fraction/gNm$^{-2}$ |

In order to yield a reasonable representation of agricultural areas on the regional scale in future studies, the default parameter set was extended with specific crop parameters for sugar beet, potatoes, and winter wheat based on the characteristics of our study sites to better fit the observed plant phenology and energy fluxes at the simulation sites. The CTFs sugar beet and potatoes are merged to the spring wheat CFT on the default parameter scheme due to the lack of crop specific parameters for these crops. For winter wheat there is a pre-existing default parameter set available in CLM5. However, this default parameterization performed poorly in representing the crop phenology for the evaluated study sites in this study. This was also reported in an earlier study by Lu et al. (2017). Thus, crop specific parameters were added for sugar beet, potatoes and winter wheat. The parameters to be modified were selected taking into account the sensitivity analysis and parameter estimation studies by Post et al. (2017) (for version 4.5), Cheng et al. (2020) and Fisher et al. (2019) (for version 5.0). Key parameters as identified by previous studies (Sulis et al., 2015; Post et al., 2017; Lu et al., 2017; Fisher et al., 2019; Cheng et al., 2020) are listed in Table 1. These parameters were added with values from the literature or site-specific observations to match observed values. General phenology parameters such as the maximum canopy height, planting temperatures, maximum LAI, maximum and minimum planting dates and days for growing were adjusted according to field data including planting and harvest dates. A list of plant types, planting and harvest dates is provided in Table A1. C/N ratios in leaves and roots for wheat and sugar beet were adapted from Whitmore and Groot (1997), Gan et al. (2011), Sánchez-Sastre et al. (2018) and Zheng et al. (2018). The specific leaf area (slatop) and the fraction of leaf N in Rubisco (flnr) for sugar beet and winter wheatwere taken from Sulis et al. (2015) and references therein and adopted also for potatoes.

Table A2 provides a full list of default and newly added crop specific parameters for the CFTs temperate corn, spring wheat, sugar beet, potatoes and winter wheat.

### 2.2.3    Cover cropping and crop rotation scheme

The effect of cover crops on the physical and biogeochemical properties of the land surface alters latent heat flux, albedo and soil carbon and nitrogen storage and can potentially impact local and regional climate (Sainju et al., 2003; Lobell et al., 2006; Möller and Reents, 2009; Plaza-Bonilla et al., 2015; Basche et al., 2016; Carrer et al., 2018; Lombardozzi et al., 2018; Hunter et al., 2019).

In the default BGC phenology, the growth algorithm starts in the beginning of each year, when the crop is not alive on the specific patch. Furthermore, the CLM structure does not allow multiple CFTs to coexist on the same column so that multiple planting phases related to cover cropping over winter months or crop rotations with winter and summer crops, both being very common practices in Europe and worldwide, cannot be accounted for. This might also be an issue when representing ecosystems where agricultural management practices involve multiple sowing and harvest cycles in accordance with the monsoon season (e.g. India). Therefore, a cover cropping subroutine was implemented in the BGC phenology module that affects the onset/offset (crop cycle/fallow) algorithm to allow a second onset period (crop cycle) on the same column.

A cover crop flag was introduced in the parameter file and in the source code. This flag can be set for any CFT in the parameter file and calls the cover-cropping subroutine when it is set to true (covercrop_flag $\neq$ 0). This allows a flexible handling of this option as well as an application on a larger scale. With this modification, the onset period can start again within one simulation year for another (or the same) CFT. For example, when the maturity

of the crop is reached and it has been harvested, the model would by default switch to the next stage (phase 4)
where the crop is not alive and the offset (fallow) period begins. The next onset period and GDD accumulation for planting would then start in the subsequent simulation year. In our modified CLM5 version, the cover-cropping subroutine is called before entering into the offset period when the cover-crop flag for the current CFT is set to true. In the cover-cropping subroutine, the CFT is then changed according to a predefined rotation scheme and another onset period and GDD accumulation for planting is initialized.

A common practice is to plough the cover crops into the soil instead of removing their biomass from the field. We simulated this by relocating the biomass of the crop into the litter pool instead of the grain product pool upon harvest using the use_grainproduct flag described below (Eq. 7).

Individual crop rotation schemes were customized within the code and depend on the currently planted crop type. For example, if a simulation starts with a crop coverage of spring wheat specified in the surface file, the new
subroutine is called after harvest of the crop. Within the subroutine, the CFT is then changed to the next crop, e.g. sugar beet. Again, after the harvest of this crop, e.g. sugar beet, the CFT is again changed to the next crop and so on. When the CFT is changed back to spring wheat, the rotation cycle starts again. This rotation is defined in a repetitive sequence based on the harvested CFT and its harvest date:

$$\text{if } harvdate(p) \geq hd_1 \text{ and } ivt(p) = crop_1 \text{ then}$$
$$ivt(p) = crop_2$$
$$croplive(p) = false$$
$$idop(p) = not\_planted$$
$$use\_grainproduct = true$$
$$\text{else if } harvdate(p) \geq hd_2 \text{ and } ivt(p) = crop_2 \text{ then}$$
$$ivt(p) = crop_3$$
$$croplive(p) = false$$
$$idop(p) = not\_planted$$
$$use\_grainproduct = true \tag{7}$$

where harvdate is the harvest day of the current simulation year and hd is the customizable harvest date of the
respective CFT, $p$ is the simulated patch on the model grid, ivt is the simulated CFT, $crop_{1\text{-}3}$ represent the user-specified CFTs to the rotated, idop is the planting day and use_grainproduct is a flag to define whether the grain carbon of simulated crop is to be harvested into the food pool or not. If this flag is set to false, the plant carbon and nitrogen are transferred to the soil litter pool and not allocated to the food product pool upon harvest of the crop.

The actual rotation of crop types can be user-customized by defining the variables hd and $crop_x$ in a list (e.g. $hd_1$ = 150 [day of year], $crop_1$= spring wheat, etc.). By including the harvest date as a dependency, it is also possible to simulate the planting of cover crops based on harvest date thresholds. A user-defined maximum harvest date for any specific cash crop can define whether a cover crop would be planted or not. This technique can be beneficial to study the effects of conceptual cover cropping scenarios on regional scales. The possibility to change the CFT
within the same year represents a significant improvement in flexibility, as CLM5 only permitted land use changes at the beginning of every year. In order to simulate cover cropping at our study site DE-RuS, we implemented a new CFT for a greening mix cover crop (or $covercrop_1$).

## 2.3    Study sites and validation data

Table 2: ICOS and TERENO cropland study site location coordinates and altitude (Alt.), soil types, Köppen-Geiger climate classification (Peel et al., 2007), mean annual temperature (T), mean annual precipitation amounts (P) and reference. Textural fractions for the top soil layers (up to 50 cm) at each study site are provided in Table A3.

| Site/ID | Project | Location | Alt. [msl] | Soil type | Climate | T [°C]* | P [mm/a]* | Ref. |
|---|---|---|---|---|---|---|---|---|
| Selhausen DE-RuS | TERENO ICOS | 50.865°N 6.447°E | 104.5 | Luvisol | Cfb - temperate maritime | 9.9 | 698 | Ney et al. (2017) |
| Merzenhausen DE-RuM | TERENO | 50.930°N 6.297°E | 100 | Cambisol | Cfb - temperate maritime | 9.9 | 698 | Bogena et al. (2018) |
| Klingenberg DE-Kli | ICOS | 50.893°N 13.522°E | 478 | Gleysoil | Cfb – suboceanic, subcontinental | 8.1 | 766 | Grünwald (personal communication, 2020) |
| Lonzée BE-Lon | ICOS | 50.553°N 4.746°E | 167 | Luvisol | Cfb - temperate maritime | 10 | 800 | Buysse et al.(2017) |

\* Reference periods: 1961-2010 for DE-RuS (adapted also for DE-RuM), 2005-2019 for DE-Kli and 2004-2017 for BE-Lon.

The CLM5 model was set up for four European cropland sites: Selhausen, Merzenhausen, Klingenberg and Lonzée (Fig. 1). These sites were selected mainly for their excellent continuous measurements of surface energy fluxes.

Selhausen (50.86589°N, 6.44712°E) is part of the TERENO Rur Hydrological Observatory (Bogena at al., 2018) as well as the Integrated Carbon Observation System (ICOS, 2020). The test site covers an area of approximately 1 km x 1 km and is located in the catchment of the Rur river (Bogena et al., 2018). Selhausen had a crop rotation of sugar beet (*Beta vulgaris*), winter wheat (*Triticum aestivum*) and winter barley (*Hordeum vulgare*), fewer times also rapeseed (*Brassica napus*) and potatoes (*Solanum tuberosum*) from 2015 to 2019. Cover crops such as oilseed radish or cover crop mixes are planted occasionally between two main crop rotations. Continuous records of meteorological variables, soil specific observations, as well as greenhouse gas and energy fluxes are available for Selhausen since 2011. Regular LAI measurements are available since 2016 (Ney and Graf, 2018).

Merzenhausen (50.93033°N, 6.29747°E) is located at approximately 14 km from Selhausen and is also part of the TERENO Rur Hydrological Observatory. The crop rotation of the site includes sugar beet (*Beta vulgaris*), winter wheat (*Triticum aestivum*), winter barley (*Hordeum vulgare*), rape seed (*Brassica napus*) and occasionally catch cover crop mixes. For Merzenhausen, continuous records of meteorological variables, soil specific observations and energy fluxes are available since 2011 and LAI measurements from 2016 – 2018.

Klingenberg (50.89306°N, 13.52238°E) is an ICOS cropland site located in the mountain foreland of the Erzgebirge that is operated by the Technical University Dresden (TU Dresden) (ICOS, 2020; Prescher et al., 2010). The site is characterized as managed cropland with a 5-year planting rotation of rapeseed (*Brassica napus*), winter wheat (*Triticum aestivum*), maize (*Zea mays*), spring and winter barley (*Hordeum vulgare*) (Kutsch et al., 2010). Since 2004, data on ecosystem fluxes (including net ecosystem and net biome productivity), meteorological variables and soil observations are collected. Furthermore, biomass observations and agricultural management information are available for this site.

The cropland site Lonzée (50.553°N 4.746°E) in Belgium is also part of ICOS (Buysse et al., 2017). It has been planted in a four-year rotation cycle with sugar beet (*Beta vulgaris*), winter wheat (*Triticum aestivum*), potato (*Solanum tuberosum*) since 2000 with Mustard as a cover crop after winter wheat harvest (Moureaux, 2006; Moureaux et al., 2008). For Lonzée, continuous records of meteorological variables, EC flux data and LAI (GLAI and GAI) measurements are available from 2004 onwards. General information on the ICOS study sites such as climatic conditions, soil types etc. is provided on the ICOS Carbon Portal under the respective site codes (ICOS, 2020).

At all sites, the application of mineral fertilizer and herbicides/pesticides as well as occasional application of organic fertilizer is regular management practice.

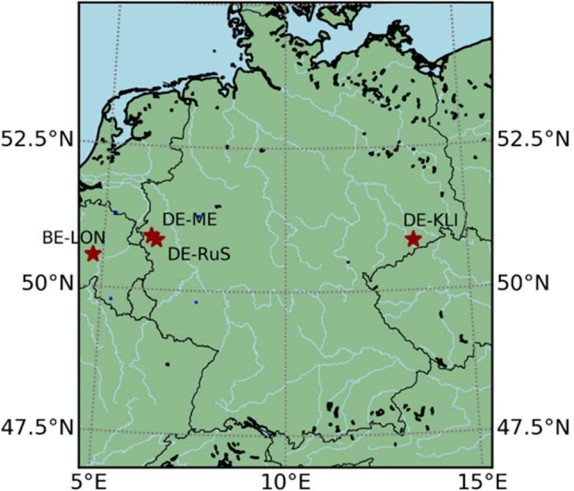

Figure 1: ICOS and TERENO cropland study sites Selhausen (DE-RuS), Merzenhausen (DE-RuM), Klingenberg (DE-Kli) and Lonzée (BE-Lon)

Station data required to force CLM, i.e. meteorological variables (see following section) were measured as block averages over 10 minutes or at higher resolutions, gap-filled using linear statistical relations to nearby stations where possible (Graf, 2017), or otherwise, by marginal distribution sampling within the software package REddyProc (Wutzler et al., 2018). Fluxes required for model validation (i.e. net ecosystem $CO_2$ exchange (NEE), latent heat flux (LE), sensible heat flux (H), soil heat flux (G) and gross primary production (GPP)) and net radiation (Rn), were either measured (G and $R_n$) or computed from turbulent raw measurements (frequency $\geq 10$ $s^{-1}$) using the eddy-covariance method, for 30-minute block averages by the site PIs. Subsequently, gaps were filled and GPP estimated from NEE using REddyProc (Wutzler et al., 2018). More details on quality control, filling of longer gaps and by nearby stations, correction of soil heat flux and energy balance closure analysis are given in Graf et al. (*in review*) and specifically for DE-RuS and DE-RuM including LAI measurements in Reichenau et al. (2020). The long-term annual energy balance closures of the sites DE-RuS, DE-Kli and BE-Lon were approximately 79%, 77% and 76%, respectively, according to analyses in Graf et al. (*in review*) and 76% at DE-RuM according to an earlier study by Eder et al. (2015). All half-hourly meteorological and flux data were aggregated to hourly averages to match our customized CLM forcing time step.

Site-specific measurement records of latent and sensible heat fluxes, net ecosystem exchange (NEE), LAI, soil temperature and soil moisture were used as validation data for the simulation runs.

Forcing variables were always used in gap-filled form, while validation variables were used in un-filled, quality-filtered form.

## 3    Experimental design and analyses

### 3.1    Model implementation

For the single point study sites, CLM was run in point mode with only one grid cell and forced with site specific hourly meteorological data. The annual fertilization amounts at the single point study sites were adjusted according to documented amounts of applied fertilizer that ranged between 12 and 20 $gNm^{-2}$. In CLM5, the potential

photosynthetic capacity as well as the total amount of assimilated carbon during the phenology stages are regulated by the availability of soil nitrogen (Lawrence et al., 2018). With modern fertilization practices in Europe, nitrogen is not assumed to be a limiting factor for the studied sites.

In order to balance ecosystem carbon and nitrogen pools, gross primary production and total water storage in the system, a spin-up is required (Lawrence et al., 2018). An accelerated decomposition spin-up of 600 years and an

additional spin-up of 400 years was conducted for each site with the BGC-Crop module (Lawrence et al., 2018; Thornton and Rosenbloom, 2005). The simulated conditions at the end of the spin-up were then used as initial conditions for the following simulations.

In order to test the winter wheat representation, several simulations were conducted for all winter wheat years at the sites DE-RuS, DE-RuM, DE-Kli and BE-Lon. In a first step, the impact of each modification was assessed

individually by simulating one winter wheat year at the site DE-RuS using four different model configurations: (1) the default model and default parameter set (control), (2) the default model with the new parameter set (control + crop specific), (3) the extended winter wheat model with the default parameter set (new routine), and (4) the extended winter wheat model with the new parameter set (new routine + crop specific). Further evaluations for the other study sites and years were conducted for the combined winter wheat modifications CLM_WW (extended

model with winter wheat subroutines and new crop specific parameterization) in comparison to control simulations (default model configuration and default parameterization of winter wheat).

For the evaluation of the crop specific parameter sets for sugar beet and potatoes, simulations were run with the new parameterizations at the sites DE-RuS and BE-Lon over several years. For both sites, control simulations were conducted without the new parameter set, in which both CFTs sugar beet and potatoes are simulated as a spring

wheat by default. Furthermore, an evaluation of the default parameterization for the CFT temperate corn at the site DE-Kli is included in the supplementary material (Fig. S1, Table S1).

The cover cropping and crop rotation scheme was tested for two practical cases at DE-RuS. From 2016 to 2017, planting was altered at DE-RuS from barley (here represented by the CFT for spring wheat) in 2016 to sugar beet in 2017 with a greening mix cover crop in between (winter months 2016/2017). In order to simulate this common

cover cropping practice, we implemented a new CFT for a greening mix cover crop (or covercrop$_1$). For the years 2017 to 2019 at DE-RuS, the subroutines ability to simulate realistic crop rotation cycles was tested by changing the simulated CFT from sugar beet (2017) to winter wheat (2017-2018) and then to potatoes (2019). In this step, simulations were run with the previously tested crop specific parameterizations for sugar beet, potatoes and winter wheat. Simulation results were again compared to a control simulation run, where a consecutive growth of spring

wheat is simulated.

## 3.2    Evaluation of model performance

For statistical evaluation of the model results, the root mean square error (RMSE), the bias (BIAS) and the Pearson correlation (r) were chosen as performance metrics:

$$RMSE = \sqrt{\frac{1}{n}\sum_{i=1}^{n}(X_i - X_{obs,i})^2}, \tag{8}$$

$$BIAS = \sum_{i=1}^{n}(X_i - X_{obs,i})/\sum_{i=1}^{n}(X_{obs,i}), \tag{9}$$

$$r = (\frac{1}{n}\sum_{i=1}^{n}(X_{obs,i} - \mu_{obs}) * (X_i - \mu_{sim}))/(\sigma_{sim} * \sigma_{obs}), \tag{10}$$

where $i$ is time step and $n$ the total number of time steps, $X_i$ and $X_{obs,i}$ are the simulated and the observed values at every time step with $\mu_{sim}$ and $\mu_{obs}$ being the respective mean values. The standard deviation of simulation results and measurement data are represented by $\sigma_{sim}$ and $\sigma_{obs}$ respectively.

The statistical evaluation was conducted for daily simulation output and daily observation data for the variables NEE, LE, H and Rn.

## 4 Results

### 4.1 Winter cereal representation

The impact of the new winter wheat specific parameterization and the new winter wheat routine, as well as the
combination of both is illustrated in Fig. 2. Here we show simulated LAI for the default model and default parameter set (control), the default model with the new parameter set (control + crop specific), the extended winter wheat model with the default parameter set (new routines) and the extended winter wheat model with the new parameter set (new routines + crop specific).

Using only the new crop specific parameter set with the default model configuration resulted in slightly higher
LAI values compared to the control run but did not reach the observed maximum LAI values and the growth cycle duration. The implementation of the winter wheat subroutines using the default parameter set led to a more realistic reproduction of the growth cycle duration compared to the control run, but did not yield good correspondence with observed LAI magnitudes. The combination of the new crop specific parameter set and the new winter wheat subroutines resulted in the most realistic LAI dynamics (Fig. 2). As previously described by Lu et al. (2017), the
default vernalization routine reaches a factor of 1 (fully vernalized) shortly after planting when the first frost occurs. This induced an unrealistically early commencement of the grain fill stage within two months after planting in the control run (November or December). The default vernalization also resulted in peak LAI occurring too early in the year, leading to significantly lower photosynthesis compared to the observations. This also applies to the implementation of the new crop-specific parameter set, which generally leads to slightly higher LAI values.
In the extended winter wheat model, the adapted vernalization routine produces lower initial vernalization factors which reduce the growing degree days. This leads to later onset of the leaf emergence and grain fill stage and allows a more realistic representation of the LAI cycle and peak in combination with the new crop specific parameterization.

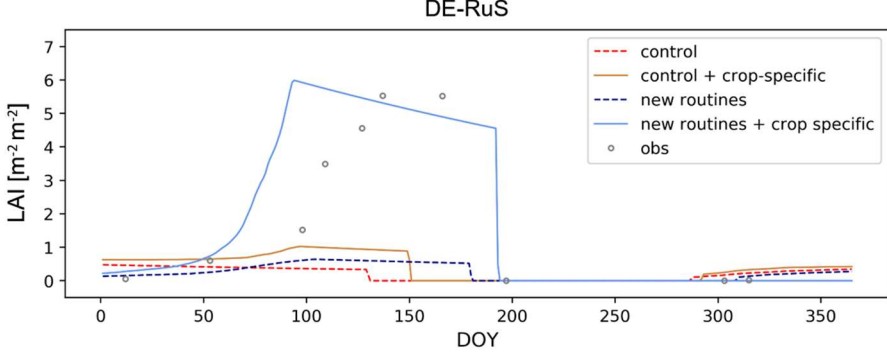

Figure 2: Daily simulation results for the LAI, simulated with default model and the default parameter set (control), the default model with new parameter set (control + crop specific), the extended winter wheat model with default parameterization (new

routines) and the extended model with the new parameter set (new routines + crop specific), compared to point observations for a winter wheat year at DE-RuS.

In further evaluations, the combined winter wheat package, including the new crop specific parameterization and
the extended winter wheat subroutines is implemented in CLM_WW simulations and compared to control runs (Fig. 3). For all study sites and simulation years, CLM_WW simulations resulted in a much better representation of the growth cycle and corresponding seasonal LAI variation and magnitudes compared to control simulations (Fig. 3). Also, the temporal pattern of energy fluxes and NEE were improved with CLM_WW compared to the control run.

In general, CLM_WW yielded LAI peak magnitudes similar to observations at the sites BE-Lon, DE-RuS and DE-RuM (Fig. 3). For DE-Kli, site-specific observations of the LAI were not available, but simulated LAI magnitudes for DE-Kli using CLM_WW are similar to those for BE-Lon. For the BE-Lon site, CLM_WW simulated peak LAI magnitudes are close to the observations. An exception is the year of 2015, where CLM_WW underestimated the unusually high LAI values observed in May and June, which ranged from 5.40 to 6.38 $m^2/m^2$. For BE-Lon,
faster growth was simulated in the early growing stage of winter wheat, resulting in a more gradual increase in LAI compared to the other sites (Figure 3). This is related to higher air temperatures at BE-Lon early in the growing stage (especially in February) that enabled more simulated growth compared to the other sites.

Overall, the LAI peak simulated with CLM_WW occurred about one month earlier than observed, suggesting that maturation was reached too early. This is also reflected in the simulated CLM_WW harvest dates that are
approximately one month earlier than the recorded dates (Table 3). While the planting date is the same for the control run and the CLM_WW simulations, CLM_WW generally resulted in a better match of simulated and recorded harvest dates (1.5 to 2 months later than control run).

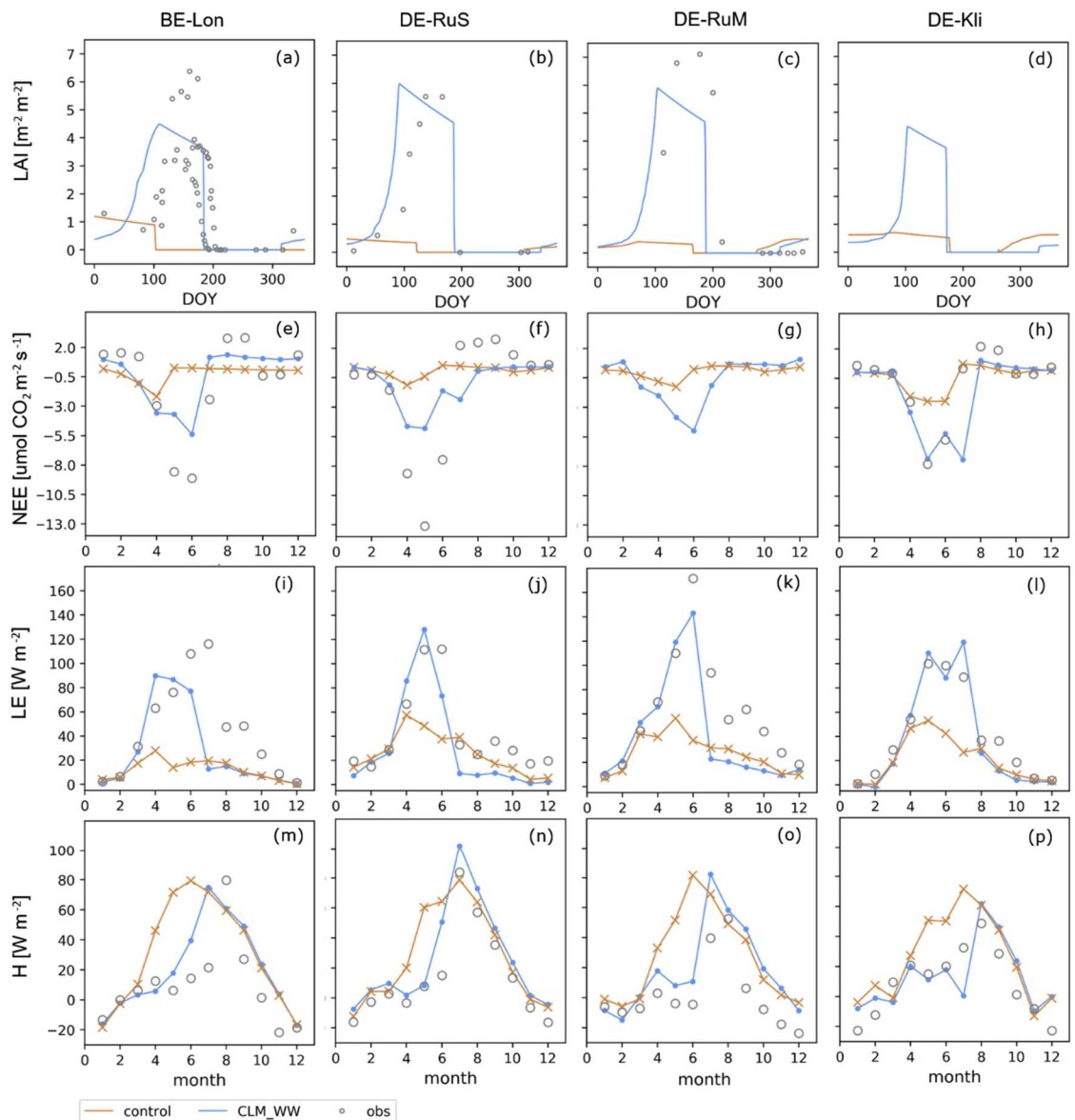

Figure 3: Simulation results of (a-d) LAI and simulation results averaged for each month of (e-h) NEE, (i-l) LE, and (m-p) H for all winter wheat years (see Table 3) at the sites (from left to right) BE-Lon, DE-RuS, DE-RuM and DE-Kli. Simulation results from the new routine with crop specific parameterization – CLM_WW (blue) are compared to control simulations (orange) and available site observations (grey) of LAI (all available point observations plotted) and fluxes (averaged over all respective years and for each month respectively). Corresponding performance statistics for daily simulation results during the crop growth cycle are listed in Table 4.

The correlation of simulated grain yield and site records was significantly improved by up to 87 % in CLM_WW simulations compared to the control run. At the DE-RuS site, CLM_WW resulted in a grain yield of 9.15 t/ha that is very close to the observed value of 9.2 t/ha, while grain yield is strongly underestimated in the control run (1.17 t/ha). For DE-Kli, the CLM_WW simulated crop yield matched the recorded yield data very well for the year 2016 and was overestimated for 2011 by approximately 16 %. The control run resulted in an underestimation of yield by more than 80 % (Fig. 4, Table 3). For BE-Lon the simulated crop yield is underestimated compared to site harvest records (Fig. 4, Table 3). While CLM_D simulations underestimated the grain yield by approximately 85 – 90 %, CLM_WW underestimated yield by only 18 - 36 % at BE-Lon. The simulated yields by CLM_WW for the individual years show only minimal variations with values from 8.12 to 8.16 t/ha, while the measured yields

ranged from 9.92 to 12.88 t/ha, indicating that CLM did not capture the inter-annual yield variation very well (Table 3).

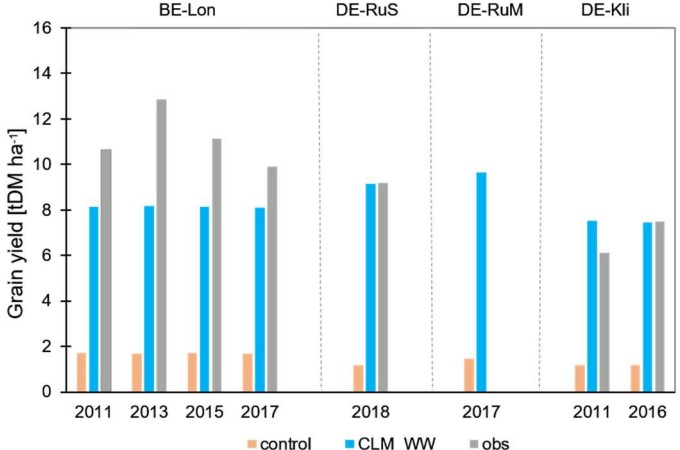

Figure 4: Annual grain yield [tDM/ha] simulated with the control run (orange) and the extended winter wheat model with crop specific parameterization (blue), compared to recorded harvest yields (grey) for all simulated winter wheat years (indicated on the x axis) at the sites BE-Lon, DE-RuS, DE-RuM and DE-Kli.

Table 3: Simulated annual planting and harvest dates and grain yield [tDM/ha] by CLM_WW and CLM_D simulations (calculated using the peak daily grain carbon throughout the growth cycle) compared to recorded harvest dates and grain yield (Obs) for all simulated winter wheat years at the sites BE-Lon, DE-RuS, DE-RuM and DE-Kli. For CLM simulation results, grain yield is calculated from grain carbon which is assumed to be 45 % of the total dry weight.

| Year | Source | Planting date | Harvest date | Grain Yield [tDM/ha] |
|------|--------|---------------|--------------|----------------------|
| | | *BE-Lon* | | |
| | CLM_D | 11.09.2010 | 10.05.2011 | 1.71 |
| 2010/2011 | CLM_WW | 11.09.2010 | 05.07.2011 | 8.14 |
| | *Obs* | *14.10.2010* | *16.08.2011* | *10.64\** |
| | CLM_D | 12.09.2012 | 19.04.2013 | 1.68 |
| 2012/2013 | CLM_WW | 12.09.2012 | 25.06.2013 | 8.16 |
| | *Obs* | *25.10.2012* | *12.08.2013* | *12.88* |
| | CLM_D | 09.09.2014 | 20.04.2015 | 1.71 |
| 2014/2015 | CLM_WW | 09.09.2014 | 01.07.2015 | 8.15 |
| | *Obs* | *15.10.2014* | *02.08.2015* | *11.13* |
| | CLM_D | 11.09.2016 | 02.05.2017 | 1.68 |
| 2016/2017 | CLM_WW | 11.09.2016 | 24.07.2017 | 8.12 |
| | *Obs* | *29.10.2016* | *30.07.2017* | *9.92* |
| | | *DE-RuS* | | |
| | CLM_D | 29.09.2017 | 17.05.2018 | 1.17 |
| 2017/2018 | CLM_WW | 29.09.2017 | 27.06.2018 | 9.15 |
| | *Obs* | *25.10.2017* | *16.07.2018* | *9.2* |
| | | *DE-RuM* | | |
| | CLM_D | 27.09.2016 | 15.05.2017 | 1.45 |
| 2016/2017 | CLM_WW | 27.09.2016 | 30.06.2017 | 9.65 |
| | *Obs* | *17.10.2016* | *22.07.2017* | *-* |
| | | *DE-Kli* | | |
| | CLM_D | 15.09.2009 | 23.07.2011 | 1.19 |
| 2010/2011 | CLM_WW | 15.09.2009 | 11.08.2011 | 7.53 |
| | *Obs* | *02.10.2010* | *22.08.2011* | *6.12* |
| | CLM_D | 17.09.2015 | 24.07.2016 | 1.17 |
| 2015/2016 | CLM_WW | 17.09.2015 | 28.07.2016 | 7.44 |
| | *Obs* | *18.09.2015* | *24.08.2016* | *7.48* |

\*: Grain yield estimated from 18.09 t/ha total biomass (stem and ear) yield according to stem and ear (grain) biomass yield ratios measured for other winter wheat years at the same site.

Table 4: Bias, root mean square error (RMSE) and Pearson correlation coefficient (r) for the control run and CLM_WW simulated daily NEE [umol CO$_2$ W m$^{-2}$ s$^{-1}$], LE [W m$^{-2}$], H [W m$^{-2}$] and Rn [W m$^{-2}$] at the sites BE-Lon, DE-RuS, DE-RuM and DE-Kli respectively. Values were calculated over the time period between recorded planting and harvest dates (averaged over all winter wheat years at each site) using simulation output and observation data at daily time step.

| CFT | WINTERWHEAT | | | | | | | |
|---|---|---|---|---|---|---|---|---|
| Site | BE-Lon | | DE-RuS | | DE-RuM | | DE-Kli | |
| Year(s) | 2010/2011 2012/2013 2014/2015 2016/2017 | | 2017/2018 | | 2016/2017 | | 2010/2011 2015/2016 | |
| Model | control | CLM_WW | control | CLM_WW | control | CLM_WW | control | CLM_WW |
| | | | | *NEE* | | | | |
| Bias | -0.87 | -0.37 | -1.01 | -0.61 | - | - | -0.56 | 0.50 |
| RMSE | 6.34 | 4.96 | 7.73 | 7.58 | - | - | 3.80 | 3.27 |
| r | -0.13 | 0.46 | 0.21 | 0.33 | - | - | 0.29 | 0.56 |
| | | | | *LE* | | | | |
| Bias | -0.72 | -0.13 | -0.47 | -0.23 | -0.55 | -0.09 | -0.47 | -0.77 |
| RMSE | 61.96 | 50.73 | 52.47 | 52.65 | 67.17 | 48.67 | 44.64 | 56.75 |
| r | 0.35 | 0.46 | 0.21 | 0.24 | 0.50 | 0.67 | 0.61 | 0.71 |
| | | | | *H* | | | | |
| Bias | 5.56 | 1.35 | 4.24 | 1.70 | -8.49 | -2.74 | 4.99 | 3.10 |
| RMSE | 45.97 | 27.63 | 40.93 | 39.94 | 47.26 | 32.81 | 49.30 | 35.08 |
| r | 0.42 | 0.50 | 0.45 | 0.48 | 0.21 | 0.36 | 0.47 | 0.63 |
| | | | | *Rn* | | | | |
| Bias | -0.18 | -0.05 | -0.17 | -0.13 | -0.09 | 0.08 | -0.03 | -0.09 |
| RMSE | 36.11 | 38.01 | 47.28 | 45.15 | 37.34 | 46.43 | 45.17 | 44.49 |
| r | 0.80 | 0.81 | 0.68 | 0.69 | 0.78 | 0.97 | 0.71 | 0.73 |

Overall, the better representation of the winter wheat growing cycle by CLM_WW can also be inferred from the simulated surface energy fluxes (Fig. 3). In terms of net radiation, both CLM_WW and the control run are very close to the observations (Table 4). However, CLM_WW was able to better capture seasonal variations of surface energy fluxes during the growing cycle of the crop (Fig. 3). The correlation coefficients for the energy fluxes (LE, H and Rn) calculated over the period from planting to harvest date for daily simulation results and daily observation data improved for all sites (Table 4). Highest correlations were reached for the sites DE-Kli with r values of 0.62 and 0.71 and for BE-Lon with r values of 0.5 and 0.46 for sensible heat and latent heat flux respectively (Table 4). Due to the simulated LAI peak being too early, latent heat flux is underestimated by CLM_WW (Fig. 3, Table 4). The high latent heat fluxes measured at BE-Lon and DE-Kli in the later months of the year (from day 220 onwards) reflect the growth of a cover crop. At both the BE-Lon site as well as at the DE-Kli site, cover crops are typically sown after harvest of winter wheat (mustard at BE-Lon, radish and brassica at DE-Kli), and they strongly affect surface energy fluxes later in the year. In contrast, in the control simulations, as well as in CLM_WW, the crop field were simulated as fallow after the harvest of winter wheat (Fig. 3, Table A1). While the correlation of the latent and sensible heat flux during the growing cycle of the crop is generally increased with the CLM_WW model, the overall annual correlation is still relatively poor due to the influence of cover cropping and poor representation of post-harvest field conditions (annual performance metrics are included in the supplementary material, Table S3). Furthermore, CLM_WW was generally better able to match NEE observations compared to control runs, partly due to the better representation of the seasonal LAI variations (Fig. 3). During the growing season of winter wheat, the negative peak in NEE, coincides with the peak in LAI. Negative NEE values indicate a carbon sink and happen when the crop gains more carbon through photosynthesis than is lost through respiration. Correlation improved (comparing CLM_WW to the control run) from 0.13 to 0.46 for BE-Lon, from 0.21 to 0.33 for DE-RuS and from 0.29 to 0.56 for DE-Kli. The resulting correlation for CLM_WW simulations is still relatively low due to an underestimation of the cumulative monthly NEE during seasons with high NEE at BE-Lon and DE-RuS. For

DE-Kli, CLM_WW was able to match NEE observed at peak LAI very well, but late seasonal NEE (July), shortly before harvest, is overestimated by CLM_WW resulting in a low overall agreement with observation data. Furthermore, post-harvest field observations at BE-Lon, DE-RuS and DE-Kli indicate that heterotrophic respiration from soil organic matter and litter results in a carbon source which is not simulated well in CLM (no GPP, near zero NEE) (Fig. 3). This poor representation of post-harvest field conditions is reflected in low correlations over the whole year (Table S3).

## 4.2    Crop specific parameterization of sugar beet and potatoes

The crop specific parameter sets were tested for several years with sugar beet and potatoes planting at BE-Lon and DE-RuS respectively. The performance in reproducing seasonal variations and magnitudes of energy fluxes was strongly improved with the crop specific parameterization. Correspondingly, simulations with the crop specific parameter sets for both sugar beet and potatoes were able to reasonably capture seasonal variations and peak values of LAI as well as growth cycle length and harvest time (Fig. 5, Fig. 6). The control run in CLM uses the spring wheat parameterization for these crop types and therefore reproduced the growth cycle and seasonal LAI of spring wheat, while simulations using the crop-specific potato and sugar beet parameterizations better captured harvest date and growth cycle of these crops.

The improved growth cycle representation with crop specific parameters also led to more accurate simulation of energy fluxes. For sugar beet at BE-Lon, the latent heat flux at peak LAI corresponds well with observed values while being underestimated before and after peak LAI and hence the sensible heat flux is overestimated at these times (Fig. 5). Seasonal variations of energy fluxes and magnitudes were also captured much better in simulations with the new parameterization. The simulations with crop specific parameters show slightly better net radiation correlations for both the sugar beet and potato CFTs at each site, compared to the control run (Table 5). The correlation between simulated and observed latent heat flux for sugar beet were strongly improved by changing the parameters (0.11 to 0.55 for DE-RuS and 0.21 to 0.55 for BE-Lon). The same is true for the simulated sensible heat flux for sugar beet (0.04 to 0.76 for DE-RuS and 0.08 to 0.51 for BE-Lon site). The NEE for the sugar beet CFT is underestimated during peak LAI periods in the control run, resulting in poorer correlations compared to latent and sensible heat flux and net radiation (Fig. 5). Simulations with the crop specific parameter set resulted in a reduction in negative bias for NEE and reached higher correlation compared to the control simulation (0.03 to 0.37 for DE-RuS and 0.05 to 0.64 for BE-Lon).

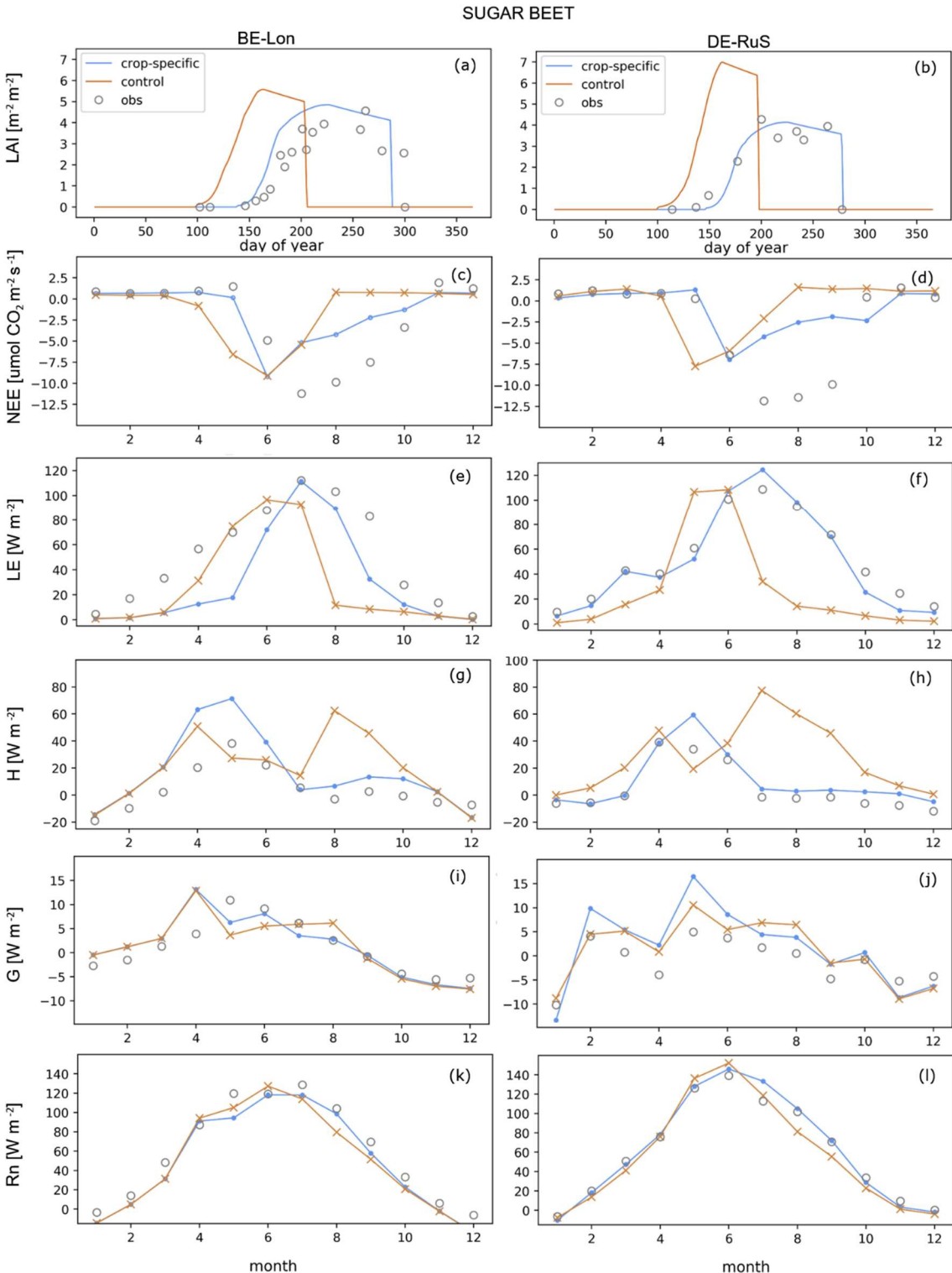

Figure 5: Simulation results of (a-b) LAI and monthly averaged simulation results of (c-d) NEE, (e-f) LE, (g-h) H, (i-j) G and (k-l) Rn for all sugar beet years (see Table 5) at the sites (left) BE-Lon and (right) DE-RuS. Simulation results for the control run (orange) and the crop specific parameter set (blue) are compared to available site observations (grey) of LAI (all available point observations plotted) and fluxes (averaged over all respective years). Corresponding performance statistics for daily simulation results are listed in Table 5.

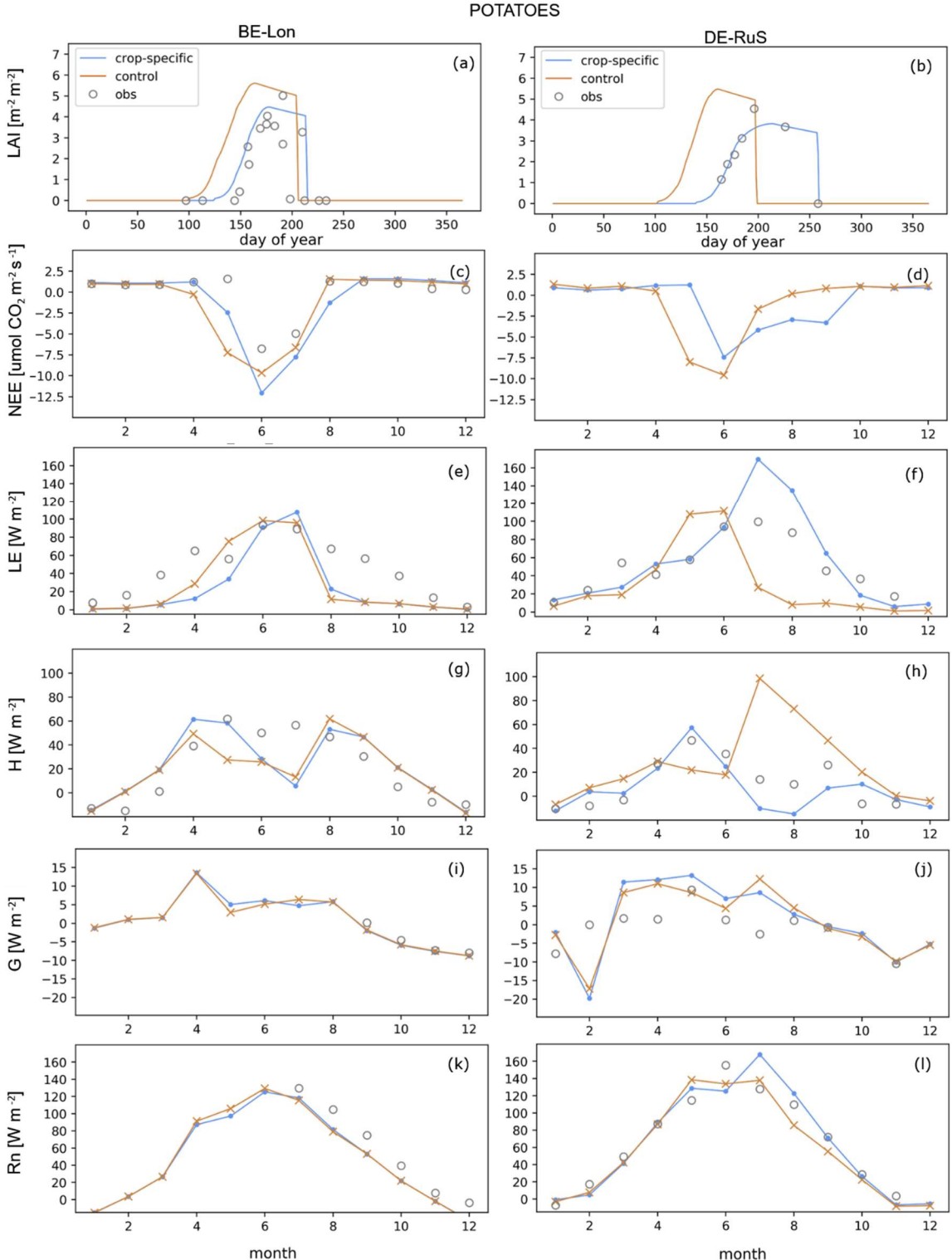

Figure 6: Simulation results of (a-b) LAI and monthly averaged simulation results of (c-d) NEE, (e-f) LE, (g-h) H, (i-j) G and (k-l) Rn for all potatoes years (see Table 5) at the sites (left) BE-Lon and (right) DE-RuS. Simulation results for the control run (orange) and the crop specific parameter set (blue) are compared to available site observations (grey) of LAI (all available observations plotted) and fluxes (averaged over all respective years). Corresponding performance statistics for daily simulation results are listed in Table 5.

Similar improvements can be observed for the new potato parameterization while the correlation of simulation results with observation data is generally lower compared to the sugar beet CFT (Fig.6, Table 5). Seasonal LAI variations, growing cycle length and corresponding energy flux variations are improved in simulations with the new parameter set. Both the latent and the sensible heat flux are strongly improved at DE-RuS with correlation coefficients of 0.54 and 0.45 respectively for CLM_WW simulations. For BE-Lon, the improvement in correlation

is slightly lower for both latent and sensible heat flux compared to DE-RuS. The seasonal variation of the NEE at BE-Lon is reasonably captured while monthly sums are overestimated with both parameterizations. Simulations of the NEE using the crop specific parameter set yielded a slightly better correlation of 0.58 compared to the control simulation that resulted in a correlation of 0.43 (Table 5).

Table 5: Bias, root mean square error (RMSE) and Pearson correlation coefficient (r) for the simulated daily NEE [μmol $CO_2$
W $m^{-2}$ $s^{-1}$], LE [W $m^{-2}$], H [W $m^{-2}$] and Rn [W $m^{-2}$] ) using the crop specific parameterization (*specific*) for the CFTs sugar beet and potatoes at the sites BE-Lon and DE-RuS respectively. Results are compared to those from the control simulation runs (*control*). Values were calculated over the time period between recorded planting and harvest dates (averaged over all respective CFT years at each site) using simulation output and observation data at daily time step.

| CFT | SUGARBEET | | | | POTATOES | | | |
|---|---|---|---|---|---|---|---|---|
| Site | DE-RuS | | BE-Lon | | DE-RuS | | BE-Lon | |
| Year(s) | 2017 | | 2008 2016 | | 2019 | | 2010 2014 2018 | |
| Parameter set | *control* | *specific* | *control* | *specific* | *control* | *specific* | *control* | *specific* |
| | | | | *NEE* | | | | |
| Bias | -0.59 | -0.75 | 0.05 | -0.05 | - | - | 19.73 | 19.56 |
| RMSE | 9.1 | 5.94 | 6.19 | 3.75 | - | - | 5.24 | 5.21 |
| r | -0.03 | 0.37 | 0.05 | 0.64 | - | - | 0.43 | 0.58 |
| | | | | *LE* | | | | |
| Bias | -0.32 | 0.01 | -0.37 | -0.35 | -0.28 | 0.25 | 0.26 | 0.09 |
| RMSE | 58.44 | 24.47 | 60.09 | 48.31 | 60.94 | 50.58 | 43.41 | 40.05 |
| r | 0.11 | 0.55 | 0.21 | 0.55 | 0.01 | 0.54 | 0.5 | 0.53 |
| | | | | *H* | | | | |
| Bias | 1.65 | 0.45 | 1.73 | 1.61 | 1.01 | -0.38 | 0.5 | 0.22 |
| RMSE | 42.77 | 17.24 | 39.75 | 33.45 | 51.61 | 29.9 | 34.06 | 31.17 |
| r | -0.04 | 0.76 | -0.08 | 0.51 | -0.1 | 0.45 | 0.18 | 0.31 |
| | | | | *Rn* | | | | |
| Bias | -0.02 | 0.04 | -0.11 | -0.11 | -0.04 | 0.04 | - | - |
| RMSE | 19.74 | 15 | 37.47 | 35.87 | 48.39 | 49.88 | - | - |
| r | 0.5 | 0.51 | -0.22 | -0.22 | 0.56 | 0.57 | - | - |

### 4.3 Cover cropping and crop rotation scheme

The cover cropping scheme was tested for two fields of application: (1) simulation of a cover crop as a second crop growth onset within a single year, and (2) a more flexible crop rotation between different cash crops. In this step, simulations were run with the previously tested crop specific parameterizations for sugar beet, potatoes and
605 winter wheat and results were again compared to a control simulation run, where a consecutive growth of spring wheat is simulated.

To test the first application of the cover cropping and crop rotation scheme, we simulated the cash crop and cover crop rotation cycle at DE-RuS from 2016 to 2017 (Fig. 7). A greening mix was planted as a cover crop in between the cash crop rotation of barley (simulated using the spring wheat CFT) in 2016 and sugar beet in 2017. While
only a consecutive growth cycle of spring wheat is simulated in the control run, the new routine was able to

represent the crop rotation from barley to sugar beet in the following year as well as a cover crop in between the cash crop cycles. Both, the simulation of a cover crop and the rotation of cash crops strongly improved the representation of LAI in simulations with the new routine over multiple years, especially during winter months (Fig. 7, Fig. 8). While in control simulations, the model assumed bare field conditions with no plant growth (LAI of 0) and very low latent heat flux, the new routine simulated the planting of a cover crop in fall of 2016, which leads to an increase in latent heat flux related to increased transpiration. Statistical evaluation of the simulated latent heat flux for the time window after harvest of the first cash crop from August 2016 to April 2017 shows that with the new routine, the negative bias was reduced from 0.74 to 0.13 compared to control simulation results, resulting in an RMSE reduction by approximately 42 % (Fig. 7).

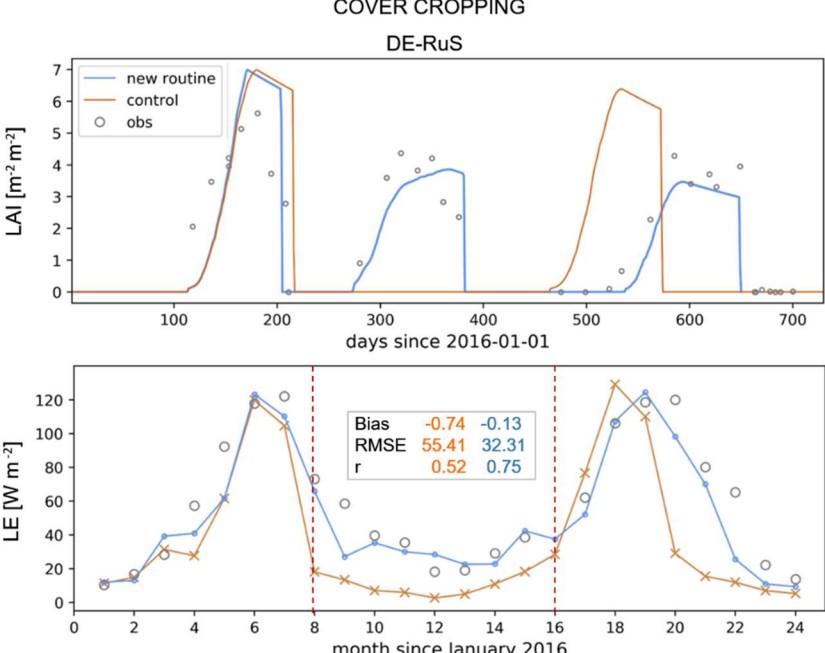

Figure 7: (Top) Simulated LAI for cover cropping at DE-RuS with a barley (2016), greening mix cover crop (2016/2017) and sugar beet (2017) using the new cover cropping subroutine (blue) in comparison to control simulation results with the default phenology algorithm of CLM5 (orange). (Bottom) Corresponding monthly averaged simulation results for the latent heat flux with respective bias, RMSE and r for the time window between the red dashed lines (calculated using simulation output and observation data at daily time step). Available observation data are plotted in grey.

For the second case (DE-RuS), which represents a higher flexibility towards cash crop rotation, we simulated the years of 2017 to 2019. Here, the crop rotation switched from sugar beet in 2017 to winter wheat in 2017/2018 to potatoes in 2019 (Fig. 8). In the control simulation, using the default CLM5 phenology algorithm, a consecutive cycle of spring wheat is simulated. The new routine was able to represent the rotation between different cash crops on the same field. This resulted in a much better correspondence of simulated LAI cycle and magnitudes with observations compared to control simulations. Statistical analysis of the latent heat flux showed an improvement of the RMSE (calculated for daily simulation output and observation data over these three years) from 43.74 to 32.94 and the correlation coefficient from 0.40 to 0.63 with the new routine. The improvement in simulated energy fluxes for each CFT individually is in accordance with the results presented in the previous chapters (4.1 and 4.2).

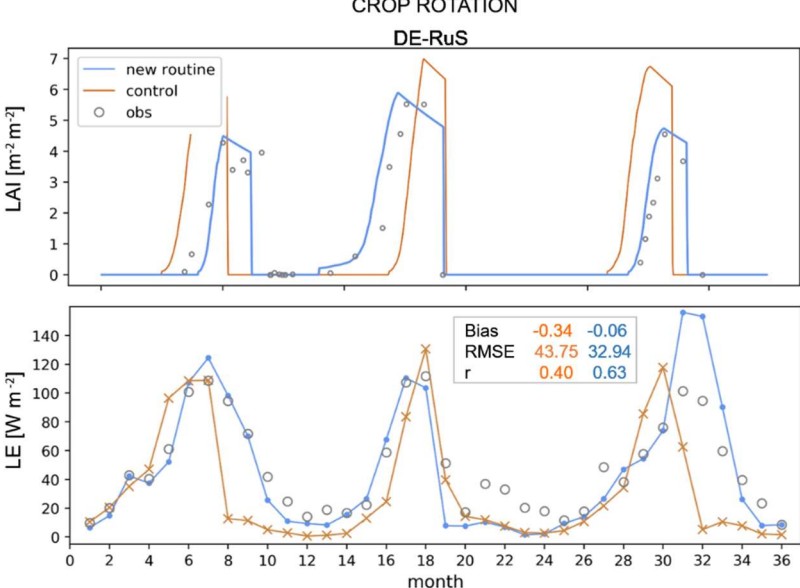

Figure 8: (Top) Simulated LAI for crop rotation from sugar beet (2017) to winter wheat (2017/2018) and to potatoes (2019) at DE-RuS using the new cover cropping subroutine (blue) in comparison to control simulation results with the default phenology algorithm of CLM5 (orange). (Bottom) Corresponding monthly averaged simulation results for the latent heat flux with respective bias, RMSE and r over the whole time interval (calculated using simulation output and observation data at daily time step). Available observation data are plotted in grey.

## 5 Discussion

All three modifications that were implemented in this study helped to improve the representation of cropland sites in CLM5. Similar to the findings of Lu et al. (2017) for CLM4.5, the implementation of their winter wheat routine resulted in a significant improvement in representing the seasonal LAI variations and surface energy fluxes during winter wheat growth. Next to maize and rice, wheat is one of the most important international food crops and among the most important cash crops in Germany (22.8 million tons winter wheat yield in 2019 nation-wide (Statista, 2020)). In Germany and other western European countries, winter cereal varieties (e.g. winter rye, barley and wheat) are more abundant than summer cereals due to climatic conditions (Palosuo et al., 2011; Semenov and Shewry, 2011; Thaler et al., 2012). With an average annual winter wheat yield of around 20 Mt/a for Germany, an improvement of 87 % in simulated yield with CLM_WW compared to the default model (as observed at the DE-Rus site in 2018) could result in a difference of several tens of millions of tons in total predicted annual yield on a nation-wide scale.

Despite the general improvement of winter wheat growth and yield simulated with the modified CLM_WW, there is still potential in further increasing the flexibility towards simulating different crop varieties and management practices. Due to the phenology algorithm of CLM5, a low simulated LAI can indicate a lower grain yield due to low biomass growth. Accordingly, the higher simulated LAI for the DE-RuS site was associated with a slightly higher simulated grain yield for DE-RuS compared to BE-Lon. However, this relationship is not reflected in the observations, as the measured grain yield is lower for DE-RuS compared to BE-Lon, although the observed LAI is higher for DE-RuS (Figure 3, Table 3).

In CLM, there are several variables that influence the simulated crop yield, such as LAI cycle and peak, length of the leaf emergence phase, harvest date, and water availability from the soil. Except for soil moisture, these

variables are strongly correlated to the GDD scheme which suggests that the simulated crop yield profoundly depends on the GDD. The high sensitivity of simulated yield in CLM towards GDD is not reflected in actual field observation, where crop yield depends on a multitude of factor, environmental conditions (weather, nutrient availability, atmospheric CO2) and management decisions. Underestimation of winter wheat yield at BE-Lon may be due to model deficiencies in representing the complex crop management practices, such as timing and type of fertilizer, ploughing crop varieties and the usage of different winter wheat varieties that can show different responses to water or heat stress, frost and have different grain productivities (White and Wilson, 2006; Bergkamp et al., 2018; Ceglar et al., 2019). In order to include different varieties of any crop, the list of CTFs could be extended with suitable plant parameterizations. However, this information is not readily available, due to combination of measurement data scarcity and the complexity of the phenology algorithm and parameter scheme. The introduction of a phenology scheme based on plant physiological trait information in CLM could be a major improvement in this field (see Fisher et al., 2019), as plant trait information becomes more readily available (e.g. TRY Plant Trait Database, Kattge et al., 2011). Whether considering different varieties and cultivars of a crop is important for regional or global scale simulations remains to be evaluated. In general, as already noted by Lu et al. (2017), a more process based vernalization and cold tolerance routine would be useful to make this subroutine more applicable to other winter crops like rapeseed.

The early leaf onset and harvest for winter wheat simulated by CLM (both with the new routine and parameter set and the control run) could be met by adjusting the minimum date for planting within the CFT parameterization. This could be useful to easily improve the crop cycle representation in regional simulations, where planting patterns are similar for larger agricultural areas. However it would restrict the flexibility of the model to prognostically simulate planting dates.

In general, the simulated plant growth and resulting yield were highly sensitive to plant parameters that govern the growing degree calculation which in turn influence the phenological development and allocation of C and N. With only a limited number of CFTs in CLM, a discretization of plant parameters or varieties on a regional scale is not possible at this point. A potential solution, without introducing additional CFT´s, could be to account for key parameters for each CFT varying with climate and soil conditions for large scale simulations (e.g. by gridded parameter sets). Furthermore, there is a need to evaluate and further discretise plant hydraulic properties (at this point one set of hydraulic parameters is applied to all types of crops) (Verhoef and Egea, 2014; Kennedy et al., 2017; Kennedy et al., 2019). Within the crop module of CLM5, the carbon allocation of crops is limited by soil water available to the plant. Thus, both an improved soil hydrology  and an improved representation of plant hydraulics could play a major role in improving the quality of yield prediction by the model (Bassu et al., 2014; Kennedy et al., 2019). These plant hydraulic properties could be estimated by inverse modelling or data assimilation (e.g. by assimilating measurement data like NEE, LAI, soil moisture and/or energy fluxes using an augmented state-vector approach). In addition, data assimilation of e.g. in situ or remotely sensed soil moisture data and/or LAI could play a major role in increasing the accuracy of regional yield predictions (e.g. Guérif and Duke, 2000; Launay and Guerif, 2005; de Wit and van Diepen, 2007; Fang et al., 2008; Vazifedoust et al., 2009; Huang et al., 2015; Jin et al., 2018).

The default CLM5 does not account for the influence of weeds or cover crops and/or its litter on the carbon balance. There is a tool available for CLM5 that enables the simulation of transient land use and land cover changes (LULCC) (Lawrence et al., 2018). It was designed to simulate the effects of changing distributions of natural and crop vegetation, e.g. land use change from forest to agricultural fields and also allows for changes in crop type

between years (Lawrence et al., 2018), but does not account for intra-annual changes of agricultural management on crop vegetated areas that happen in double and triple cropping scenarios. While this tool is useful to study general land use changes by changing the land cover type of individual land units, we found it lacks flexibility in accounting for changes within land units of the same land cover and does not account for all 64 CFTs. Furthermore, this tool changes the CFT of each column on the 1st of January every year according to prescribed values (customized). Thus, when using the CLM5 land-use change tool, for example to simulate the crop rotation from sugar beet in 2017 to winter wheat in 2017/2018 at DE-RuS, winter wheat would not be planted before fall of 2018 (rather than in the same year as sugar beet is harvested) resulting in a long period of fallow field when switching from summer to winter crop (Fig. 8). Here, the implementation of our cover cropping routine enabled a second onset of plant growth within a year (including the switch to another CFT). This resulted in a pronounced improvement in LAI curves and latent heat flux, especially during winter months, by simulating the growth of a cover crop. It also proved to be beneficial in representing realistic agricultural field conditions by allowing crop rotations with higher flexibility than the default model.

This new routine can be used to study cover cropping scenarios in future large-scale simulations. The effect of a cover crop during winter months on all crop land units where cash crops are grown in summer could be tested. This could also be tested for specific cash crops only. In addition, it is possible to simulate cover crop plantations based on harvest date thresholds. A defined maximum harvest date for any specific cash crop could define whether a cover crop such as winter wheat would be planted or not. For example, for all sugar beet land units with harvest dates before a certain threshold (e.g. day 290 of any given year) winter wheat could be planted as a cover crop during winter. If this harvest threshold were not reached and the summer crop is harvested late in the year, no cover crop would be planted. Alternatively, these harvest thresholds could define the type of cover crop, e.g. early harvest - winter wheat, late harvest – simple greening mix, etc. Also, historical land use information could be used to simulate realistic cover cropping and crop rotation scenarios. The succession of different crops from historical data could also be used to model the succession of crops for the future. In order to study large scale effects of cover cropping and common crop rotations, the CLM5 model would greatly benefit from further crop specific parameter sets for cover crops such as mustard, and further important cash crops.

In their approach, Lombardozzi et al. (2018) studied the effects of idealized cover crop scenarios by simulating winter crops in all crop regions throughout North America. They found that the effects of cover crops on winter temperatures is strongly related to plant height and LAI and emphasized the importance of biogeophysical effects and varietal selection when evaluating the climate mitigation potential of cover cropping (Lombardozzi et al., 2018). With our new routine, it is now possible to evaluate the biogeophysical effects of cover crops over longer time scales and in combination with typical cash crop rotations throughout agricultural areas. Also the ecological potential of different cover crop varieties could be evaluated. We anticipate that this modification will allow a more realistic representation of seasonal LAI in ecosystems where cover cropping and crop rotations are common management practices. The application of this routine is also of interest for areas with several cash crop cycles within a year like multiple annual crop cycles in India and China (Biradar and Xiao, 2011; Li et al., 2014; Sharma et al., 2015). We see further development potential for this routine and corresponding data sets to account for typical crop rotations and cover cropping scenarios for regional scale simulations (e.g. EU regulations and goals on the adoption of cover crops for climate change mitigation (Smit et al., 2019)).

## 6    Conclusion

The default CLM5 was extended by adopting the winter wheat representation of Lu et al. (2017), by including crop specific parameterization for winter wheat, sugar beet and potatoes and by the addition of a cover cropping subroutine that allows several growth cycles within one year. The model modifications were tested for the respective crops at four TERENO and ICOS cropland sites in Germany and Belgium, Selhausen (DE-RuS), Merzenhausen (DE-RuM), Klingenberg (DE-Kli) and Lonzée (BE-Lon), for multiple years. The main results drawn from this study are as follows:

- The implementation of the winter wheat subroutines led to a significant simulation improvement in terms of energy fluxes, leaf area index, net ecosystem exchange and crop yield (reduction of underestimation from 80 – 90 % to 18 – 36 % at test site BE-Lon, good match for the test sites DE-RuS and DE-Kli in 2016 and slight overestimation at test site DE-Kli in 2011)

- The model performance was strongly improved with the  crop specific parameter sets for sugar beet and potatoes: seasonal variations and magnitudes of energy fluxes and LAI were better reproduced with RMSE reduction during the crop cycle by up to 57 % for latent and 59 % for sensible heat flux at test site DE-RuS.

- In most cases the modification of CLM5 led to better reproduction of measured NEE at the test sites. However, the model showed a general weakness in reasonably simulating the NEE on agricultural fields, especially the peak value and post-harvest conditions.

- The implementation of our cover cropping routine enabled a second onset of plant growth within a year and thus was able to better capture realistic field conditions after harvest. Winter time RMSE for latent heat flux was reduced by 42 %. Also, a higher flexibility in terms of crop rotations is now possible with CLM5.

We anticipate that our implementation of the winter wheat representation and specified parameterization will markedly improve yield predictions at regional scale for regions with a high abundance of winter cereal varieties. The cover cropping routine offers an improved basis on which to study the effects of large scale cover cropping on energy fluxes, soil water storage, soil carbon and nitrogen pools, as well as to investigate the role of different cover crops as natural fertilizer in future studies with CLM5. A more realistic representation of post-harvest field conditions can play a crucial part in better representing the role of agriculture on regional and global energy and carbon fluxes and will be further developed and tested for regional scale simulations in future studies.

Despite our improvements, there is still a need to further develop certain functionalities and specific routines regarding the crop representation and land management in CLM5 in order to achieve better model performance for agricultural land. The applicability of the routines to large scale simulations would strongly benefit from additional crop specific parameterizations for important cash and cover crops. Also a better representation of ploughing and tillage needs be included in future model versions in order to better account for the effects of cover crops on the terrestrial carbon cycle and their biogeochemical benefits.

Further general examples for improvements include: (1) an improved representation of plant and soil hydrology that may be highly beneficial for yield predictions, (2) a more detailed representation of agricultural management practices (e.g. tillage, C/N turnover, post-harvest surface conditions, fertilizer types and applications), (3) tools to account for spatial variability in plant physiological parameters, and (4) the discretization of plant hydraulic properties as opposed to using one parametrization for all crops.

## 7    Appendix

Table A1: Sowing and harvest dates at the ICOS and TERENO cropland study sites

| Site code | Site | Years | Crop | Sowing | Harvest/plowing |
|---|---|---|---|---|---|
| DE-RuS | Selhausen | 2015-2016 | Winter barley | 29.09.2015 | 10.07.2016 |
| | | 2016 | Greening mix cover crop | 22.08.2016 | 06.01.2017 |
| | | 2017 | Sugar beet | 31.03.2017 | 05.10.2017 |
| | | 2017-2018 | Winter wheat | 25.10.2017 | 16.07.2018 |
| | | 2019 | Potato | 26.04.2019 | 03.10.2019 |
| DE-RuM | Merzenhausen | 2016 | Potato | 12.04.2016 | 24.08.2016 |
| | | 2016-2017 | Winter wheat | 17.10.2016 | 22.07.2017 |
| | | 2017-2018 | Rapeseed | 30.08.2017 | 16.07.2018 |
| DE-Kli | Klingenberg | 2003-2004 | Winter barley | 06.09.2003 | 31.07.2004 |
| | | 2004-2005 | Rapeseed | 18.08.2004 | 02.08.2005 |
| | | 2005-2006 | Winter wheat | 25.09.2005 | 06.09.2006 |
| | | 2007 | Corn | 23.04.2007 | 02.10.2007 |
| | | 2008-2009 | Winter barley | 25.04.2008 | 27.08.2008 |
| | | | | 12.09.2008 | 22.07.2009 |
| | | 2009-2010 | Rapeseed | 25.08.2009 | 24.08.2010 |
| | | 2010-2011 | Winter wheat | 02.10.2010 | 22.08.2011 |
| | | 2012 | Corn | 25.04.2012 | 18.09.2012 |
| | | 2013-2014 | Winter barley | 17.04.2013 | 24.08.2013 |
| | | | | 01.10.2013 | 20.07.2014 |
| | | 2014-2015 | Rapeseed | 21.08.2014 | 08.08.2015 |
| | | 2015-2016 | Winter wheat | 18.09.2015 | 24.08.2016 |
| | | 2016-2017 | Radish and Brassica cover crop | 01.09.2016 | 15.03.2017 |
| | | 2017-2018 | Winter barley | 02.04.2017 | 25.08.2017 |
| | | 2016-2017 | Radish and Brassica cover crop | 13.09.2017 | 13.04.2018 |
| | | 2018 | Corn | 02.05.2018 | 04.09.2018 |
| | | 2019 | Bean | 23.03.2019 | 18.08.2019 |
| BE-Lon | Lonzée | 2006-2007 | Winter wheat | 13.10.2006 | 05.08.2007 |
| | | 2008 | Sugar beet | 22.04.2008 | 04.11.2008 |
| | | 2008-2009 | Winter wheat | 13.11.2008 | 07.08.2009 |
| | | 2009 | Mustard | 01.09.2009 | 01.12.2009 |
| | | 2010 | Potato | 25.04.2010 | 05.09.2010 |
| | | 2010-2011 | Winter wheat | 14.10.2010 | 16.08.2011 |
| | | 2012 | Corn | 14.05.2012 | 13.10.2012 |
| | | 2012-2013 | Winter wheat | 25.10.2012 | 12.08.2013 |
| | | 2013 | Mustard | 05.09.2013 | 15.11.2013 |
| | | 2014 | Potato | 07.04.2014 | 22.08.2014 |
| | | 2014-2015 | Winter wheat | 15.10.2014 | 02.08.2015 |
| | | 2015 | Mustard | 26.08.2015 | 09.12.2015 |
| | | 2016 | Sugar beet | 12.04.2016 | 27.10.2016 |
| | | 2016-2017 | Winter wheat | 29.10.2016 | 30.07.2017 |
| | | 2017 | Mustard | 07.09.2017 | 08.12.2017 |
| | | 2018 | Potato | 23.04.2018 | 11.09.2018 |
| | | 2018-2019 | Winter wheat | 10.10.2018 | 01.08.2019 |

Table A2: Default (*control*) and new crop specific (*new*) phenology and CN allocation parameters for the CFTs sugar beet and potatoes (control parameters are those for the CFT spring wheat) and winter wheat.

| CFT | | Sugar beet | | Potatoes | | Winter wheat | |
|---|---|---|---|---|---|---|---|
| **Parameter set** | | *control* | *new* | *control* | *new* | *control* | *new* |
| **Variable** | **Units** | *Phenology* | | | | | |
| **min_NH_planting_date** | MMDD | 401 | 401 | 401 | 401 | 901 | 901 |
| **max_NH_planting_date** | MMDD | 615 | 530 | 615 | 530 | 1130 | 1130 |
| **planting_temp** | K | 280.15 | 280.15 | 280.15 | 277.15 | 1000 | 1000 |
| **min_planting_temp** | K | 272.15 | 272.15 | 272.15 | 272.15 | 283.15 | 283.15 |
| **gddmin** | °days | 50 | 60 | 50 | 60 | 50 | 100 |
| **mxmat** | days | 150 | 180 | 150 | 180 | 330 | 400 |
| **baset** | °days | 0 | 0 | 0 | 0 | 0 | 0 |
| **mxtmp** | °C | 26 | 30 | 26 | 30 | 26 | 26 |
| **hybgdd** | - | 1700 | 2000 | 1700 | 2000 | 1700 | 2000 |
| **lfemerg** | % | 0.05 | 0.05 | 0.05 | 0.05 | 0.03 | 0.03 |
| **grnfill** | % | 0.6 | 0.65 | 0.6 | 0.65 | 0.4 | 0.6 |
| **ztopmx** | m | 1.2 | 0.5 | 1.2 | 0.5 | 1.2 | 1.2 |
| **laimx** | $m^2/m^2$ | 7 | 6 | 7 | 6 | 7 | 7 |
| **slatop** | $m^2/gC$ | 0.035 | 0.02 | 0.035 | 0.02 | 0.035 | 0.028 |
| **Variable** | **Units** | *CN ratios and allocation* | | | | | |
| **leafcn** | gC/gN | 20 | 11 | 20 | 11 | 20 | 20 |
| **leafcn_min** | gC/gN | 15 | 8 | 15 | 8 | 15 | 15 |
| **leafcn_max** | gC/gN | 35 | 20 | 35 | 20 | 35 | 35 |
| **frootcn** | gC/gN | 42 | 42 | 42 | 42 | 42 | 43 |
| **graincn** | gC/gN | 50 | 50 | 50 | 50 | 50 | 15 |
| **flnr** | $fraction/gNm^{-2}$ | 0.41 | 0.15 | 0.41 | 0.15 | 0.41 | 0.3 |

Table A3: Textural fractions (sand, silt and clay percentages) for the ICOS and TERENO cropland study sites averaged for the upper soil layers (up to 50 cm) with corresponding reference.

| Site/ID | Sand [%] | Silt [%] | Clay [%] | Ref. |
|---|---|---|---|---|
| Selhausen/DE-RuS | 16.4 | 63.4 | 14.9 | Brogi et al. (2019) |
| Merzenhausen/DE-RuM | 16.4* | 63.4* | 14.9* | - |
| Klingenberg/DE-Kli | 21.5 | 22.8 | 55.7 | Grünwald (personal communication, 2020) |
| Lonzée/BE-Lon | 5-10 | 68-77 | 18-22 | Moureaux et al. (2006) |

*adopted from the DE-RuS site

## 7.1 Winter cereal representation (extended)

The temperature at the crown of the plant ($T_{crown}$) is assumed to be slightly higher than the 2-m air temperature ($T_{2m}$) in winter when covered by snow, and the same as the 2-m air temperature without snow cover. Within CLM5, it is calculated separately for temperatures below and above the freezing temperature ($T_{frz}$):

$$T_{crown} = 2 + (T_{2m} - T_{frz}) * (0.4 + 0.0018 * (\min(D_{snow} * 100, 15) - 15)^2$$

for $T_{2m} < T_{frz}$   (A1)

$$T_{crown} = T_{2m} - T_{frz}$$

for $T_{2m} > T_{frz}$   (A2)

where $T_{crown}$ [K] is the calculated crown temperature, $T_{2m}$ [K] is the 2-m air temperature, $T_{frz}$ [K] is the freezing point and $D_{snow}$ [m] is the snow height.

The temperature at which 50 % of the plant is damaged ($LT_{50}$) is calculated interactively at each time step ($LT_{50t}$) depending on the previous time step ($LT_{50t-1}$) and on several accumulative parameters. These parameters are the exposure to near-lethal temperatures ($rate_s$), the stress due to respiration under snow ($rate_r$), the cold hardening or low temperature acclimation (contribution of hardening – $rate_h$) and the loss of hardening due to the exposure to a period of higher temperatures (dehardening – $rate_d$) that are each functions of the crown temperature (Lu et al., 2017 and references therein):

$$LT_{50t} = LT_{50t-1} - rate_h + rate_d + rate_s + rate_r \tag{A3}$$

The exposure to near-lethal temperatures is based on the winter survival model after (Fowler et al., 1999) and is calculated as follows:

$$rate_s = \frac{LT_{50t-1} - T_{crown}}{e^{-1.9(LT_{50t-1} - T_{crown}) - 3.74}} \tag{A4}$$

The stress due to respiration under snow is calculated as a function of snow depth (dsnow) that ranges from 0 to 1 for snow cover up to 12.5 cm (equal to 1 for all snow depth higher than 12.5), and a specific respiration factor (RE):

$$rate_r = R \times RE \times f(dsnow)$$

$$R = 0.54 \quad f(dsnow) = min(dsnow, 12.5)/12.5$$

$$RE = \frac{e^{0.84 + 0.051\,T_{crown} - 2}}{1.85} \tag{A5}$$

The contribution of hardening and dehardening are calculated within certain temperature ranges as follows:

For $T_{crown} < 10°C$

$$rate_h = 0.0093(10 - max(T_{crown}, 0))(LT_{50t-1} - LT_{50c}) \tag{A6}$$

For $T_{crown} \geq 10°C$ when vf < 1 (not fully vernalized), and $T_{crown} \geq -4°C$ when vf =1 (fully vernalized)

$$rate_d = 2.7 \times 10^{-5}(LT_{50i} - LT_{50t-1})(T_{crown} + 4)^3 \tag{A7}$$

where $LT_{50c}$ is the maximum frost tolerance of -23 °C and $LT_{50i}$ represents the $LT_{50}$ for an unacclimated plant ($LT_{50i} = -0.6 + 0.142\,LT_{50c}$).

The survival rate ($f_{surv}$) is then calculated as a function of $LT_{50}$ and the crown temperature. The probability of survival is a function of $T_{crown}$ in time (t). It increases once $T_{crown}$ is higher than $LT_{50}$ or decreases when it is lower (Vico et al., 2014):

$$f_{surv}(T_{crown}, t) = 2^{-\frac{T_{crown}}{LT_{50}}^{\alpha_{surv}}} \tag{A8}$$

where $\alpha_{surv}$ is a shape parameter of 4.

The winter killing degree day (WDD) is calculated as a function of crown temperature and survival probability, where the maximum function limits the integration to the potentially damaging periods, when the air temperature (T) is lower than the base temperature ($T_{base}$) of 0°C (Vico et al., 2014):

$$WDD = \int_{winter} max[(T_{base} - T_{crown}), 0] [1 - f_{surv}(T_{crown}, t)]dt \tag{A9}$$

Lower $LT_{50}$ indicate a higher frost tolerance and would result in higher survival rates, smaller WDD and less cold damage to the plant. Thus, when the survival probability and crown temperature are low, the WDD will be high (Vico et al., 2014).

The survival probability and the WDD are then used to estimate instant and accumulated frost damage to the crop during the leaf emergence phase (Lu et al., 2017). Instant frost damage is assumed to happen at the beginning of the growing season when the plants are not fully vernalized (vf < 0.9) when the growth of leaves (especially new

leaves or small seedlings) due to an exposure to low temperatures. It is simulated by reducing the leaf carbon at low survival probabilities (whenever $f_{surv}$ is below 1). The leaf carbon is reduced by an amount of 5 gC m$^{-2}$ scaled by a factor of 1- $f_{surv}$ that is moved to the carbon litter pool, up to a minimum value of 10 gC m$^{-2}$ leaf carbon:

$$\text{leafc}_t = \text{leafc}_{t-1} - \text{leafc}_{\text{damage}}(1 - f_{\text{surv}})$$

for vf < 0.9, WDD > 0, $f_{surv}$ < 1, and leafc$_t$ > 10 (A10)

where leafc$_t$ is the simulated leaf carbon of the current time step, leafc$_{t-1}$ is the leaf carbon of the previous step and leafc$_{damage}$ is equivalent to 5 gC m$^{-2}$.

When the plant is close to vernalization towards the end of the leaf emergence phase, it is not as susceptible to suffer from instantaneous frost damage as in the beginning of this phase. Still, an extended period of freezing temperatures can potentially induce damage to the plant (Lu et al., 2017). This accumulated frost damage is simulated based on the accumulated WDD and average survival probability. When the accumulated WDD reaches a value higher than 1° days, the leaf carbon from the previous time step (leafc$_{t-1}$), scaled by the average $f_{surv}$, is moved to the soil carbon litter pool:

$$\text{leafc}_t = \text{leafc}_{t-1}(1 - \text{average } f_{\text{surv}})$$

for vf $\geq$ 0.9 and WDD > 1 (A11)

Once this has occurred, the accumulated WDD is reset to 0 and the tracking of the average $f_{surv}$ is restated. Corresponding to the leaf carbon reduction, the leaf nitrogen is reduced from the leaf nitrogen pool to the soil nitrogen litter pool scaled with the parameterized leaf C/N ratio for winter wheat of 20.

*Code availability.* The modified model version CLM_WW_CC is freely available via Zenodo, doi:10.5281/zenodo.3978092.

*Data availability.* For the TERENO sites Selhausen (TERENO ID: SE_EC_001 and SE_BK_001) and Merzenhausen (TERENO ID: ME_EC_001, ME_BCK_001), all EC and meteorological data are freely available via the TERENO data portal TEODOOR (http://teodoor.icg.kfa-juelich.de/): Selhausen – ID SE_EC_001 doi:20.500.11952/TERENO/00000004; Selhausen – ID SE_BDK_001 doi:20.500.11952/TERENO/00000068; Merzenhausen – ID ME_EC_001 doi:20.500.11952/TERENO/00000434; Merzenhausen – ID ME_BCK_001 doi:20.500.11952/TERENO/00000166. EC data for the ICOS study sites Lonzée (ICOS ID: BE-Lon) and Selhausen (ICOS ID: DE-RuS) is available via the ICOS data portal (https://www.icos-cp.eu/). Additional data on vegetation and management practices (e.g. LAI, NDVI, canopy heights etc.) were kindly provided by the respective site operators.

*Competing interests.* The authors declare that they have no conflict of interest.

*Author contribution.*T. B. developed the modified model code, designed, performed and analysed the simulation experiments and prepared the manuscript with contributions from all co-authors. H.B., H.J.H.F., D.R., A.W. and H.V. supervised the research, co-designed the experiments and contributed to the manuscript. M. S., B. G. and B. H. performed pre-processing (e.g. quality control, gap-filling) of the respective site data.

*Acknowledgements.* The authors thankfully acknowledge Yaqiong Lu for providing the source code to the winter cereal implementation to CLM4.5 (Lu et al., 2017). Furthermore, the authors gratefully acknowledge the

computing time granted on the supercomputer JURECA by the Jülich Supercomputing Centre (JSC). This study was partly funded by the Jülich-University of Melbourne Postgraduate Academy (JUMPA), an international research collaboration between the University of Melbourne, Australia, and the Research Centre Jülich, Germany. This work used data acquired and shared by the TERENO project of the Helmholtz Association, the EU ICOS project of the European Research Infrastructure programme (ESFRI) and by the Deutsche Forschungsgemeinschaft (DFG, German Research Foundation) under Germany's Excellence Strategy - EXC 2070–390732324, project PhenoRob.

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
