# Peer review of "Improving the representation of cropland sites in the Community Land Model (CLM) version 5.0"

_Geoscientific Model Development, 2020_

## Referee Comment (RC1) · Anonymous Referee #1 · 8 Sep 2020

Boas et al. used ground data on three crops (two cash crops (sugar beet and potatoes) and winter wheat) to improve their representation in CLM5. Overall, I agree that the use of data is an important step for improving the predictions of crops in CLM5 and I like the idea of representing processes in the model that occur in real life in field. The main contribution from the authors end are: (1) implementation of the winter wheat subroutine (2) parameterizing for all three crops and (3) implementation of a method that allow rotation of crops within a year. After all these three changes, the authors compared modeled outputs of carbon cycling (leaf area index and net ecosystem exchange) and energy cycling (latent and sensible heat) processes of the above three crops with the measurements, and argue that their model perform better than the model (standard

CLM5) that used default processes/parameters.

My main critique is that it is difficult to understand the impacts of (1) and (3) in particular. I tried to look into the code, but couldn't quite locate (1) and (3). I suggest the authors to make it clear in the code where these implementations are (perhaps mark it) so that I can follow how much the codes were changed relative to the standard. To gauge the impacts of (1), I would like to see a winter wheat simulation only at the "DE-RuS" site for with and without (1). You could show how much leaf area index, latent heat and sensible heat of winter wheat changes with this assumption. Similarly, to examine the impact of (3), you could do a simulation of sugar beet and Winter wheat at "DE-RuS" (in this case sugar beet will be rotated, which you already did) and a simulation of Winter wheat only at "DE-RuS". You could also show how much leaf area index, latent heat and sensible heat of winter wheat changes if there was no rotation. Additionally, you can check whether rotation has any impacts on the modeled nitrogen leaching and fixation rates.

I list my specific comments as follows:

(1) While I appreciate some of the details in sections 2.1, and 2.2.1, it would be appropriate it put most of the text in the Appendix section. For example, the paragraph that starts with the description of the default crop phenology scheme (lines 139 to 152 and additional lines) is not new to this study but rather standard CLM5 documentation notes and therefore, they can be put in the Appendix. Similarly, the section about Winter cereal representation that begins with "Vernalization" is also not new to this study. The default phenology scheme of CLM5 has a Vernalization subroutine.

(2) The authors emphasize the importance of cash crops (e.g. sugar beets and potatoes). I would like the authors to comment on the spatial coverage of these crops in Germany and whether the famers are smallholder or largescale holder plantation owners. Along similar lines, it would be good if the authors could comment on how they plan to carry out the large scale simulations or regional simulations for these crops

given that you need time series information about the rotation of these crops and also that some crops might be planted every two years or so.

(3) A number of statements in the results section is difficult to follow. For example, in lines 407 to 412, there is no reference to any figures. What is green leaf area index in line 408? Do you mean before maturity, during maturity or after maturity?

(4) I think the poor seasonal dynamics and low magnitude of the leaf area index in Figures 2-5 of CLM-D could also be related to the parameter values rather than the winter wheat subroutine that was introduced in this study. There are at least 3 parameter values that are considerably different compared to the default parameters of CLM ('gddmin','hybggg' and 'graincn'). For example, I see that the default gddmin is 50 in the default but 100 in the modified case (this study). Also hybgdd in the modified case is 30 more than the default. So couldn't these likely explain poor seasonal dynamics and low magnitude of the leaf area index in Figures 2-5 of CLM-D?

Some of the minor comments are as follows:

(1) In line 70, Bilinois et al. (2015) is cited but I think the reference is missing.

(2) Please provide fractions of sand, silt and clay in Table 2, maybe up to 5 cm or 10 cm?

(3) While I agree with the statement (line 289) that "CLM5 only permits land use changes at the beginning of every year", users can start a CLM5 simulation in any month the land use change actually happens in real life by performing a 'clear-cut' following spin-up, for example.

(4) At the "BE-Lon" site, the LAI curve of winter wheat from DOY= 0 to DOY = 100 seems to have a relatively gradual and smooth growth (Figure 2) while at sites "DE-RuS", "DE-RuM", "DE-Kli", the growth is relatively sudden and steep during the same period. I would like the authors to provide some explanations for this difference.

(5) In lines 602 to 603, the authors claim that CLM5 does not represent timing of

fertilizer. Please provide a citation for this?

(6) In line 603, the authors state that CLM5 does not consider varieties of winter wheat. I agree with this statement but at the same time, many land surface models don't consider varieties or cultivars of crops. Crops can be genetically modified to boost productivity. This means there could large differences in the parameter estimates among varieties/cultivars. The authors could discuss the variation in the parameter estimates if they are measured at their sites.

(7) The authors mention in lines 626 to 629 the following: "There is a tool available for CLM5 that enables the simulation of transient land use and land cover changes (LULCC) (Lawrence et al., 2018). It was designed to simulate and study the effects of changing distributions of natural and crop vegetation, e.g. land use change from forest to agricultural fields (Lawrence et al., 2018), rather than inter-annual changes of agricultural management on crop vegetated areas." I'm confused about the last part "rather than inter-annual changes of agricultural management on crop vegetated areas". Please explain what do you mean by this? Do you mean you cannot change the Nitrogen fertilization rate from year to year in CLM5?

———————————————————

---

## Referee Comment (RC2) · Anonymous Referee #2 · 17 Sep 2020

Overall: Boas et al. have done considerable work to modify and evaluate simulations of European agriculture in CLM. This work is very exciting and shows significant improvement in model parameterizations and capabilities to add cover crop and crop rotation management practices.

Despite the importance of this work, the text needs to be revised before it is suitable for publication. Some sections require reorganization for clarity, while others will benefit from streamlining to remove redundancies or adding necessary detail. Comments highlighting these sections are included below, as well as specific line comments.

Abstract. Line 25: Is cover cropping only common in humid and sub-humid regions?

[Figure]

Perhaps it would be more informative to rephrase to something similar to: "...which is an agricultural management technique commonly used in the regions evaluated in this study." Alternatively, you can say that it is a technique growing in popularity to improve soil health and carbon storage.

Line 26: Are you referring to the parameterization of new CFTs? Please clarify. Line 27: Please move the reference of RSME for LH and SH to just after the energy fluxes rather than after NEE. Line 31: When you refer to the "LAI curve", is this the same as the season cycle of LAI? If so, please modify the wording to reflect this. Lines 31-33: It would be more impactful if you strengthened the last sentence in the abstract. Here is one suggestion: "Our modifications significantly improve model simulations and should therefore be used in future simulations to better understand large-scale impacts of agricultural management on carbon, water, and energy fluxes."

Introduction: Overall, the introduction needs some reorganization. You need to more clearly highlight the role of management (make this a separate paragraph, include cover crops but also other types of management). The new representation of cover crops is a primary contribution to this paper and is barely mentioned here. The introduction also needs a broader overview of crops in LSMs (it currently only focuses on AgroIBIS and CLM). Last, most of the introduction emphasizes the global nature of models and that the variation in soils, plants, climate is important. When the reader finally gets to the end of the introduction, which highlights that you focus on a few sites in Europe (which some may argue has narrower variation in soils, plants, climate than if you were to compare to locations from other continents), it makes this study seem limited. It might help to instead describe that models are still limited by their ability to represent many crop types and important management practices, emphasizing the importance of your work adding these new capabilities, and also to highlight that Europe is a major agricultural hub for global food production.

Lines 44-49: The mention of cover crops here seems a bit out of place. The earlier part of this paragraph and the start of the next paragraph is focused on adaptation to

climate change, whereas the description of cover crops here focuses on soil benefits and climate mitigation. I suggest reorganizing, moving the cover crop description to later in the introduction.

Lines 68-70: I'm not sure I entirely understand the point of this sentence. Is this just to highlight the evaluation of crops in CLM4.5?

Lines 73-76: You should also reference the CLM5 crop overview paper here, which evaluates global crop yields: Lombardozzi, D. L., Y. Lu, P. J. Lawrence, D. M. Lawrence, S. Swenson, K. W. Oleson, W. R. Wieder, and E. A. Ainsworth (2020), Simulating Agriculture in the Community Land Model Version 5, J Geophys Res-Biogeo, 125(8), 927–19, doi:10.1029/2019JG005529.

Lines 77-84: This paragraph seems too detailed for the introduction. I suggest summarizing and merging with the previous paragraph. For example: "The few studies that have evaluated CLM5 suggest inaccurate phenology and overestimated crop yields (Sheng et al. 2018)." However, you'll probably want to change/update this to also incorporate results from the Lombardozzi et al. CLM5 paper mentioned above.

Methods Overall: The methods section needs to be tidied up. There are redundancies in the first section, and a lack of detail in the cover crop description. Please pay careful attention to providing enough detail that the reader isn't left wondering how something was done, but keep the text succinct.

You reference Lawrence et al. 2018 in several places throughout the text. However, I believe this paper was published in 2019 (not 2018). Please double-check.

Section 2.1:

When describing the crop model, please also cite Lombardozzi et al. (2020), as this has much more detailed information about the crop model updates than Lawrence et al. 2019.

The methods should be streamlined to avoid repetition. For example, allocation is

mentioned in lines 134-138, and then again in the paragraph starting at line 153.

When referring to C allocation, you state that it varies throughout the growing season (e.g., line 156), whereas the reference to N allocation states that it uses two different C/N ratios (lines 161-162). However, these are treated the same way in the model. Please update for consistency.

I suggest switching the order of Eq. 1 and Eq. 2.

Line 114: Please define "CFT" the first time you use this term.

Line 115: land units are not separated by fertilizer, only by irrigation. Please update.

Lines 204-206: This is a bit confusing and could use clarification. Does the vernalization factor always range from 0-1? Is it applied to GDD for air and soil temperatures (e.g., does it affect all phenological phases)? If it is only applied to grain C allocation, where does the remaining C get allocated?

Section 2.2.1

It would be helpful to start with an overview of how winter cereal representation differs from other crops. I suggest a high level overview of why it's important to include both vernalization and cold tolerance before diving into the details of each.

Equation 4: You specify that Tcrown is slightly higher than the freezing temperature when covered by snow. I see that snow height is used in the calculation, but where is the plant height? Without including the plant height, how do you know whether the plant is covered by snow?

Line 213: The text describes what the accumulative parameters are, but what about the previous time step is used? It would also be useful to include a brief description of how some of the accumulative parameters accumulate (e.g., are these all based on some aspect of accumulated temperature?)

Equation 6: Please define the "alpha-surv" and the "t" variables in this equation.

Equation 7: I am confused by this, partly because it's not clear what the equation is taking the max of. Also, can Tcrown be negative? That seems to be the only way the solution to this equation isn't 0. Please update to clarify. Also, I think 'fsurf' should in fact be 'fsurv'.

Paragraph starting at line 227: I find the description here a little confusing. Can you revise this to more clearly articulate the difference between survival probability and WDD? Is survival probability just a step function, where any value <1 causes the same amount of damage (simulated as part of the C and N pools being transferred to litter)? Should I be thinking of survival probability as the proportion of the plant that survives, or the probability that the whole plant survives? Also, part of my confusion is that this is the first place that a frost damage function is mentioned.

Table 1: This is a useful summary, but I'm not sure it adds much information to the main text.

Section 2.2.2:

Since you use a pre-existing winter wheat parameterization, it would be helpful to include some information about what you changed in the parameterization and why.

Section 2.2.3: How do you determine when the cover crops (or rotations) are planted and the subsequent phenology phases? Is it based on GDD? Did you have to modify GDD parameters or add new ones? Did you add new CFTs to accomplish this? How is allocation determined? This section needs more detail about how modifications were made, as it is the bulk of the development work in this paper.

Lines 267-270: It's great to hear that you introduced a flag to use the cover crop option, but I'm not sure you need to include that description here.

Lines 276-277: How are you predefining a rotation scheme?

Line 283: "catch crop" — this is the first time you mention it. Are you using this interchangeably with cover crop (which is how you described this in the previous sentence),

or are you using a new phrase to distinguish this from cover crop? Please be clear and consistent with word choices.

Line 283: You mention plowing the crop into the soil. However, CLM does not represent plowing. How did you accomplish this. Do you assume that the plant biomass is transferred to the litter pool? Also, how did you decide when this happens?

Section 2.3:

I think it would help to describe the sites before the validation data, and/or mention whether you run CLM simulations at these sites. This section starts by describing validation data, but does not mention what is being validated.

Table 2: Useful information about the sites, but I think the map describes the locations quite well, and most of the other information included in the table is not used in the simulation. Therefore, I'm not sure that this table is necessary in the main text.

Lines 318-319: You mention winter wheat twice here.

Lines 341-342: CLM's default time step is 30 minutes.

Section 3.1: Throughout this section, the differences in model version versus parameter set seem to be conflated. Please make this much clearer throughout, explaining what each of the model versions includes and what the default versus modified parameter sets include.

Table 3: Which simulations include the potato and sugar beet parameterization? It looks like it's the CLM_WW simulation, but this needs to be explicitly mentioned in the table description.

Lines 364-366: This text is confusing: It is not clear what the difference is between the default model and the modified model. I assumed the "default" model did not include winter wheat, but this text suggests that it does. How, then, is the default model run with the modified winter wheat parameters different from the winter wheat model with

the modified parameters?

Lines 369-370: What are the default parameterizations of sugar been and potatoes? These aren't included in CLM, so is there a "default"?

Section 4

In general, I find the use of CLM_D, CLM_WW, and CLM_WW_CC to be confusing, as the changes included in each are not clearly described. Additionally, it seems that sugar beet and potato parameterizations are added to CLM_WW. It might be more helpful to instead refer to CLM_D as "control" or "default" and then refer to updated parameterization (e.g., "improvements to winter wheat" rather than "CLM_WW" in Section 4.1 and "new potato" or "new sugar beet" parameterization in Section 4.2).

Additionally, throughout this section, figures should include estimates of uncertainty.

Section 4.1: Throughout this section, the text could be streamlined to avoid repeating the description of trends for each site (see note below about Figs. 2-5). Additionally, the trends in energy fluxes are barely mentioned, leaving the reader wondering why you show these in Fig. 2-5, particularly since their mention focuses on cumulated monthly sums (which aren't shown). Also, yields are discussed frequently throughout the text in this section. Is it worth making a bar chart of yields to more clearly illustrate their evaluation? I realize that a bar chart may look busy, but perhaps averaging across years for the sites with multiple years and including standard deviations will work. Related, how are you calculating yields from CLM simulations? It's important to use the peak daily grain carbon value for the entire growth cycle rather than averaging this over some period of time.

I suggest reorganizing the text (and figures) have 4 paragraphs, focusing on the descriptions of: 1) LAI ; 2) yields; 3) NEE; and 4) energy fluxes. Highlight differences among sites within each paragraph. You can also include an opening paragraph that mentions that CLM_WW improves trends for nearly all variables compared to CLM_D,

so the remainder of the discussion focuses on the evaluation of CLM_WW.

Figures 2-5: —Is it possible to compile these into a single, multi-panel figure? Given that they all show the same variables for different sites, a single panel would allow the reader to compare across sites more easily. Another, possibly better, alternative is to combine all sites and separate the figures into LAI (Fig. 2) and energy fluxes (LH, SH in Fig. 3). It would also allow you to streamline the description of trends throughout Section 4.1. —If I understand the legends correctly, simulations and observations in Figs. 2 and 5 are averaged over multiple years. Can you add uncertainty estimates to these plots? If you plot all individual years (it looks like you possibly do that for observations, but not model), it might be easier to plot averages across years and then plot the uncertainty range associated with interannual variability. — Fig. 2 states that the observations are GLAI, whereas Figs. 3-5 state that the observations are LAI. Are the observations LAI, GLAI, or does this vary by location? If it is different by location and both LAI and GLAI are used, how might this change the ability to evaluate CLM? — Fig. 5: There aren't any LAI observations plotted in panel a, yet the figure legend suggests that there should be site observation data for LAI.

Lines 394-5: As you state, it looks like the LAI peak is indeed too early. However, even more noticeable (and not mentioned) is the fact that the LAI peak looks to be dramatically underestimated.

Lines 413-4: Table 4 suggests that crops are only harvested $\sim$ 1 month too early, but there are higher observed LH fluxes later in the season than just one month. Is this due to cover cropping, which is not included here?

Line 420: I think the phrasing "overestimated early growing season LAI" is potentially misleading. While it is technically correct, the simulated peak LAI values are actually similar to observed peak LAI values, but happen earlier in the year. I think it might be more informative to state that the peak magnitudes are similar, but that the peaks happen too early in the year.

Lines 422-3: What does "growing cycle" refer to hear? As you mentioned earlier, LAI peaks too early and planting and harvest start early, suggesting that phenology is not accurate. Therefore, it is unclear what you mean by "generally good correspondence in growing cycle and LAI".

Lines 437-8: How can you say that CLM_WW resulted in more realistic magnitudes when you stated in the previous sentence that observations aren't available?

Lines 438-9: This is confusing. Does it refer to only the simulations, or also reference the observations? I get the sense that you are conflating simulated peak LAI with simulated and observed crop yields. It implies that lower LAI causes the lower crop yields, although I don't think you can say that for sure.

Line 440: I think this may be backwards. Table 4 suggests that yields are overestimated in 2011 and match really well for 2016.

Lines 453-4: Are all the subsequent mentions (including the metrics in Table 5) calculated using the cumulative monthly sums?

Line 459: You just stated that the BE-Lon sites high some of the highest correlations in the previous sentence, and here single out this site as having high RMSE and biases with low correlations.

Lines 460-461: This sentence should be moved to above, where you briefly mention the mismatch in late-season LH. Also, how does this affect the metrics in Table 5 (see above comments as well).

Line 464: Are you referring to CLM_WW? I suggest clarifying here, as you do include simulations that represent cover crops.

Lines 471-2: It is not quite accurate to say that NEE observations match better due to improved LAI. Consider changing to: "in part due to the better representation of LAI".

Line 473: Are you actually using cumulative monthly values? Fig. 6 show NEE in

unites of umol CO2/mˆ2/sˆ1.

Line 475: Both sites? You mentioned three in the previous sentence. If only referring to two sites, please specify which ones.

Section 4.2: Perhaps this should be titled "New Parameterizations" or "Sugar beet and Potato Parameterizations" to distinguish from the modified winter wheat parameterization in Section 4.1

The evaluation of corn here seems a bit out of place since this section focuses primarily on the new parameterizations. I'm not sure where it goes (perhaps in supplemental?), though. Perhaps this section could be refocused as "Evaluation of other crop types", which includes corn and also the new crop types.

Lines 489-91: I suggest rephrasing to add some detail: "The modifications to winter wheat in CLM_WW do not affect other crop types. Therefore, we add new parameterizations for sugar beet and potatoes to this code."

Lines 502-4: Is this parameter set modified, or new? What is it strongly improved compared to, if these didn't exist in CLM? I assume it was compared to the default CLM crop model (where the crop might be represented by another type of crop), and it would help to know for sure.

Line 507: You reference spring wheat here. Is this the crop type that default CLM uses for these sites? If so, you might want to make this clearer (and mention it earlier). For example: "The default parameterization in CLM uses spring wheat for these crop types and effectively reproduced the growth cycle and seasonal LAI, simulations using the potato and sugar beet parameterizations better captured harvest date and growth cycle. "

Line 509: As in previous comments, I don't think "modified" is the best way to describe this. I suggest using "crop-specific parameters" or "parameterizations for new crop types" or similar. As far as I understand, parameters for new crop types were added,

not modified.

Lines 510-2: It looks like the latent heat flux is very similar for the other site, which might be worth mentioning.

Lines 528-30: Performed better for NEE? Please clarify.

Figures 8 & 9: — I suggest updating the use of "default" and "modified" here based on above comments — Please specify that the LAI results are daily (if they indeed are) — In previous figures, NEE is described as "cumulative monthly", but here is described as "monthly averaged". Can these be calculated and referred to in the same way for consistency?

Section 4.3 It seems that this section focuses on crop rotation as much as cover cropping. I suggest updating the heading to "Cover cropping and crop rotation" or similar to reflect this.

Lines 553-4: Is the simulation of a second crop growth onset for the same crop or for the cover crop? The current wording suggests that a second onset is for the same crop within one year AND for the cover crop. If this isn't intended, perhaps change to "simulation of a cover crop as a second crop growth onset within a single year"

Line 556: "Greening mix" — is this the same as cover crop, catch crop? Please be consistent in your terminology throughout.

Lines 556-557: Perhaps it would be more accurate to say "the cash crop rotation of barley (simulated using the spring wheat CFT)".

Line 557: Spring wheat in CLM is not considered a perennial. It can simulate multiple years of spring wheat in a row, but that doesn't make it perennial.

Lines 559-561: Can the effects of planting cover crops and the crop rotation be isolated?

Line 563: Please change "plantation" to "planting"

Line 576: Similar to above, spring wheat is not a perennial crop in CLM, as it's planted every growing season.

Figures 10-11: It looks like these are for the same site and continuous. Why not plot the full time series on the same panel, adding lines or shading to show the transitions and associated crop type labels. Also, do you not have observational data for LH for 2017-2019 (Fig. 11)?

Section 5

In addition to the benefits and challenges of the new model developments that you include, I was hoping to see further big-picture discussion, for example about how these new developments might improve future large-scale simulations, possible interactions with climate, etc. Consider adding a paragraph to highlight how your improvements can improve our understanding of larger-scale processes. Also, NEE isn't mentioned at all. Why do you think that NEE didn't improve as much as energy fluxes?

Lines 597-8: As mentioned in a previous comment, higher LAI does not mean higher grain yield. There are many factors that affect yield, including photosynthetic rate, nutrient availability, etc. Also, the results presented in this sentence further support that LAI does not directly correspond to yield: grain yield was higher at BE-Lon (which had lower LAI) than DE-Rus.

Line 603: CLM may not represent different varieties, but the parameters could be changed (as you did here) to represent different varieties, especially when simulated at single points.

Line 607: It might be clearer to say "The early leaf onset and harvest for winter wheat simulated by CLM. . ."

Lines 619-22: Can this be more specific? How would discretizing plant hydraulic properties improve yield prediction? Also, why does the reference include "Daniel"? How could the properties (parameters?) be estimated by inverse modeling or data assimi-

lation?

Lines 629-31: Why isn't it applicable to regional simulations? If a simulation is set up to use land use change, the distributions of vegetation, including crop types, will change, even on a point scale, and can be customized by the user if desired.

Line 634: Do you mean before fall of 2018? Fall of 2017 would be the same year.

Line 635: I don't see Figure 12.

Section 6

Line 665: Is higher flexibility for crop rotations possible beyond your study and beyond single point simulations? Because it isn't clear how cover cropping was incorporated in the methods, the applicability of this beyond your study or single point sites isn't clear.

Lines 675-8: I appreciate that there are numerous improvements that will improve CLM. However, none of these seem strongly related to the work presented here. For example, there is no evidence that lack of management or incorrect plant hydraulic properties are contributing to model biases.

―――――――――――――――――――

---

## Author Comment (AC1) · 18 Sep 2020

**Replies to comment by Anonymous Referee #1, 8 September 2020:**

My main critique is that it is difficult to understand the impacts of (1) and (3) in particular. I tried to look into the code, but couldn't quite locate (1) and (3). I suggest the authors to make it clear in the code where these implementations are (perhaps mark it) so that I can follow how much the codes were changed relative to the standard. To gauge theimpacts of (1), I would like to see a winter wheat simulation only at the "DE-RuS" site for with and without (1). You could show how much leaf area index, latent heat and sensible heat of winter wheat changes with this assumption. Similarly, to examine the impact of (3), you could do a simulation of sugar beet and Winter wheat at "DE-RuS" (in this case sugar beet will be rotated, which you already did) and a simulation of Winter wheat only at "DE-RuS". You could also show how much leaf area index, latent heat and sensible heat of winter wheat changes if there was no rotation. Additionally, you can check whether rotation has any impacts on the modeled nitrogen leaching and fixation rates.

Thanks for the suggestion, we will provide the source code files that were mainly modified in pdf format with the most relevant code changes marked as supplement.

In Figure 1 we show the individual effect of the winter wheat subroutines (1) and the modified parameters for winter wheat (2), as well as a combination of both. Figure 1 shows the simulation results of the LAI for one winter wheat year at the DE-RuS site, simulated with: the default CLM configuration using the default parameter set – CLM_D_d (red dashed line), the default CLM configuration using the modified parameter set CLM_D_m (orange), the modified CLM configuration using the default parameter set - CLM_WW_d (blue dahed line), and the modified CLM configuration with the modified parameter set - CLM_WW_m (lightblue).

Using only the modified parameter set with the default model configuration (CLM_D_m) resulted in slightly higher LAI values compared to the default model version (CLM_D_d) but did not reach the observed maximum LAI values and the growth cycle duration. The implementation of the winter wheat subroutines using the default parameter set CLM_WW_d led to a more realistic reproduction of the growth cycle duration compared to CLM_D_d. The combination of the modified parameter set the modified CLM configuration (CLM_WW_m) resulted in the most realistic LAI dynamics.

With the implementation of the cover cropping subroutine we present a rather technical solution to consecutively simulate crop rotations, especially those that include two plantations within one calendar year in order to realistically represent cropland sites. In this study, we did not focus on the biochemical benefits of cover crops or crop rotations but agree that this is an interesting area of application for this routine. For the revised manuscript we will add some more plots on the effects of the crop rotation with the modified CLM_WW_CC compared to what is possible with the default model configuration.

[Figure]

Figure R1: Simulation results of the LAI for one winter wheat year at the DE-RuS site, simulated with: (red dashed line) the default CLM configuration using the default parameter set – CLM_D_d, (orange) the default CLM configuration using the modified parameter set CLM_D_m, (blue dahed line) the modified CLM configuration using the default parameter set - CLM_WW_d, and (lightblue) the modified CLM configuration with the modified parameter set - CLM_WW_m.

**Replies to the list of specific comments:**

(1) While I appreciate some of the details in sections 2.1, and 2.2.1, it would be appropriate it put most of the text in the Appendix section. For example, the paragraph that starts with the description of the default crop phenology scheme (lines 139 to 152 and additional lines) is not new to this study but rather standard CLM5 documentation notes and therefore, they can be put in the Appendix. Similarly, the section about Winter cereal representation that begins with "Vernalization" is also not new to this study. The default phenology scheme of CLM5 has a Vernalization subroutine.

We agree and we will shorten and reorganize this section accordingly.

(2) The authors emphasize the importance of cash crops (e.g. sugar beets and potatoes). I would like the authors to comment on the spatial coverage of these crops in Germany and whether the famers are smallholder or largescale holder plantation owners. Along similar lines, it would be good if the authors could comment on how they plan to carry out the large scale simulations or regional simulations for these crops given that you need time series information about the rotation of these crops and also that some crops might be planted every two years or so.

We agree and will add additional statements on the local importance of the discussed cash crops in the revised manuscript.

Many thanks for the suggestion. In the revised version we will discuss how large scale simulations could be used to test 'conceptual' cover cropping schemes. For example, the effect of an overall coverage of greening mix during winter months on all crop land units where summer cash crops are planted and that would otherwise be fallow by default during winter. This could also be tested for specific cash crops only. Also it would be possible to simulate cover crop plantations based on harvest date thresholds. Here, a defined maximum harvest date for any specific cash crop could define whether a cover crop such as winter wheat would be planted or not. For example, for all sugar beet land units with harvest dates before a certain threshold (e.g. day 290 of any given year) winter wheat could be planted as a cover crop during winter. If this harvest threshold is not reached and the summer crop is

harvested late in the year, no cover crop would be planted. Alternatively, these harvest thresholds could define the type of cover crop, e.g. early harvest - winter wheat, late harvest – simple greening mix, etc.

(3) A number of statements in the results section is difficult to follow. For example, in lines 407 to 412, there is no reference to any figures. What is green leaf area index in line 408? Do you mean before maturity, during maturity or after maturity?

(new line 407-412) For the BE-Lon site, CLM_WW simulated average LAI peak magnitudes, as well as seasonal LAI variations, are close to the observations. An exception is the year of 2015, where unusually high LAI values where observed in May and June, ranging from 5.40 to 6.38 $m^2/m^2$ (Figure 2). In general, the peak of CLM_WW simulated LAI occurred approximately one month earlier than observed in the field which in turn led to an earlier simulated harvest date of the crop compared to the observed harvest dates (Figure 2, Table 4).

(4) I think the poor seasonal dynamics and low magnitude of the leaf area index in Figures 2-5 of CLM-D could also be related to the parameter values rather than the winter wheat subroutine that was introduced in this study. There are at least 3 parameter values that are considerably different compared to the default parameters of CLM ('gddmin','hybggg' and 'graincn'). For example, I see that the default gddmin is 50 in the default but 100 in the modified case (this study). Also hybgdd in the modified case is 30 more than the default. So couldn't these likely explain poor seasonal dynamics and low magnitude of the leaf area index in Figures 2-5 of CLM-D?

We think this is already answered by Figure R1 where we distinguished the effects of the modified parameter set and the new winter cereal representation, as well as a combination of both.

**Replies to the list of minor comments:**

(1) In line 70, Bilinois et al. (2015) is cited but I think the reference is missing.

Thanks, we will correct this.

(2) Please provide fractions of sand, silt and clay in Table 2, maybe up to 5 cm or 10 cm?

We agree and will add Table R1 to the revised manuscript.

Table R1: Textural fractions (sand, silt and clay percentages) at the ICOS and TERENO cropland study sites averaged for the upper soil layers (up to 50 cm) with corresponding reference.

| Site/ID | Sand [%] | Silt [%] | Clay [%] | Ref. |
|---|---|---|---|---|
| Selhausen/DE-RuS | 16.4 | 63.4 | 14.9 | Brogi et al. (2018) |
| Merzenhausen/DE-RuM | 16.4* | 63.4* | 14.9* | - |

| Klingenberg/DE-Kli | 21.5 | 22.8 | 55.7 | Grünwald (personal communication, 2020) |
|---|---|---|---|---|
| Lonzée/BE-Lon | 5-10 | 68-77 | 18-22 | Moreaux et al. (2006) |

*adopted from the DE-RuS site

(3) While I agree with the statement (line 289) that "CLM5 only permits land use changes at the beginning of every year", users can start a CLM5 simulation in any month the land use change actually happens in real life by performing a 'clear-cut' following spin-up, for example.

Yes, re-starting a simulation at any month is possible in order to change CFT. However, this would require a manual restart at every time the CFT/PFT changes and does not allow a consecutive simulation with flexible land use changes. While this can be done for point cases, it is not feasible for regional scale cases where CFTs might change at different times on different land units.

(4) At the "BE-Lon" site, the LAI curve of winter wheat from DOY= 0 to DOY = 100 seems to have a relatively gradual and smooth growth (Figure 2) while at sites "DERuS", "DE-RuM", "DE-Kli", the growth is relatively sudden and steep during the same period. I would like the authors to provide some explanations for this difference.

We think this is related to the temperature at the BE-Lon site. Here, recorded temperatures very higher in February and allowed for more simulated growth compared to the other sites. We will add a more detailed discussion on this to the revised version of the paper.

(5) In lines 602 to 603, the authors claim that CLM5 does not represent timing of fertilizer. Please provide a citation for this?

Unfortunately, CLM5 is not flexible enough to represent the complex management practices concerning timing and type of fertilization (Lawrence et al., 2018). Fertilization dynamics and annual fertilizer amounts depend on the crop functional types. For all cropping units, mineral fertilizer application starts during the leaf emergence phase of crop growth and continues for 20 days. We will add a short explanation in the revised version of the paper.

(6) In line 603, the authors state that CLM5 does not consider varieties of winter wheat. I agree with this statement but at the same time, many land surface models don't consider varieties or cultivars of crops. Crops can be genetically modified to boost productivity. This means there could large differences in the parameter estimates among varieties/cultivars. The authors could discuss the variation in the parameter estimates if they are measured at their sites.

Thanks for pointing this out. We will add a discussion that the structure of CLM5 allows to include easily more CFTs, e.g. from increasing availability of plant physiological trait information. There is substantial development work being done for CLM in order to include plant trait information, e.g. to allow the prediction of biome boundaries directly from plant physiological traits via their competitive interactions (Fisher et al., 2015).

(7) The authors mention in lines 626 to 629 the following: "There is a tool available for CLM5 that enables the simulation of transient land use and land cover changes (LULCC) (Lawrence et al., 2018). It was designed to simulate and study the effects of changing distributions of natural and crop vegetation, e.g. land use change from forest to agricultural fields (Lawrence et al., 2018), rather than inter-annual changes of agricultural management on crop vegetated

areas." I'm confused about the last part "rather than inter-annual changes of agricultural management on crop vegetated areas". Please explain what do you mean by this? Do you mean you cannot change the Nitrogen fertilization rate from year to year in CLM5?

With this we wanted to emphasize that although this tool allows changes in land use every year (on 1st of January), it does not account for changes happening during the year (e.g. several crop growth cycles or changes from summer to winter crop in fall) or multiple crop growth cycles within one year (e.g. multiple growth cycles of the same cash crop within one year in India due to several monsoon seasons). The annual amount of mineral nitrogen fertilization is assigned by plant/crop functional type and can be changed manually for each year. We will clarify this is the revised version of the manuscript.

---

## Author Comment (AC2) · 5 Oct 2020

**Abstract**.

**Line 25**: Is cover cropping only common in humid and sub-humid regions? Perhaps it would be more informative to rephrase to something similar to: ".. which is an agricultural management technique commonly used in the regions evaluated in this study." Alternatively, you can say that it is a technique growing in popularity to improve soil health and carbon storage.

> Thank you very much for your suggestion, which we will gladly incorporate.

**Line 26:** Are you referring to the parameterization of new CFTs? Please clarify.

> Yes, we will clarify this by using the new wording as suggested below.

**Line 27:** Please move the reference of RSME for LH and SH to just after the energy fluxes rather than after NEE.

> Thanks, we will incorporate this suggestion.

**Line 31:** When you refer to the "LAI curve", is this the same as the season cycle of LAI? If so, please modify the wording to reflect this.

> We will rephrase this into "terms of LAI magnitudes and seasonal cycle of LAI".

**Lines 31-33:** It would be more impactful if you strengthened the last sentence in the abstract. Here is one suggestion: "Our modifications significantly improve model simulations and should therefore be used in future simulations to better understand large-scale impacts of agricultural management on carbon, water, and energy fluxes."

> Thanks, we agree and will rephrase accordingly.

**Introduction**: Overall, the introduction needs some reorganization. You need to more clearly highlight the role of management (make this a separate paragraph, include cover crops but also other types of management). The new representation of cover crops is a primary contribution to this paper and is barely mentioned here. The introduction also needs a broader overview of crops in LSMs (it currently only focuses on AgroIBIS and CLM). Last, most of the introduction emphasizes the global nature of models and that the variation in soils, plants, climate is important. When the reader finally gets to the end of the introduction, which highlights that you focus on a few sites in Europe (which some may argue has narrower variation in soils, plants, climate than if you were to compare to locations from other continents), it makes this study seem limited. It might help to instead describe that models are still limited by their ability to represent many crop types and important management practices, emphasizing the importance of your work adding these new capabilities, and also to highlight that Europe is a major agricultural hub for global food production.

> Thank you for your constructive suggestions. We agree and will re-evaluate and reorganize our introduction for the revised manuscript.

**Lines 44-49:** The mention of cover crops here seems a bit out of place. The earlier part of this paragraph and the start of the next paragraph is focused on adaptation to climate change, whereas the description of cover crops here focuses on soil benefits and climate mitigation. I suggest reorganizing, moving the cover crop description to later in the introduction.

> Thanks for your suggestions, we agree and will restructure the text.

**Lines 68-70:** I'm not sure I entirely understand the point of this sentence. Is this just to highlight the evaluation of crops in CLM4.5?

> Here we give a short overview of improvements in earlier versions of CLM. We agree that it might not be necessary here and will restructure the text accordingly.

**Lines 73-76:** You should also reference the CLM5 crop overview paper here, which evaluates global crop yields: Lombardozzi, D. L., Y. Lu, P. J. Lawrence, D. M. Lawrence, S. Swenson,

K. W. Oleson, W. R. Wieder, and E. A. Ainsworth (2020), Simulating Agriculture in the Community Land Model Version 5, J Geophys Res-Biogeo, 125(8), 927–19, doi:10.1029/2019JG005529.

> Thanks for pointing this out, we will add this to our references.

**Lines 77-84:** This paragraph seems too detailed for the introduction. I suggest summarizing and merging with the previous paragraph. For example: "The few studies that have evaluated CLM5 suggest inaccurate phenology and overestimated crop yields (Sheng et al. 2018)." However, you'll probably want to change/update this to also incorporate results from the Lombardozzi et al. CLM5 paper mentioned above.

> Thanks a lot for your suggestions on the introduction. We will restructure the introduction taking your comments into account and agree that Lombardozzi et al. (2020) should be added and cited here and will do so in the revised manuscript.

**Methods Overall**: The methods section needs to be tidied up. There are redundancies in the first section, and a lack of detail in the cover crop description. Please pay careful attention to providing enough detail that the reader isn't left wondering how something was done, but keep the text succinct. You reference Lawrence et al. 2018 in several places throughout the text. However, I believe this paper was published in 2019 (not 2018). Please double-check.

> Thanks for your suggestions. We will re-evaluate and restructure several parts of our methods section in the revised manuscript as discussed below. We agree that Lawrence et al. (2019) should be added and will do so in the revised manuscript.

**Section 2.1:**
When describing the crop model, please also cite Lombardozzi et al. (2020), as this has much more detailed information about the crop model updates than Lawrence et al. 2019.
The methods should be streamlined to avoid repetition. For example, allocation is mentioned in lines 134-138, and then again in the paragraph starting at line 153. When referring to C allocation, you state that it varies throughout the growing season (e.g., line 156), whereas the reference to N allocation states that it uses two different C/N ratios (lines 161-162). However, these are treated the same way in the model. Please update for consistency. I suggest switching the order of Eq. 1 and Eq. 2.

> Thanks, we will incorporate this suggestion.

**Line 114:** Please define "CFT" the first time you use this term.

> We will do this as suggested by the reviewer.

**Line 115:** land units are not separated by fertilizer, only by irrigation. Please update.

> We agree that the phrasing is misleading and will update it accordingly.

**Lines 204-206:** This is a bit confusing and could use clarification. Does the vernalization factor always range from 0-1? Is it applied to GDD for air and soil temperatures (e.g., does it affect all phenological phases)? If it is only applied to grain C allocation, where does the remaining C get allocated?

> Yes, the vernalization factor ranges from 0 to 1 (fully vernalized) and affects the GDD in the phenology phase after planting (vernalization starts after leaf emergence and ends before flowering). This leads to a reduced growth when the plant is not fully vernalized and the vf is smaller than 1:
>
> For vf <1
>
> $GDD_o * vf = GDD_n$ with $GDD_n < GDD_o$
>
> where the subscripts $o$ and $n$ stand for original and updated GDD.

We will better clarify the methodology of this routine by giving more details in section 2.2.1 or in the appendix (please see our replies below to section 2.2.1).

**Section 2.2.1**
It would be helpful to start with an overview of how winter cereal representation differs from other crops. I suggest a high level overview of why it's important to include both vernalization and cold tolerance before diving into the details of each.

Thanks for your suggestions. We will add more detail to this part in the revised manuscript.

**Equation 4**: You specify that Tcrown is slightly higher than the freezing temperature when covered by snow. I see that snow height is used in the calculation, but where is the plant height? Without including the plant height, how do you know whether the plant is covered by snow?

In CLM, the crown temperature is the crown depth soil temperature calculated as a function of 2m air temperature and snow depth. The crown is the connecting tissue between the roots and the shoots at the base of the plant. For winter wheat, the crown node is located at about 3 – 5 cm soil depth (Aase and Siddoway, 1979). We will add this explanation and reference in the revised manuscript.

**Line 213:** The text describes what the accumulative parameters are, but what about the previous time step is used? It would also be useful to include a brief description of how some of the accumulative parameters accumulate (e.g., are these all based on some aspect of accumulated temperature?)

We will include a more detailed description and add some equations from the source literature Lu et al. (2017) in order to clarify this.

**Equation 6:** Please define the "alpha-surv" and the "t" variables in this equation.

Thanks, we will add this.

**Equation 7:** I am confused by this, partly because it's not clear what the equation is taking the max of. Also, can Tcrown be negative? That seems to be the only way the solution to this equation isn't 0. Please update to clarify. Also, I think 'fsurf' should in fact be 'fsurv'.

Thanks, $f_{surv}$ is correct.

**Paragraph starting at line 227:** I find the description here a little confusing. Can you revise this to more clearly articulate the difference between survival probability and WDD? Is survival probability just a step function, where any value <1 causes the same amount of damage (simulated as part of the C and N pools being transferred to litter)? Should I be thinking of survival probability as the proportion of the plant that survives, or the probability that the whole plant survives? Also, part of my confusion is that this is the first place that a frost damage function is mentioned.

Thanks for your suggestions on this section. We are currently evaluating whether to move some of the very detailed description to the appendix as it is not new to our study (as suggested by the other referee). We hope to clarify several of your specific comments above by adding more detail to these descriptions in the revised version of the manuscript. In section 2.2.1 of the paper we would then give a slightly broader overview on winter cereal representation following your suggestions.

**Section 2.2.2:**
Since you use a pre-existing winter wheat parameterization, it would be helpful to include some information about what you changed in the parameterization and why.

For the CFTs sugar beet and potatoes, the same parameters as for spring wheat are used on the default parameter set due to the lack of parameters specifically calibrated for these crops. For winter wheat, there are pre-existing crop-specific parameters available on the default parameter set. However, this default parameterization of winter wheat performed poorly in representing the crop phenology for the evaluated study sites in this study. This was also reported in an earlier study by Lu et al. (2017). Thus, crop specific parameters were also added for the winter wheat CFT.

We will include an additional text in the revised manuscript to clarify this.

**Table 1:** This is a useful summary, but I'm not sure it adds much information to the main text.

We believe that this overview table is very helpful information for the reader. Thus, we would like to keep it.

**Section 2.2.3:** How do you determine when the cover crops (or rotations) are planted and the subsequent phenology phases? Is it based on GDD? Did you have to modify GDD parameters or add new ones? Did you add new CFTs to accomplish this? How is allocation determined? This section needs more detail about how modifications were made, as it is the bulk of the development work in this paper.

The rotation schemes are hardcoded in the new cover cropping subroutine. Basically, in the new routine, the phenology algorithm is reset and restarted after harvest of any crop that is assigned with the cover crop flag. We are currently working on a version where the rotation is more flexible and user-friendly defined by a control file.

**Lines 267-270:** It's great to hear that you introduced a flag to use the cover crop option, but I'm not sure you need to include that description here.

We will add more detail to the description of the new routine and will keep this information in the revised manuscript.

**Lines 276-277:** How are you predefining a rotation scheme?

At this stage this is hardcoded in the new cover cropping subroutine. However, we are currently working on a version where the rotation is more flexible and user-friendly defined by a control file

**Line 283:** "catch crop" this is the first time you mention it. Are you using this interchangeably with cover crop (which is how you described this in the previous sentence), or are you using a new phrase to distinguish this from cover crop? Please be clear and consistent with word choices.

Here the terms "cover crop" and "catch crop" were used synonymously. We will correct the wording to be more consistent.

**Line 283:** You mention plowing the crop into the soil. However, CLM does not represent plowing. How did you accomplish this. Do you assume that the plant biomass is transferred to the litter pool? Also, how did you decide when this happens?

We agree that this could be misleading. The plant biomass is moved to the litter pool instead of to the grain product pool. This is done upon harvest of the crop. We will clarify this in the revised manuscript.

**Section 2.3:**
I think it would help to describe the sites before the validation data, and/or mention whether you run CLM simulations at these sites. This section starts by describing validation data, but does not mention what is being validated.

We will restructure the text accordingly.

**Table 2:** Useful information about the sites, but I think the map describes the locations quite well, and most of the other information included in the table is not used in the simulation. Therefore, I'm not sure that this table is necessary in the main text.

We think this table gives the reader a nice overview without having to read this section in detail and therefore would like to keep it in the main text. We will include an additional table with textural fractions at the study sites in the appendix of the revised paper as requested by RC1.

**Lines 318-319**: You mention winter wheat twice here.

Thanks, we will correct this.

**Lines 341-342:** CLM's default time step is 30 minutes.

Here we mean the customized time step of input forcing data, which was set to hourly. Not all meteorological input data was available half-hourly, thus an hourly temporal resolution was used. The internal model time step remains at 30 minutes.

**Section 3.1:** Throughout this section, the differences in model version versus parameter set seem to be conflated. Please make this much clearer throughout, explaining what each of the model versions includes and what the default versus modified parameter sets include.

Thanks for your suggestions on this section. We will change wording accordingly, as discussed above, in the revised manuscript.

**Table 3:** Which simulations include the potato and sugar beet parameterization? It looks like it's the CLM_WW simulation, but this needs to be explicitly mentioned in the table description.

We agree that more clarification is needed in this section. This table was meant to give an overview of all the simulation runs that were compared in this study. We will change the wording and include clearer descriptions in the text.

**Lines 364-366:** This text is confusing: It is not clear what the difference is between the default model and the modified model. I assumed the "default" model did not include winter wheat, but this text suggests that it does. How, then, is the default model run with the modified winter wheat parameters different from the winter wheat model with the modified parameters?

The CFT of winter wheat is included in the default model but its specific parameter set yielded very poor representation of simulated winter wheat phenology at our sites and also in previous studies. Thus, next to the implementation of vernalization and cold tolerance representation in the model code, new crop specific parameters were supplied in order to optimize the model performance. Please see also our replies to RC1, where we supplied an additional plot showing the differences in simulation results using both the default model with and without new parameterization and the extended model with and without new parameterization.

We will reorganize the text and table to increase the comprehensibility.

**Lines 369-370:** What are the default parameterizations of sugar been and potatoes? These aren't included in CLM, so is there a "default"?

Sugar beet and potatoes are included in the structure of the CLM5 crop module and are amongst the 64 CFTs. The CFTs sugar beet and potatoes do not have assigned parameters specifically calibrated for these crops, instead the same parameters as for spring wheat are set as default for these CFTs. We will rephrase the text using the term 'control' instead of 'CLM_D'.

**Section 4**

In general, I find the use of CLM_D, CLM_WW, and CLM_WW_CC to be confusing, as the changes included in each are not clearly described. Additionally, it seems that sugar beet and potato parameterizations are added to CLM_WW. It might be more helpful to instead refer to CLM_D as "control" or "default" and then refer to updated parameterization (e.g., "improvements to winter wheat" rather than "CLM_WW" in Section 4.1 and "new potato" or "new sugar beet" parameterization in Section 4.2). Additionally, throughout this section, figures should include estimates of uncertainty.

> Thanks for your suggestions. We agree that the different model versions and parameterization approaches should be described in more detail. We will improve the descriptions of the model versions and the table in section 3.1 accordingly.
>
> Due to the small number of compared years (2 to max. 6 years), uncertainty estimates might not add much value to the plots. We may present the uncertainties in additional plots in the supplement.

**Section 4.1:** Throughout this section, the text could be streamlined to avoid repeating the description of trends for each site (see note below about Figs. 2-5). Additionally, the trends in energy fluxes are barely mentioned, leaving the reader wondering why you show these in Fig. 2-5, particularly since their mention focuses on cumulated monthly sums (which aren't shown). Also, yields are discussed frequently throughout the text in this section. Is it worth making a bar chart of yields to more clearly illustrate their evaluation? I realize that a bar chart may look busy, but perhaps averaging across years for the sites with multiple years and including standard deviations will work. Related, how are you calculating yields from CLM simulations? It's important to use the peak daily grain carbon value for the entire growth cycle rather than averaging this over some period of time.

I suggest reorganizing the text (and figures) have 4 paragraphs, focusing on the descriptions of: 1) LAI ; 2) yields; 3) NEE; and 4) energy fluxes. Highlight differences among sites within each paragraph. You can also include an opening paragraph that mentions that CLM_WW improves trends for nearly all variables compared to CLM_D, so the remainder of the discussion focuses on the evaluation of CLM_WW.

> Thanks for your suggestions. We will consider rearranging the text and figures accordingly. Also, we will consider adding a plot similar to Figure R1 below to the main text or the appendix in the revised manuscript.
>
> The simulated crop yield was calculated from the peak value of daily grain carbon.

[Figure]

Figure R1: Simulated annual grain yield [tDM/ha] using the default model (orange) and the modified model and crop-specific parameterization (blue), compared to recorded harvest dates (grey) for all simulated winter wheat years at the sites BE-Lon, DE-RuS, DE-RuM and DE-Kli.

**Figures 2-5:** Is it possible to compile these into a single, multi-panel figure? Given that they all show the same variables for different sites, a single panel would allow the reader to compare across sites more easily. Another, possibly better, alternative is to combine all sites and separate the figures into LAI (Fig. 2) and energy fluxes (LH, SH in Fig. 3). It would also allow you to streamline the description of trends throughout Section 4.1. ã˘A˘ TIf I understand the legends correctly, simulations and observations in Figs. 2 and 5 are averaged over multiple years. Can you add uncertainty estimates to these plots? If you plot all individual years (it looks like you possibly do that for observations, but not model), it might be easier to plot averages across years and then plot the uncertainty range associated with interannual variability.

Thanks for your suggestion. We will re-evaluate the plots and try out other options, such as the suggested multi-panel plot. Originally, these results were plotted individually for each site to be able to look at the data in more detail. In addition, we will add estimates of uncertainty to the figures when appropriate.

**Fig. 2** states that the observations are GLAI, whereas Figs. 3-5 state that the observations are LAI. Are the observations LAI, GLAI, or does this vary by location? If it is different by location and both LAI and GLAI are used, how might this change the ability to evaluate CLM?

For winter wheat, the green leaf area index (one-sided green leaf area per unit ground surface area) is measured in the field and compared to CLM simulated LAI (defined as one-sided leaf area index, no burying by snow). We will add this in the revised manuscript.

**Fig. 5:** There aren't any LAI observations plotted in panel a, yet the figure legend suggests that there should be site observation data for LAI.

Unfortunately, there are no LAI observations available for that year. The legend also refers to the two plots below and therefore shows observations, too. We will consider this when rearranging the plots as suggested above.

**Lines 394-5:** As you state, it looks like the LAI peak is indeed too early. However, even more noticeable (and not mentioned) is the fact that the LAI peak looks to be dramatically underestimated.

Thanks, we will add this comment in the revised manuscript.

**Lines 413-4:** Table 4 suggests that crops are only harvested _ 1 month too early, but there are higher observed LH fluxes later in the season than just one month. Is this due to cover cropping, which is not included here?

Yes, the later peak in LH fluxes is related to cover cropping. At the Be-Lon site, mustard is often planted after winter wheat which is not represented in this simulations. This issue is discussed in a later section but we will include a short mention in this section as well.

**Line 420:** I think the phrasing "overestimated early growing season LAI" is potentially misleading. While it is technically correct, the simulated peak LAI values are actually similar to observed peak LAI values, but happen earlier in the year. I think it might be more informative to state that the peak magnitudes are similar, but that the peaks happen too early in the year.

We agree and will rephrase.

**Lines 422-3:** What does "growing cycle" refer to hear? As you mentioned earlier, LAI peaks too early and planting and harvest start early, suggesting that phenology is not accurate. Therefore, it is unclear what you mean by "generally good correspondence in growing cycle and LAI".

What is meant is the reasonable agreement of modelled and observed LAI magnitudes throughout the growing cycle, which is also reflected in the simulated grain yield, despite shifted planting and harvest dates. We will reformulate this statement to make it more comprehensible.

**Lines 437-8:** How can you say that CLM_WW resulted in more realistic magnitudes when you stated in the previous sentence that observations aren't available?

Although there are no observations available for the LAI at this site, we can still see that CLM_WW simulated more meaningful magnitudes of LAI compared to default simulations.

**Lines 438-9:** This is confusing. Does it refer to only the simulations, or also reference the observations? I get the sense that you are conflating simulated peak LAI with simulated and observed crop yields. It implies that lower LAI causes the lower crop yields, although I don't think you can say that for sure.

Thanks for pointing this out. For the phenology algorithm within CLM5, a lower LAI can indicate lower grain yield due to less growth. For observed crop yields, there is a multitude of factors that influences the grain yield, which is not necessarily reflected in the LAI. Here, we see that the lower simulated LAI corresponds to a lower simulated grain yield in comparison to the other sites. However, for the BE-Lon site for example simulated LAI is lower but yield is higher compared to DE-RuS site. This is explainable by an exposure of low temperatures at DE-RuS (temperatures are generally slightly higher at BE-Lon early in the year) that affected the grain carbon due to the frost damage routine that was implemented (section 2.2.1). Damage due to low temperatures affects grain yield more than growth and is therefore not represented in the LAI. We agree that a short discussion on this is missing in the manuscript and we will extend the revised version accordingly.

**Line 440:** I think this may be backwards. Table 4 suggests that yields are overestimated in 2011 and match really well for 2016.

Thanks, we will correct this.

**Lines 453-4:** Are all the subsequent mentions (including the metrics in Table 5) calculated using the cumulative monthly sums?

The metrics were calculated using simulation output and observation data at daily time step. We will add this information in the text.

**Line 459:** You just stated that the BE-Lon sites high some of the highest correlations in the previous sentence, and here single out this site as having high RMSE and biases with low correlations.

Thanks for pointing this out. We also calculated metrics for the whole year instead of only the time between planting and harvest, where low correlations were found for BE-Lon. We do not show these metrics and will consider adding them in the Annex of the revised manuscript. Although energy fluxes match reasonable well during the growing cycle of the crop, the annual correlation is relatively poor due to the influence of cover cropping and poor representation of post-harvest field conditions. We will correct this in the revised manuscript.

**Lines 460-461:** This sentence should be moved to above, where you briefly mention the mismatch in late-season LH. Also, how does this affect the metrics in Table 5 (see above comments as well).

Thanks for pointing this out. The metrics in Table 5 are calculated for the time between recorded planting and harvest of the crop and thus not affected by this. Please also see our reply to the comment above.

**Line 464:** Are you referring to CLM_WW? I suggest clarifying here, as you do include simulations that represent cover crops.

Up to this point, all simulations were run with either the default model or the model including the new winter cereal representation. The cover cropping approach is then introduced below. We see that throughout the text the usage of model versions can be confusing and will try to amend this by reorganizing wording and structure as mentioned above.

**Lines 471-2:** It is not quite accurate to say that NEE observations match better due to improved LAI. Consider changing to: "in part due to the better representation of LAI".

We agree and will change the text accordingly.

**Line 473:** Are you actually using cumulative monthly values? Fig. 6 show NEE in unites of umol CO2/mˆ2/sˆ1.

We are using average NEE rates in umol $CO_2/m^{-2}/s^{-1}$ for the respective month. We will rephrase in the figures and text for better clarity.

**Line 475:** Both sites? You mentioned three in the previous sentence. If only referring to two sites, please specify which ones.

We were referring to BE-Lon and DE-RuS.

**Section 4.2:**
Perhaps this should be titled "New Parameterizations" or "Sugar beet and Potato Parameterizations" to distinguish from the modified winter wheat parameterization in Section 4.1 The evaluation of corn here seems a bit out of place since this section focuses primarily on the new parameterizations. I'm not sure where it goes (perhaps in supplemental?), though. Perhaps this section could be refocused as "Evaluation of other crop types", which includes corn and also the new crop types.

We will think about splitting this into two separate sections (i.e. Evaluation of CLM5 default crop types and Evaluation of new crop-specific parameterization for sugar beet and potatoes), and maybe moving parts of it to the appendix.

**Lines 489-91:** I suggest rephrasing to add some detail: "The modifications to winter wheat in CLM_WW do not affect other crop types. Therefore, we add new parameterizations for sugar beet and potatoes to this code."

We agree and will change the text accordingly.

**Lines 502-4:** Is this parameter set modified, or new? What is it strongly improved compared to, if these didn't exist in CLM? I assume it was compared to the default CLM crop model (where the crop might be represented by another type of crop), and it would help to know for sure.

We used the term 'modified' as there is already a 'placeholder' for both sugar beet and potatoes available in the default CLM structure (with parameters adapted from the spring wheat CFT) and we did not have to change or add much to the structure of the model but rather replace single parameters on the input plant parameter file. As mentioned above, we will change wording to improve the clarity of the text.

**Line 507:** You reference spring wheat here. Is this the crop type that default CLM uses for these sites? If so, you might want to make this clearer (and mention it earlier). For example:

"The default parameterization in CLM uses spring wheat for these crop types and effectively reproduced the growth cycle and seasonal LAI, simulations using the potato and sugar beet parameterizations better captured harvest date and growth cycle. "

> Thanks for your suggestion. We will also clarify this according to your comments above at an earlier stage in the text.

**Line 509:** As in previous comments, I don't think "modified" is the best way to describe this. I suggest using "crop-specific parameters" or "parameterizations for new crop types" or similar. As far as I understand, parameters for new crop types were added, not modified.

> We agree to use better wording such as 'crop-specific parameters'.

**Lines 510-2:** It looks like the latent heat flux is very similar for the other site, which might be worth mentioning.

> Thanks for pointing this out. We will mention this fact in the revised manuscript.

**Lines 528-30:** Performed better for NEE? Please clarify.

> Simulations of the NEE using the crop specific parameter set yielded a slightly better correlation of 0.58, compared to the control simulation that resulted in a correlation of 0.43 (Table 6). We will clarify this in the revised manuscript

**Figures 8 & 9:** I suggest updating the use of "default" and "modified" here based on above comments ⌣A⌣T Please specify that the LAI results are daily (if they indeed are).
In previous figures, NEE is described as "cumulative monthly", but here is described as "monthly averaged". Can these be calculated and referred to in the same way for consistency?

> Thanks for pointing this out. The NEE is the average daily NEE per month, not cumulative monthly sums. We will rephrase this in the text and figures accordingly. LAI observations are single field measurements (point observations).

**Section 4.3** It seems that this section focuses on crop rotation as much as cover cropping. I suggest updating the heading to "Cover cropping and crop rotation" or similar to reflect this.

> Thanks for the suggestion. We agree and will change this in the revised manuscript.

**Lines 553-4:** Is the simulation of a second crop growth onset for the same crop or for the cover crop? The current wording suggests that a second onset is for the same crop within one year AND for the cover crop. If this isn't intended, perhaps change to "simulation of a cover crop as a second crop growth onset within a single year"

> The focus is set on the second onset within a single year. Both a second onset of the cash crop, as well as the onset of a cover crop are possible. We will rephrase the text for more clarity.

**Line 556:** "Greening mix" ⌣A⌣Tis this the same as cover crop, catch crop? Please be consistent in your terminology throughout.

> Greening mix here is simply an example for a cover crop that was planted that year at DE-RuS. We used the terms 'catch crop' and 'cover crop' synonymously and will update this in the text and figures.

**Lines 556-557:** Perhaps it would be more accurate to say "the cash crop rotation of barley (simulated using the spring wheat CFT)".

> Thanks, we will rephrase this.

**Line 557:** Spring wheat in CLM is not considered a perennial. It can simulate multiple years of spring wheat in a row, but that doesn't make it perennial.

Thank you for pointing this out. We did not mean to say that spring wheat is a perennial crop but that it is simulated repeatedly over multiple years. We agree that the use of 'perennial' is wrong in this context and will correct this.

**Lines 559-561:** Can the effects of planting cover crops and the crop rotation be isolated?

Here, we wanted to show that not only an easier crop rotation is possible (especially from summer to winter crop) but also the simulation of a crop that is not considered a cash crop. Technically, this follows the same schematic.

**Line 563:** Please change "plantation" to "planting"

Thanks, we will correct this.

**Line 576:** Similar to above, spring wheat is not a perennial crop in CLM, as it's planted every growing season.

Thank you for pointing this out. Please see our reply to line comment 557 above.

**Figures 10-11:** It looks like these are for the same site and continuous. Why not plot the full time series on the same panel, adding lines or shading to show the transitions and associated crop type labels. Also, do you not have observational data for LH for 2017-2019 (Fig. 11)?

We will take your suggestion into account and re-evaluate these two plots. Also, we will add a plot for latent heat flux to Figure 11.

**Section 5**
In addition to the benefits and challenges of the new model developments that you include, I was hoping to see further big-picture discussion, for example about how these new developments might improve future large-scale simulations, possible interactions with climate, etc. Consider adding a paragraph to highlight how your improvements can improve our understanding of larger-scale processes. Also, NEE isn't mentioned at all. Why do you think that NEE didn't improve as much as energy fluxes?

Thanks for your suggestions, we will add more detail to our discussion in the revised manuscript.

Field observations indicate that heterotrophic respiration from soil organic matter and litter acts as a carbon source, which is not simulated well in CLM. This is one of the reasons why the correlation of the NEE is relatively low.

**Lines 597-8:** As mentioned in a previous comment, higher LAI does not mean higher grain yield. There are many factors that affect yield, including photosynthetic rate, nutrient availability, etc. Also, the results presented in this sentence further support that LAI does not directly correspond to yield: grain yield was higher at BE-Lon (which had lower LAI) than DE-Rus.

Thanks for pointing this out. We agree that LAI is not the only factor governing yield and will change the text accordingly. Please see also our reply to line comment 438-9 above.

**Line 603:** CLM may not represent different varieties, but the parameters could be changed (as you did here) to represent different varieties, especially when simulated at single points.

In this study, different cultivars were not considered as CLM5 offers only one CFT for winter wheat representing all varieties. The list of CTFs could be extended with suitable plant parameterizations, but this information is not available due to missing measurements and due to the complexity of the phenology algorithm and parameter scheme. The introduction of a phenology scheme based on plant physiological trait information could be a major improvement in this field (e.g. Fisher et al., 2019), as plant trait information becomes more readily available. Whether including different varieties

and cultivars of one crop would have a significant impact on the performance of regional or global scale simulations remains to be evaluated. We will rephrase this accordingly in the revised version of the manuscript.

**Line 607:** It might be clearer to say "The early leaf onset and harvest for winter wheat simulated by CLM: : :"

We agree, thanks.

**Lines 619-22:** Can this be more specific? How would discretizing plant hydraulic properties improve yield prediction? Also, why does the reference include "Daniel"? How could the properties (parameters?) be estimated by inverse modeling or data assimilation?

There are many variables that influence CLM simulated yield, e.g. LAI cycle and peak, length of the leaf emergence phase, harvest date and soil water available for root water uptake. Except for soil moisture, these variables are strongly determined by the GDD scheme, which suggests that the simulated crop yield strongly depends on the GDD. Also, the carbon allocation of crops is strongly limited by soil water available to the plant. An improved soil hydrology (i.e. the inter-annual variability of soil moisture) and representation of plant hydrology could help to improve future yield predictions with CLM5. Crop properties could be estimated by assimilating measurement data like NEE, LAI, soil moisture and/or energy fluxes using for example an augmented state-vector approach.

We will add more detail to this part of the discussion in the revised manuscript.

**Lines 629-31:** Why isn't it applicable to regional simulations? If a simulation is set up to use land use change, the distributions of vegetation, including crop types, will change, even on a point scale, and can be customized by the user if desired.

This tool is useful to study general land use changes by changing the land cover of individual land units, e.g. from naturally vegetated to cropland or urban, from forest to cropland, from C3 to C4 cropping etc. However, it lacks flexibility in accounting for changes within land units of the same land cover and does not account for all 64 CFTs. Also, changes always happen on the 1st of January of the given year. We agree that it is generally applicable to regional scale and will rephrase the text for better comprehensibility.

**Line 634:** Do you mean before fall of 2018? Fall of 2017 would be the same year.

Yes, we meant before fall of 2018, thanks for pointing this out.

**Line 635:** I don't see Figure 12.

It should be Figure 11, we will correct this.

**Section 6**
**Line 665**: Is higher flexibility for crop rotations possible beyond your study and beyond single point simulations? Because it isn't clear how cover cropping was incorporated in the methods, the applicability of this beyond your study or single point sites isn't clear.

In the revised manuscript we will discuss how large scale simulations could be used to test 'conceptual' cover cropping schemes. For example, the effect of an overall coverage of greening mix during winter months on all crop land units where summer cash crops are planted and that would otherwise be fallow by default during winter. This could also be tested for specific cash crops only. In addition, it would be possible to simulate cover crop plantations based on harvest date thresholds. Here, a defined maximum harvest date for any specific cash crop could define whether a cover crop such as winter wheat would

be planted or not. For example, for all sugar beet land units with harvest dates before a certain threshold (e.g. day 290 of any given year) winter wheat could be planted as a cover crop during winter. If this harvest threshold would not be reached and the summer crop is harvested late in the year, no cover crop would be planted. Alternatively, these harvest thresholds could define the type of cover crop, e.g. early harvest - winter wheat, late harvest – simple greening mix, etc.

**Lines 675-8:** I appreciate that there are numerous improvements that will improve CLM. However, none of these seem strongly related to the work presented here. For example, there is no evidence that lack of management or incorrect plant hydraulic properties are contributing to model biases.

We agree. Before addressing general points of improvement for the crop module, we will discuss how the modification presented in this study could be further developed and evaluated in order to increase their applicability.

---

## Author Response (AR1)

Dear Associate Editor,

We want to thank the Reviewers for their thorough review of our manuscript and their constructive suggestions which helped us improve the quality of this study.

We gladly incorporated these suggestions in the revised manuscript and corrected all minor inconsistencies as pointed out by the Reviewers. It became clear to us from both reviews that more clarity was needed to distinguish the different simulation run scenarios. We therefore changed the naming of the respective parameter set and model version combinations throughout the text and on figures and tables in order to improve the comprehensibility of the study.

Further major changes to the manuscript include the restructuring of several sections, especially the introduction and Section 4.1 and adding more detailed descriptions of the representation of winter cereal (now partly in the Appendix), the new cover cropping and crop rotation routines (Sections 2.2.1 and 2.2.3), and a new figure on latent heat flux over a period of 3 year crop rotation. In addition, some short paragraphs have been added to the discussion and conclusions sections, e.g. on the applicability of the new routines presented in this study for large scale simulations.

Below we give detailed responses to each comment including citations from the revised manuscript.

Again, we thank the Reviewers and the Editor for their time and effort.

Sincerely,

Theresa Boas on behalf of all Co-authors

**Replies to comments by Anonymous Referee #1, 8 September 2020**

My main critique is that it is difficult to understand the impacts of (1) and (3) in particular. I tried to look into the code, but couldn't quite locate (1) and (3). I suggest the authors to make it clear in the code where these implementations are (perhaps mark it) so that I can follow how much the codes were changed relative to the standard. To gauge theimpacts of (1), I would like to see a winter wheat simulation only at the "DE-RuS" site for with and without (1). You could show how much leaf area index, latent heat and sensible heat of winter wheat changes with this assumption. Similarly, to examine the impact of (3), you could do a simulation of sugar beet and Winter wheat at "DE-RuS" (in this case sugar beet will be rotated, which you already did) and a simulation of Winter wheat only at "DE-RuS". You could also show how much leaf area index, latent heat and sensible heat of winter wheat changes if there was no rotation. Additionally, you can check whether rotation has any impacts on the modeled nitrogen leaching and fixation rates.

> Thank you very much for your constructive comments and suggestions. Please find our detailed replies below.
>
> We included a new Figure (Fig. 2) in section 4.1 highlighting the individual effect of the winter wheat subroutines (1) and the modified parameters for winter wheat (2), as well as a combination of both.
>
> (Line 439-449): "The impact of the new winter wheat specific parameterization and the new winter wheat routine, as well as the combination of both is illustrated in Fig. 2. Here we show simulated LAI for the default model and default parameter set (control), the default model with the new parameter set (control + crop specific), the extended winter wheat model with the default parameter set (new routines) and the extended winter wheat model with the new parameter set (new routines + crop specific).
>
> Using only the new crop specific parameter set with the default model configuration resulted in slightly higher LAI values compared to the control run but did not reach the observed maximum LAI values and the growth cycle duration. The implementation of the winter wheat subroutines using the default parameter set led to a more realistic reproduction of the growth cycle duration compared to the control run, but did not yield good correspondence with observed LAI magnitudes. The combination of the new crop specific parameter set and the new winter wheat subroutines resulted in the most realistic LAI dynamics (Fig. 2). "

[Figure]

> Figure 1: Daily simulation results for the LAI, simulated with default model and the default parameter set (control), the default model with new parameter set (control + crop specific), the extended winter wheat model with default parameterization (new routines) and the extended model with the new parameter set (new routines + crop specific), compared to point observations for a winter wheat year at DE-RuS.

**Replies to the list of specific comments by reviewer #1:**

(1) While I appreciate some of the details in sections 2.1, and 2.2.1, it would be appropriate it put most of the text in the Appendix section. For example, the paragraph that starts with the description of the default crop phenology scheme (lines 139 to 152 and additional lines) is not new to this study but rather standard CLM5 documentation notes and therefore, they can be put in the Appendix. Similarly, the section about Winter cereal representation that begins with "Vernalization" is also not new to this study. The default phenology scheme of CLM5 has a Vernalization subroutine.

> In the revised manuscript, we decided to keep a short description of the winter cereal subroutines and corresponding equations in the main text, for reasons of internal consistency of the paper and for comprehensibility to readers less familiar with specific CLM5 formulations. We added more detail to the

description by providing additional equations and explanations to the routine in the Appendix (Eq. A1-A9).

(2) The authors emphasize the importance of cash crops (e.g. sugar beets and potatoes). I would like the authors to comment on the spatial coverage of these crops in Germany and whether the famers are smallholder or largescale holder plantation owners. Along similar lines, it would be good if the authors could comment on how they plan to carry out the large scale simulations or regional simulations for these crops given that you need time series information about the rotation of these crops and also that some crops might be planted every two years or so.

> Thanks a lot for the suggestions. We added a paragraph on the local importance of winter wheat, sugar beet and potatoes to the introduction (line 86-92). Also, we added a comment on the applicability of our modifications for large scale simulations in Section 5 (line 717-729).

> (Line 86-92): "In Western Europe, a large proportion of arable land is cultivated with rotations of different non-perennial cash crops (Kollas et al., 2015; Eurostat, 2018). The most important cash crops grown in the European Union (EU) are cereals such as wheat (mostly winter wheat varieties in Western Europe), barley and maize, root crops such as sugar beet and potatoes, and oilseed crops such as rape, turnip rape, and sunflower (Eurostat, 2018). Cereals account for the majority of all crop production in the EU, contributing up to 12 % to global cereal grain production (Eurostat, 2018). The EU production of sugar beet accounts for about half of the global production (Eurostat, 2018)."

> (Line 717-729): "This new routine can be used to study cover cropping scenarios in future large-scale simulations. The effect of a cover crop during winter months on all crop land units where cash crops are grown in summer could be tested. This could also be tested for specific cash crops only. In addition, it is possible to simulate cover crop plantations based on harvest date thresholds. A defined maximum harvest date for any specific cash crop could define whether a cover crop such as winter wheat would be planted or not. For example, for all sugar beet land units with harvest dates before a certain threshold (e.g. day 290 of any given year) winter wheat could be planted as a cover crop during winter. If this harvest threshold were not reached and the summer crop is harvested late in the year, no cover crop would be planted. Alternatively, these harvest thresholds could define the type of cover crop, e.g. early harvest - winter wheat, late harvest – simple greening mix, etc. Also, historical land use information could be used to simulate realistic cover cropping and crop rotation scenarios. The succession of different crops from historical data could also be used to model the succession of crops for the future. In order to study large scale effects of cover cropping and common crop rotations, the CLM5 model would greatly benefit from further crop specific parameter sets for cover crops such as mustard, and further important cash crops."

(3) A number of statements in the results section is difficult to follow. For example, in lines 407 to 412, there is no reference to any figures. What is green leaf area index in line 408? Do you mean before maturity, during maturity or after maturity?

> Thanks for pointing this out. We rephrased and restructured certain parts in this section.

> (Line 472-480): "For the BE-Lon site, CLM_WW simulated peak LAI magnitudes are close to the observations. An exception is the year of 2015, where CLM_WW underestimated the unusually high LAI values observed in May and June, which ranged from 5.40 to 6.38 $m^2/m^2$. For BE-Lon, faster growth was simulated in the early growing stage of winter wheat, resulting in a more gradual increase in LAI compared to the other sites (Figure 3). This is related to higher air temperatures at BE-Lon early in the growing stage (especially in February) that enabled more simulated growth compared to the other sites.

> Overall, the LAI peak simulated with CLM_WW occurred about one month earlier than observed, suggesting that maturation was reached too early. This is also reflected in the simulated CLM_WW harvest dates that are approximately one month earlier than the recorded dates (Table 3)."

(4) I think the poor seasonal dynamics and low magnitude of the leaf area index in Figures 2-5 of CLM-D could also be related to the parameter values rather than the winter wheat subroutine that was introduced in this study. There are at least 3 parameter values that are considerably different compared to the default parameters of CLM ('gddmin','hybggg' and 'graincn'). For example, I see that the default gddmin is 50 in the default but 100 in the modified case (this study). Also hybgdd in the modified case is 30 more than the default. So couldn't these likely explain poor seasonal dynamics and low magnitude of the leaf area index in Figures 2-5 of CLM-D?

> We believe that we have answered this comment by including Figure 2 and a corresponding discussion in Section 4.1, where we have distinguished the effects of the modified parameter set and the new winter cereal representation, as well as a combination of both.

> Please see our first comment above and lines 439-449 in the revised manuscript.

(1) In line 70, Bilinois et al. (2015) is cited but I think the reference is missing.

Thanks a lot for pointing this out. This reference and corresponding paragraph were removed from the revised manuscript in the course of restructuring this section (according to suggestions from RC2).

(2) Please provide fractions of sand, silt and clay in Table 2, maybe up to 5 cm or 10 cm?

We added a table (Table A3) on textural fractions at our study sites to the appendix in the revised manuscript.

(Line 788-790): "Table A1: Textural fractions (sand, silt and clay percentages) for the ICOS and TERENO cropland study sites averaged for the upper soil layers (up to 50 cm) with corresponding reference.

| Site/ID | Sand [%] | Silt [%] | Clay [%] | Ref. |
|---|---|---|---|---|
| Selhausen/DE-RuS | 16.4 | 63.4 | 14.9 | Brogi et al. (2019) |
| Merzenhausen/DE-RuM | 16.4 * | 63.4 * | 14.9 * | - |
| Klingenberg/DE-Kli | 21.5 | 22.8 | 55.7 | Grünwald (personal communication, 2020) |
| Lonzée/BE-Lon | 5-10 | 68-77 | 18-22 | Moureaux et al. (2006) |

*adopted from the DE-RuS site"

(3) While I agree with the statement (line 289) that "CLM5 only permits land use changes at the beginning of every year", users can start a CLM5 simulation in any month the land use change actually happens in real life by performing a 'clear-cut' following spin-up, for example.

Yes, re-starting a simulation at any month is possible in order to change CFT. However, this would require a manual restart every time the CFT/PFT changes and does not allow a consecutive simulation with flexible land use changes. While this can be done for point cases, it is not feasible for regional scale cases where CFTs might change at different times and for different land units.

(4) At the "BE-Lon" site, the LAI curve of winter wheat from DOY= 0 to DOY = 100 seems to have a relatively gradual and smooth growth (Figure 2) while at sites "DERuS", "DE-RuM", "DE-Kli", the growth is relatively sudden and steep during the same period. I would like the authors to provide some explanations for this difference.

Thanks for pointing this out. We added a short statement on this.

(Line 474-477): "For BE-Lon, faster growth was simulated in the early growing stage of winter wheat, resulting in a more gradual increase in LAI compared to the other sites (Figure 3). This is related to higher air temperatures at BE-Lon early in the growing stage (especially in February) that enabled more simulated growth compared to the other sites."

(5) In lines 602 to 603, the authors claim that CLM5 does not represent timing of fertilizer. Please provide a citation for this?

Fertilization dynamics and annual fertilizer amounts depend on the crop functional types in CLM5. For all cropping units, mineral fertilizer application starts during the leaf emergence phase of crop growth and continues for 20 days, so there is not much flexibility to represent different fertilization practices (e.g. timing, multiple applications, type of fertilizer etc.).

(Line 142-146): "In CLM5, land fractions with natural vegetation are not influenced by fertilizer application. In cropping units, mineral fertilizer application starts during the leaf emergence phase of crop growth and continues for 20 days. Manure nitrogen is applied at slower rates (0.002 kg N m$^{-2}$ per year by default) to prevent rapid denitrification rates that were observed in earlier CLM versions so that more uptake by the plant is achieved (Lawrence et al., 2018)."

(6) In line 603, the authors state that CLM5 does not consider varieties of winter wheat. I agree with this statement but at the same time, many land surface models don't consider varieties or cultivars of crops. Crops can be genetically modified to boost productivity. This means there could large differences in the parameter estimates among varieties/cultivars. The authors could discuss the variation in the parameter estimates if they are measured at their sites.

> Thanks for pointing this out. We added a short discussion on the representation of crop varieties in CLM5 in section 5.
>
> (Line 670-676): "In order to include different varieties of any crop, the list of CTFs could be extended with suitable plant parameterizations. However, this information is not readily available, due to combination of measurement data scarcity and the complexity of the phenology algorithm and parameter scheme. The introduction of a phenology scheme based on plant physiological trait information in CLM could be a major improvement in this field (see Fisher et al., 2019), as plant trait information becomes more readily available (e.g. TRY Plant Trait Database, Kattge et al., 2011). Whether considering different varieties and cultivars of a crop is important for regional or global scale simulations remains to be evaluated."

(7) The authors mention in lines 626 to 629 the following: "There is a tool available for CLM5 that enables the simulation of transient land use and land cover changes (LULCC) (Lawrence et al., 2018). It was designed to simulate and study the effects of changing distributions of natural and crop vegetation, e.g. land use change from forest to agricultural fields (Lawrence et al., 2018), rather than inter-annual changes of agricultural management on crop vegetated areas." I'm confused about the last part "rather than inter-annual changes of agricultural management on crop vegetated areas". Please explain what do you mean by this? Do you mean you cannot change the Nitrogen fertilization rate from year to year in CLM5?

> With this we wanted to emphasize that although this tool allows changes in land use every year (on 1$^{st}$ of January), it does not account for changes happening during the year (e.g. several crop growth cycles or changes from summer to winter crop in fall) or multiple crop growth cycles within one year (e.g. multiple growth cycles of the same cash crop within one year in India due to several monsoon seasons). The annual amount of mineral nitrogen fertilization is assigned by plant/crop functional type and can be changed manually for each year.
>
> We rephrased our discussion on this tool accordingly in the revised manuscript.
>
> (Line 701 -709): "There is a tool available for CLM5 that enables the simulation of transient land use and land cover changes (LULCC) (Lawrence et al., 2018). It was designed to simulate the effects of changing distributions of natural and crop vegetation, e.g. land use change from forest to agricultural fields and also allows for changes in crop type between years (Lawrence et al., 2018), but does not account for intra-annual changes of agricultural management on crop vegetated areas that happen in double and triple cropping scenarios. While this tool is useful to study general land use changes by changing the land cover type of individual land units, we found it lacks flexibility in accounting for changes within land units of the same land cover and does not account for all 64 CFTs. Furthermore, this tool changes the CFT of each column on the 1$^{st}$ of January every year according to prescribed values (customized)."

**Replies to comments by Anonymous Referee #2, 17 September 2020**

Overall: Boas et al. have done considerable work to modify and evaluate simulations of European agriculture in CLM. This work is very exciting and shows significant improvement in model parameterizations and capabilities to add cover crop and crop rotation management practices.

Despite the importance of this work, the text needs to be revised before it is suitable for publication. Some sections require reorganization for clarity, while others will benefit from streamlining to remove redundancies or adding necessary detail. Comments highlighting these sections are included below, as well as specific line comments.

> Thank you very much for your constructive feedback and detailed suggestions. Please find our detailed replies to the review comments below.

**Abstract**.

**Line 25**: Is cover cropping only common in humid and sub-humid regions? Perhaps it would be more informative to rephrase to something similar to: ".which is an agricultural management technique commonly used in the regions evaluated in this study." Alternatively, you can say that it is a technique growing in popularity to improve soil health and carbon storage.

> Thanks, we incorporated your suggestion in the revised manuscript.

> (Line 23-26): "The latter modification allows the simulation of cropping during winter months before usual cash crop planting begins in spring, which is an agricultural management technique with a long history that is regaining popularity to reduce erosion and improve soil health and carbon storage and is commonly used in the regions evaluated in this study."

**Line 26:** Are you referring to the parameterization of new CFTs? Please clarify.

> We have clarified this as follows:

> (Line 27-31): "We compared simulation results with field data and found that both the new crop specific parameterization, as well as the winter wheat subroutines, led to a significant simulation improvement in terms of energy fluxes (RMSE reduction for latent and sensible heat by up to 57 % and 59 %, respectively), leaf area index (LAI), net ecosystem exchange and crop yield (up to 87 % improvement in winter wheat yield prediction) compared with default model results."

**Line 27:** Please move the reference of RSME for LH and SH to just after the energy fluxes rather than after NEE.

> Thanks, we changed the text accordingly. Please see our response above and line 26-30 in the revised manuscript.

**Line 31:** When you refer to the "LAI curve", is this the same as the season cycle of LAI? If so, please modify the wording to reflect this.

> We rephrased accordingly.

> (Line 31-33): "The cover cropping subroutine yielded a substantial improvement in representation of field conditions after harvest of the main cash crop (winter season) in terms of LAI magnitudes and seasonal cycle of LAI, and latent heat flux (reduction of winter time RMSE for latent heat flux by 42 %)."

**Lines 31-33:** It would be more impactful if you strengthened the last sentence in the abstract. Here is one suggestion: "Our modifications significantly improve model simulations and should therefore be used in future simulations to better understand large-scale impacts of agricultural management on carbon, water, and energy fluxes."

> Thanks for your suggestion, we added this accordingly.

> (Line 34-36): "Our modifications significantly improved model simulations and should therefore be applied in future studies with CLM5 to improve regional yield predictions and to better understand large-scale impacts of agricultural management on carbon, water and energy fluxes."

**Introduction**: Overall, the introduction needs some reorganization. You need to more clearly highlight the role of management (make this a separate paragraph, include cover crops but also other types of management). The new representation of cover crops is a primary contribution to this paper and is barely mentioned here. The introduction also needs a broader overview of crops in LSMs (it currently only focuses on AgroIBIS and CLM). Last, most of the introduction emphasizes the global nature of models and that the variation in soils, plants, climate is important. When the reader finally gets to the end of the introduction, which highlights that you focus on a few sites in Europe (which some may argue has narrower variation in soils, plants, climate than if you were to compare to locations from other continents), it makes this study seem limited. It might help to instead describe that models are still

limited by their ability to represent many crop types and important management practices, emphasizing the importance of your work adding these new capabilities, and also to highlight that Europe is a major agricultural hub for global food production.

> Thank you for your constructive suggestions on the introduction.
>
> Major restructuring was done for this section according to the review comments. Most of the paragraphs were kept and reorganized in the revised manuscript. We added a short overview of crop modules and recent developments in LSMs other than CLM. We commented the regional importance of the simulated cash crops (based on EU statistics) and added more detail to the importance of management practices such as cover cropping and crop rotation. Furthermore, several new references are cited in this section.
>
> Please see Section 1 in the revised manuscript from line 38-110.

**Lines 44-49:** The mention of cover crops here seems a bit out of place. The earlier part of this paragraph and the start of the next paragraph is focused on adaptation to climate change, whereas the description of cover crops here focuses on soil benefits and climate mitigation. I suggest reorganizing, moving the cover crop description to later in the introduction.

> Thanks for your suggestions. We restructured the text accordingly.
>
> (Line 92-100): "The use of cover crops is a common agricultural management practice to reduce soil erosion, soil compaction, and nitrogen leaching as well as to increase agricultural productivity by nitrogen fixation (Sainju et al., 2003; Lobell et al., 2006; Basche et al., 2014; Plaza-Bonilla et al., 2015; Tiemann et al., 2015; Kaye and Quemada, 2017). The biogeochemical effects and benefits of cover crops as well as their potential to mitigate climate change are the focus of many studies (e.g. Sainju et al., 2003; Lobell et al., 2006; Groff, 2015; Plaza-Bonilla et al., 2015; Basche et al., 2016; Carrer et al., 2018; Lombardozzi et al., 2018; Hunter et al., 2019). Despite recent development efforts, the representation of these management practices has not yet been included in CLM5. Furthermore, in a previous study by Lu et al. (2017) the default representation of winter cereals performed poorly in simulating the phenology of winter wheat."

**Lines 68-70**: I'm not sure I entirely understand the point of this sentence. Is this just to highlight the evaluation of crops in CLM4.5?

> We agree and removed this sentence from the text in the revised manuscript.

**Lines 73-76:** You should also reference the CLM5 crop overview paper here, which evaluates global crop yields: Lombardozzi, D. L., Y. Lu, P. J. Lawrence, D. M. Lawrence, S. Swenson, K. W. Oleson, W. R. Wieder, and E. A. Ainsworth (2020), Simulating Agriculture in the Community Land Model Version 5, J Geophys Res-Biogeo, 125(8), 927–19, doi:10.1029/2019JG005529.

> Thanks for pointing this out, we added Lombardozzi et al. (2020) to our list of references.

**Lines 77-84:** This paragraph seems too detailed for the introduction. I suggest summarizing and merging with the previous paragraph. For example: "The few studies that have evaluated CLM5 suggest inaccurate phenology and overestimated crop yields (Sheng et al. 2018)." However, you'll probably want to change/update this to also incorporate results from the Lombardozzi et al. CLM5 paper mentioned above.

> Thanks a lot for your suggestions on the introduction. We restructured this section and added the study of Lombardozzi et al. (2020) to the references in the revised manuscript.
>
> Please see section 1 in the revised manuscript from line 38-110.

**Methods Overall**: The methods section needs to be tidied up. There are redundancies in the first section, and a lack of detail in the cover crop description. Please pay careful attention to providing enough detail that the reader isn't left wondering how something was done, but keep the text succinct. You reference Lawrence et al. 2018 in several places throughout the text. However, I believe this paper was published in 2019 (not 2018). Please double-check.

> Thank you for your constructive suggestions on this section.
>
> We restructured several parts of this section and added Lawrence et al. (2019) to our references. We provided more detail to the winter cereal representation in the appendix, Section 7.1, equations A1-A9.
>
> Additionally, we extended the description of the cover cropping and crop rotation routine in Section 2.2.3.
>
> (Line 303-324): "Individual crop rotation schemes were customized within the code and depend on the currently planted crop type. For example, if a simulation starts with a crop coverage of spring wheat specified in the surface file, the new subroutine is called after harvest of the crop. Within the subroutine,

the CFT is then changed to the next crop, e.g. sugar beet. Again, after the harvest of this crop, e.g. sugar beet, the CFT is again changed to the next crop and so on. When the CFT is changed back to spring wheat, the rotation cycle starts again. This rotation is defined in a repetitive sequence based on the harvested CFT and its harvest date:

if $harvdate(p) \geq hd_1$ and $ivt(p) = crop_1$ then

$ivt(p) = crop_2$

$croplive(p) = false$

$idop(p) = not\_planted$

$use\_grainproduct = true$

else if $harvdate(p) \geq hd_2$ and $ivt(p) = crop_2$ then

$ivt(p) = crop_3$

$croplive(p) = false$

$idop(p) = not\_planted$

$use\_grainproduct = true$ (7)

where harvdate is the harvest day of the current simulation year and hd is the customizable harvest date of the respective CFT, $p$ is the simulated patch on the model grid, ivt is the simulated CFT, $crop_{1-3}$ represent the user-specified CFTs to the rotated, idop is the planting day and use_grainproduct is a flag to define whether the grain carbon of simulated crop is to be harvested into the food pool or not. If this flag is set to false, the plant carbon and nitrogen are transferred to the soil litter pool and not allocated to the food product pool upon harvest of the crop."

**Section 2.1:**
When describing the crop model, please also cite Lombardozzi et al. (2020), as this has much more detailed information about the crop model updates than Lawrence et al. 2019.
The methods should be streamlined to avoid repetition. For example, allocation is mentioned in lines 134-138, and then again in the paragraph starting at line 153. When referring to C allocation, you state that it varies throughout the growing season (e.g., line 156), whereas the reference to N allocation states that it uses two different C/N ratios (lines 161-162). However, these are treated the same way in the model. Please update for consistency. I suggest switching the order of Eq. 1 and Eq. 2.

Thanks, we revised the text and included a reference to Lombardozzi et al. (2020).

(Line 162-167): "The allocation of carbon and nitrogen also follows the phenology phases. During the leaf emergence phase, carbon from the seed carbon pool is transferred to the leaf carbon pool. Nitrogen is supplied through the soil mineral nitrogen pool. During the grain fill phases, nitrogen from the leaf and stem of the plant is translocated to the grain pool. Allocation ends upon harvest of the crop where grain carbon and nitrogen are transferred from the grain pool to the grain product pool and, a small amount of $3g\,C\,m^{-2}$, to the seed carbon pool for the next planting (Lawrence et al., 2018; Lombardozzi et al., 2020)."

**Line 114:** Please define "CFT" the first time you use this term.

Thanks, we corrected this.

**Line 115:** land units are not separated by fertilizer, only by irrigation. Please update.

We agree that the phrasing is misleading and updates it accordingly:

(Line 128-130): "Vegetated land units are separated into natural vegetation and crop land units, with only one crop functional type (CFT) on each soil column, including irrigation as a CFT specific land management technique ( Lawrence et al., 2018; Lombardozzi et al., 2020)."

**Lines 204-206:** This is a bit confusing and could use clarification. Does the vernalization factor always range from 0-1? Is it applied to GDD for air and soil temperatures (e.g., does it affect all phenological phases)? If it is only applied to grain C allocation, where does the remaining C get allocated?

Yes, the vernalization factor ranges from 0 to 1 (fully vernalized) and affects the GDD in the phenology phase after planting (vernalization starts after leaf emergence and ends before flowering). This leads to a reduced growth when the plant is not fully vernalized and the vf is smaller than 1:

For vf <1

$GDD_o * vf = GDD_n$ with $GDD_n < GDD_o$

where the subscripts $_o$ and $_n$ stand for original and updated GDD.

We added more details to the description of winter cereal representation in the appendix.

(Line 214-216): "The vernalization factor can range between 0 (not vernalized) and 1 (fully vernalized). It is multiplied with the GDD during the phenology phase after planting and the grain carbon allocation coefficient which leads to a reduced growth rate in the beginning of the phenology cycle until the plant is fully vernalized."

**Section 2.2.1**
It would be helpful to start with an overview of how winter cereal representation differs from other crops. I suggest a high level overview of why it's important to include both vernalization and cold tolerance before diving into the details of each.

Thanks for your suggestions. We added a more general paragraph on winter wheat and the two processes – vernalization and cold tolerance.

(Line 182-193): "Winter wheat is an important crop for global food production and covers a significant fraction of the European croplands. (Chakraborty and Newton, 2011; Vermeulen et al., 2012). In general, winter wheat is exposed to a different range of environmental stresses compared to summer crops such low temperatures. In regions with sufficiently cold winters, the main processes that allow a successful cultivation of winter wheat during the colder months are vernalization and cold tolerance (Barlow et al., 2015; Chouard, 1960). Vernalization represents the process that an exposure to a period of non-lethal low temperatures is required to enter the flowering stage for winter crops. In general, the vernalization process ensures that the reproductive development of plants growing over winter (winter crops and also natural vegetation) does not start in late summer or fall but rather in late winter or spring. The other process, cold tolerance, ensures that the crop can acclimate to low temperatures and thus survive cold temperatures and even freeze-thaw cycles. However, cold damage to the crop can occur when the crop is exposed to low temperatures at a certain development stage. These damages have been documented to have significant impacts in crop yield (Lu et al., 2017)."

**Equation 4**: You specify that Tcrown is slightly higher than the freezing temperature when covered by snow. I see that snow height is used in the calculation, but where is the plant height? Without including the plant height, how do you know whether the plant is covered by snow?

(Line 201-204): "The vernalization process starts after leaf emergence and ends before flowering (Streck et al., 2003) and is dependent on the crown temperature ($T_{crown}$) (see Eq. A1). The crown is the connecting tissue between the roots and the shoots at the base of the plant. For winter wheat, the crown node is located at about 3 – 5 cm soil depth (Aase and Siddoway, 1979)."

**Line 213:** The text describes what the accumulative parameters are, but what about the previous time step is used? It would also be useful to include a brief description of how some of the accumulative parameters accumulate (e.g., are these all based on some aspect of accumulated temperature?)

(Line 801-823): "The temperature at which 50 % of the plant is damaged ($LT_{50}$) is calculated interactively at each time step ($LT_{50t}$) depending on the previous time step ($LT_{50t-1}$) and on several accumulative parameters. These parameters are the exposure to near-lethal temperatures ($rate_s$), the stress due to respiration under snow ($rate_r$), the cold hardening or low temperature acclimation (contribution of hardening – $rate_h$) and the loss of hardening due to the exposure to a period of higher temperatures (dehardening – $rate_d$) that are each functions of the crown temperature (Lu et al., 2017 and references therein):

$$LT_{50t} = LT_{50t-1} - rate_h + rate_d + rate_s + rate_r \qquad (A3)$$

The exposure to near-lethal temperatures is based on the winter survival model after (Fowler et al., 1999) and is calculated as follows:

$$rate_s = \frac{LT_{50t-1} - T_{crown}}{e^{-1.9(LT_{50t-1} - T_{crown}) - 3.74}} \qquad (A4)$$

The stress due to respiration under snow is calculated as a function of snow depth (dsnow) that ranges from 0 to 1 for snow cover up to 12.5 cm (equal to 1 for all snow depth higher than 12.5), and a specific respiration factor (RE):

$$rate_r = R \times RE \times f(dsnow)$$

$$R = 0.54 \ f(dsnow) = min(dsnow, 12.5) / 12.5$$

$$RE = \frac{e^{0.84+0.051\,T_{crown}-2}}{1.85} \qquad (A5)$$

The contribution of hardening and dehardening are calculated within certain temperature ranges as follows:

For $T_{crown} < 10°C$

$$rate_h = 0.0093(10 - \max(T_{crown}, 0))(LT_{50t-1} - LT_{50c}) \qquad (A6)$$

For $T_{crown} \geq 10°C$ when vf < 1 (not fully vernalized), and $T_{crown} \geq -4°C$ when vf =1 (fully vernalized)

$$rate_d = 2.7 \times 10^{-5}(LT_{50i} - LT_{50t-1})(T_{crown} + 4)^3 \qquad (A7)$$

where $LT_{50c}$ is the maximum frost tolerance of -23 °C and $LT_{50i}$ represents the $LT_{50}$ for an unacclimated plant ($LT_{50i}$ = -0.6+0.142 $LT_{50c}$)."

**Equation 6:** Please define the "alpha-surv" and the "t" variables in this equation.

Thanks, we corrected this.

(Line 824-828): "The survival rate ($f_{surv}$) is then calculated as a function of $LT_{50}$ and the crown temperature. The probability of survival is a function of $T_{crown}$ in time (t). It increases once $T_{crown}$ is higher than $LT_{50}$ or decreases when it is lower (Vico et al., 2014):

$$f_{surv}(T_{crown}, t) = 2^{-\frac{T_{crown}^{\alpha_{surv}}}{LT_{50}}} \qquad (12)$$

where $\alpha_{surv}$ is a shape parameter of 4."

**Equation 7:** I am confused by this, partly because it's not clear what the equation is taking the max of. Also, can Tcrown be negative? That seems to be the only way the solution to this equation isn't 0. Please update to clarify. Also, I think 'fsurf' should in fact be 'fsurv'.

Thanks, we corrected this.

$T_{crown}$ is assumed to be the same as the 2-m air temperature without snow cover and thus can be negative.

(Line 792-800): "The temperature at the crown of the plant ($T_{crown}$) is assumed to be slightly higher than the 2-m air temperature ($T_{2m}$) in winter when covered by snow, and the same as the 2-m air temperature without snow cover. Within CLM5, it is calculated separately for temperatures below and above the freezing temperature ($T_{frz}$):

$$T_{crown} = 2 + (T_{2m} - T_{frz}) * (0.4 + 0.0018 * (\min(D_{snow} * 100, 15) - 15)^2$$

for $T_{2m} < T_{frz}$ $\qquad (A1)$

$$T_{crown} = T_{2m} - T_{frz}$$

for $T_{2m} > T_{frz}$ $\qquad (A2)$

where $T_{crown}$ [K] is the calculated crown temperature, $T_{2m}$ [K] is the 2-m air temperature, $T_{frz}$ [K] is the freezing point and $D_{snow}$ [m] is the snow height."

**Paragraph starting at line 227:** I find the description here a little confusing. Can you revise this to more clearly articulate the difference between survival probability and WDD? Is survival probability just a step function, where any value <1 causes the same amount of damage (simulated as part of the C and N pools being transferred to litter)? Should I be thinking of survival probability as the proportion of the plant that survives, or the probability that the whole plant survives? Also, part of my confusion is that this is the first place that a frost damage function is mentioned.

Thanks for your suggestions on this section. We added several more equations and explanations to the appendix. Please see line 789-832 in the revised paper. For a more detailed description we refer to the source literature by Lu et al. (2017) and references therein.

The survival probability is used to calculate the WDD. During the early growing season when the plant is not fully vernalized (vf < 1) and is exposed to subzero temperatures (negative $T_{crown}$), the survival probability will be low and thus the WDD will be high. It is furthermore used in the subsequent steps to estimate frost damage to the crop. We included two more equations on frost damage in the revised manuscript (Eq. A10 and A11).

(Line 824-856): "The survival rate ($f_{surv}$) is then calculated as a function of $LT_{50}$ and the crown temperature. The probability of survival is a function of $T_{crown}$ in time (t). It increases once $T_{crown}$ is higher than $LT_{50}$ or decreases when it is lower (Vico et al., 2014):

$$f_{surv}(T_{crown}, t) = 2^{-\frac{T_{crown}^{\alpha_{surv}}}{LT_{50}}} \tag{A8}$$

where $\alpha_{surv}$ is a shape parameter of 4.

The winter killing degree day (WDD) is calculated as a function of crown temperature and survival probability, where the maximum function limits the integration to the potentially damaging periods, when the air temperature (T) is lower than the base temperature ($T_{base}$) of 0°C (Vico et al., 2014):

$$WDD = \int_{winter} max[(T_{base} - T_{crown}), 0] \, [1 - f_{surv}(T_{crown}, t)]dt \tag{A9}$$

Lower $LT_{50}$ indicate a higher frost tolerance and would result in higher survival rates, smaller WDD and less cold damage to the plant. Thus, when the survival probability and crown temperature are low, the WDD will be high (Vico et al., 2014).

Lower $LT_{50}$ indicate a higher frost tolerance and would result in higher survival rates, smaller WDD and less cold damage to the plant. Thus, when the survival probability and crown temperature are low, the WDD will be high (Vico et al., 2014).

The survival probability and the WDD are then used to estimate instant and accumulated frost damage to the crop during the leaf emergence phase (Lu et al., 2017). Instant frost damage is assumed to happen at the beginning of the growing season when the plants are not fully vernalized (vf < 0.9) when the growth of leaves (especially new leaves or small seedlings) due to an exposure to low temperatures. It is simulated by reducing the leaf carbon at low survival probabilities (whenever $f_{surv}$ is below 1). The leaf carbon is reduced by an amount of 5 gC m$^{-2}$ scaled by a factor of 1- $f_{surv}$ that is moved to the carbon litter pool, up to a minimum value of 10 gC m$^{-2}$ leaf carbon:

$$leafc_t = leafc_{t-1} - leafc_{damage}(1 - f_{surv})$$

for vf < 0.9, WDD > 0, $f_{surv}$ < 1, and $leafc_t$ > 10 $\qquad$ (A10)

where $leafc_t$ is the simulated leaf carbon of the current time step, $leafc_{t-1}$ is the leaf carbon of the previous step and $leafc_{damage}$ is equivalent to 5 gC m$^{-2}$.

When the plant is close to vernalization towards the end of the leaf emergence phase, it is not as susceptible to suffer from instantaneous frost damage as in the beginning of this phase. Still, an extended period of freezing temperatures can potentially induce damage to the plant (Lu et al., 2017). This accumulated frost damage is simulated based on the accumulated WDD and average survival probability. When the accumulated WDD reaches a value higher than 1° days, the leaf carbon from the previous time step ($leafc_{t-1}$), scaled by the average $f_{surv}$, is moved to the soil carbon litter pool:

$$leafc_t = leafc_{t-1}(1 - \text{average } f_{surv})$$

for vf $\geq$ 0.9 and WDD > 1 $\qquad$ (A11)

Once this has occurred, the accumulated WDD is reset to 0 and the tracking of the average $f_{surv}$ is restated. Corresponding to the leaf carbon reduction, the leaf nitrogen is reduced from the leaf nitrogen pool to the soil nitrogen litter pool scaled with the parameterized leaf C/N ratio for winter wheat of 20."

**Section 2.2.2:**

Since you use a pre-existing winter wheat parameterization, it would be helpful to include some information about what you changed in the parameterization and why.

Thanks for pointing this out, we added some more explanations to this part of the text.

(Line 256-263): "In order to yield a reasonable representation of agricultural areas on the regional scale in future studies, the default parameter set was extended with specific crop parameters for sugar beet, potatoes, and winter wheat based on the characteristics of our study sites to better fit the observed plant phenology and energy fluxes at the simulation sites.

The CTFs sugar beet and potatoes are merged to the spring wheat CFT on the default parameter scheme due to the lack of crop specific parameters for these crops. For winter wheat there is a pre-existing default parameter set available in CLM5. However, this default parameterization performed poorly in representing the crop phenology for the evaluated study sites in this study. This was also reported in an earlier study by Lu et al. (2017). Thus, crop specific parameters were added for sugar beet, potatoes and winter wheat."

**Table 1:** This is a useful summary, but I'm not sure it adds much information to the main text.

We believe that this overview table is helpful information for the reader. Thus, we kept it in the revised manuscript.

**Section 2.2.3:** How do you determine when the cover crops (or rotations) are planted and the subsequent phenology phases? Is it based on GDD? Did you have to modify GDD parameters or add new ones? Did you add new CFTs to accomplish this? How is allocation determined? This section needs more detail about how modifications were made, as it is the bulk of the development work in this paper.

The rotation schemes are hardcoded in the new cover cropping subroutine. Basically, in the new routine, the phenology algorithm is reset and restarted after harvest of any crop that is assigned with the cover crop flag. We are currently working on a version to make the application more user-friendly, e.g. rotation defined by a control file.

We added more detail to this section in the revised manuscript.

(Line 303-332): "Individual crop rotation schemes were customized within the code and depend on the currently planted crop type. For example, if a simulation starts with a crop coverage of spring wheat specified in the surface file, the new subroutine is called after harvest of the crop. Within the subroutine, the CFT is then changed to the next crop, e.g. sugar beet. Again, after the harvest of this crop, e.g. sugar beet, the CFT is again changed to the next crop and so on. When the CFT is changed back to spring wheat, the rotation cycle starts again. This rotation is defined in a repetitive sequence based on the harvested CFT and its harvest date:

if $\mathrm{harvdate}(p) \geq \mathrm{hd}_1$ and $\mathrm{ivt}(p) = \mathrm{crop}_1$ then

$\mathrm{ivt}(p) = \mathrm{crop}_2$

$\mathrm{croplive}(p) = \mathrm{false}$

$\mathrm{idop}(p) = \mathrm{not\_planted}$

$\mathrm{use\_grainproduct} = \mathrm{true}$

else if $\mathrm{harvdate}(p) \geq \mathrm{hd}_2$ and $\mathrm{ivt}(p) = \mathrm{crop}_2$ then

$\mathrm{ivt}(p) = \mathrm{crop}_3$

$\mathrm{croplive}(p) = \mathrm{false}$

$\mathrm{idop}(p) = \mathrm{not\_planted}$

$\mathrm{use\_grainproduct} = \mathrm{true}$ $\hspace{4cm}$ (7)

where harvdate is the harvest day of the current simulation year and hd is the customizable harvest date of the respective CFT, $p$ is the simulated patch on the model grid, ivt is the simulated CFT, $\mathrm{crop}_{1\text{-}3}$ represent the user-specified CFTs to the rotated, idop is the planting day and use_grainproduct is a flag to define whether the grain carbon of simulated crop is to be harvested into the food pool or not. If this flag is set to false, the plant carbon and nitrogen are transferred to the soil litter pool and not allocated to the food product pool upon harvest of the crop.

The actual rotation of crop types can be user-customized by defining the variables hd and $\mathrm{crop}_x$ in a list (e.g. $\mathrm{hd}_1 = 150$ [day of year], $\mathrm{crop}_1 =$ spring wheat, etc.). By including the harvest date as a dependency, it is also possible to simulate the planting of cover crops based on harvest date thresholds. A user-defined maximum harvest date for any specific cash crop can define whether a cover crop would be planted or not. This technique can be beneficial to study the effects of conceptual cover cropping scenarios on regional scales. The possibility to change the CFT within the same year represents a significant improvement in flexibility, as CLM5 only permitted land use changes at the beginning of every year. In order to simulate cover cropping at our study site DE-RuS, we implemented a new CFT for a greening mix cover crop (or $\mathrm{covercrop}_1$)."

**Lines 267-270:** It's great to hear that you introduced a flag to use the cover crop option, but I'm not sure you need to include that description here.

We believe it is important to inform the reader and potential CLM5 user about the new cover cropping flag.

**Lines 276-277:** How are you predefining a rotation scheme?

At this stage this is hardcoded in the new subroutine. Please see the revised section 2.2.3 in the manuscript and our response to the comments above.

**Line 283:** "catch crop" this is the first time you mention it. Are you using this interchangeably with cover crop (which is how you described this in the previous sentence), or are you using a new phrase to distinguish this from cover crop? Please be clear and consistent with word choices.

Here the terms 'cover crop' and 'catch crop' were used synonymously. We corrected this by using only 'cover crop' in the revised manuscript for consistency.

**Line 283:** You mention plowing the crop into the soil. However, CLM does not represent plowing. How did you accomplish this. Do you assume that the plant biomass is transferred to the litter pool? Also, how did you decide when this happens?

(Line 300-302): "A common practice is to plough the cover crops into the soil instead of removing their biomass from the field. We simulated this by relocating the biomass of the crop into the litter pool instead of the grain product pool upon harvest using the use_grainproduct flag described below (Eq. 7)."

**Section 2.3:**
I think it would help to describe the sites before the validation data, and/or mention whether you run CLM simulations at these sites. This section starts by describing validation data, but does not mention what is being validated.

We restructured the text accordingly.

(Line 338-339): "The CLM5 model was set up for four European cropland sites: Selhausen, Merzenhausen, Klingenberg and Lonzée (Fig. 1). These sites were selected mainly for their excellent continuous measurements of surface energy fluxes."

**Table 2:** Useful information about the sites, but I think the map describes the locations quite well, and most of the other information included in the table is not used in the simulation. Therefore, I'm not sure that this table is necessary in the main text.

We think this table gives the reader a nice overview without having to read this section in detail and therefore would like to keep it in the main text. We will include an additional table with textural fractions at the study sites in the appendix of the revised paper as requested by RC1.

**Lines 318-319**: You mention winter wheat twice here.

Thanks, we correct this.

**Lines 341-342:** CLM's default time step is 30 minutes.

Here we mean the customized time step of input forcing data, which was set to hourly. Not all meteorological input data was available half-hourly, thus an hourly temporal resolution was used. The internal model time step remains at 30 minutes.

**Section 3.1:** Throughout this section, the differences in model version versus parameter set seem to be conflated. Please make this much clearer throughout, explaining what each of the model versions includes and what the default versus modified parameter sets include.

Thanks a lot for your suggestion on this section. Changing the wording for our simulation scenarios as well as their description in Section 2.3 as suggested by the reviewer helped us to significantly improve the comprehensibility of our manuscript in this regard.

(Line 403-425): "In order to test the winter wheat representation, several simulations were conducted for all winter wheat years at the sites DE-RuS, DE-RuM, DE-Kli and BE-Lon. In a first step, the impact of each modification was assessed individually by simulating one winter wheat year at the site DE-RuS using four different model configurations: (1) the default model and default parameter set (control), (2) the default model with the new parameter set (control + crop specific), (3) the extended winter wheat model with the default parameter set (new routine), and (4) the extended winter wheat model with the new parameter set (new routine + crop specific). Further evaluations for the other study sites and years were conducted for the combined winter wheat modifications CLM_WW (extended model with winter wheat subroutines and new crop specific parameterization) in comparison to control simulations (default model configuration and default parameterization of winter wheat).

For the evaluation of the crop specific parameter sets for sugar beet and potatoes, simulations were run with the new parameterizations at the sites DE-RuS and BE-Lon over several years. For both sites, control

simulations were conducted without the new parameter set, in which both CFTs sugar beet and potatoes are simulated as a spring wheat by default. Furthermore, an evaluation of the default parameterization for the CFT temperate corn at the site DE-Kli is included in the supplementary material (Fig. S1, Table S1).

The cover cropping and crop rotation scheme was tested for two practical cases at DE-RuS. From 2016 to 2017, planting was altered at DE-RuS from barley (here represented by the CFT for spring wheat) in 2016 to sugar beet in 2017 with a greening mix cover crop in between (winter months 2016/2017). In order to simulate this common cover cropping practice, we implemented a new CFT for a greening mix cover crop (or covercrop$_1$). For the years 2017 to 2019 at DE-RuS, the subroutines ability to simulate realistic crop rotation cycles was tested by changing the simulated CFT from sugar beet (2017) to winter wheat (2017-2018) and then to potatoes (2019). In this step, simulations were run with the previously tested crop specific parameterizations for sugar beet, potatoes and winter wheat. Simulation results were again compared to a control simulation run, where a consecutive growth of spring wheat is simulated."

**Table 3:** Which simulations include the potato and sugar beet parameterization? It looks like it's the CLM_WW simulation, but this needs to be explicitly mentioned in the table description.

We removed the table from the revised manuscript and instead included a more detailed description of the conducted simulation experiments. Please see line 432-457 in the revised manuscript.

**Lines 364-366:** This text is confusing: It is not clear what the difference is between the default model and the modified model. I assumed the "default" model did not include winter wheat, but this text suggests that it does. How, then, is the default model run with the modified winter wheat parameters different from the winter wheat model with the modified parameters?

The CFT of winter wheat is included in the default model but its specific parameter set yielded very poor representation of simulated winter wheat phenology at our sites and also in previous studies. Thus, next to the implementation of vernalization and cold tolerance representation in the model code, new crop specific parameters were supplied in order to optimize the model performance. Please see our response above to the comment on section 3.1.

**Lines 369-370:** What are the default parameterizations of sugar been and potatoes?
These aren't included in CLM, so is there a "default"?

Sugar beet and potatoes are included in the structure of the CLM5 crop module and are amongst the 64 CFTs. The CFTs sugar beet and potatoes do not have assigned parameters specifically calibrated for these crops, instead the same parameters as for spring wheat are set as default for these CFTs. We changed the terminology from 'CLM_D' to 'control' throughout the text for better comprehensibility.

(Line 413-415): ". For both sites, control simulations were conducted without the new parameter set, in which both CFTs sugar beet and potatoes are simulated as a spring wheat by default. "

**Section 4**
In general, I find the use of CLM_D, CLM_WW, and CLM_WW_CC to be confusing, as the changes included in each are not clearly described. Additionally, it seems that sugar beet and potato parameterizations are added to CLM_WW. It might be more helpful to instead refer to CLM_D as "control" or "default" and then refer to updated parameterization (e.g., "improvements to winter wheat" rather than "CLM_WW" in Section 4.1 and "new potato" or "new sugar beet" parameterization in Section 4.2). Additionally, throughout this section, figures should include estimates of uncertainty.

We appreciate your suggestions and incorporated them in the revised manuscript by changing the terminology throughout the text and on figures and tables. Please also see our responses to the reviewer comments on Section 2.3 above.

Due to the small number of compared years (2 to max. 6 years), uncertainty estimates do not add much value to the plots. As briefly discussed in Section 4.1, CLM did not capture inter-annual differences in yield well, showing only minor variations between simulated years. This is also reflected in corresponding simulated LAI curves and energy fluxes that differ only insignificantly from year to year.

(Line 498-501): ". The simulated yields by CLM_WW for the individual years show only minimal variations with values from 8.12 to 8.16 t/ha, while the measured yields ranged from 9.92 to 12.88 t/ha, indicating that CLM did not capture the inter-annual yield variation very well (Table 3)."

**Section 4.1:** Throughout this section, the text could be streamlined to avoid repeating the description of trends for each site (see note below about Figs. 2-5). Additionally, the trends in energy fluxes are barely mentioned, leaving the reader wondering why you show these in Fig. 2-5, particularly since their mention focuses on cumulated monthly sums (which aren't shown). Also, yields are discussed frequently throughout the text in this section. Is it worth making a bar chart of yields to more clearly illustrate their evaluation? I realize that a bar chart may look

busy, but perhaps averaging across years for the sites with multiple years and including standard deviations will work. Related, how are you calculating yields from CLM simulations? It's important to use the peak daily grain carbon value for the entire growth cycle rather than averaging this over some period of time.

I suggest reorganizing the text (and figures) have 4 paragraphs, focusing on the descriptions of: 1) LAI ; 2) yields; 3) NEE; and 4) energy fluxes. Highlight differences among sites within each paragraph. You can also include an opening paragraph that mentions that CLM_WW improves trends for nearly all variables compared to CLM_D, so the remainder of the discussion focuses on the evaluation of CLM_WW.

> Thanks for your suggestions. This section has been extended by a paragraph and new figure (Figure 2) focusing on individual effect of the winter wheat subroutines and the new parameter set for winter wheat. The following text was restructured into four paragraphs, each focusing on certain evaluation variables (LAI, yield, energy fluxes, NEE). The figures (previously Figs. 2 – 6) were merged into a multi-panel figure as suggested by the reviewer (now Fig. 3). Please see also our response to the comment below. Annual performance metrics for the respective simulation runs were added to the supplementary material (Table S2).

> Furthermore, a bar plot (Figure 4) showing simulated and observed annual grain yield was added to the manuscript. The simulated crop yield was calculated from the peak value of daily grain carbon.

> (Line 502-505): "

[Figure]

> Figure 2: Annual grain yield [tDM/ha] simulated with the control run (orange) and the extended winter wheat model with crop specific parameterization (blue), compared to recorded harvest yields (grey) for all simulated winter wheat years (indicated on the x axis) at the sites BE-Lon, DE-RuS, DE-RuM and DE-Kli."

**Figures 2-5:** Is it possible to compile these into a single, multi-panel figure? Given that they all show the same variables for different sites, a single panel would allow the reader to compare across sites more easily. Another, possibly better, alternative is to combine all sites and separate the figures into LAI (Fig. 2) and energy fluxes (LH, SH in Fig. 3). It would also allow you to streamline the description of trends throughout Section 4.1. ˘A˘ TIf I understand the legends correctly, simulations and observations in Figs. 2 and 5 are averaged over multiple years. Can you add uncertainty estimates to these plots? If you plot all individual years (it looks like you possibly do that for observations, but not model), it might be easier to plot averages across years and then plot the uncertainty range associated with interannual variability.

> Thanks for your suggestion. We rearranged the plots in section 4.1 into one multi-panel figure. Due to the small number of compared years (max. 4 years), uncertainty estimates do not add much value to the plots. As briefly discussed in section 4.1, CLM did not capture inter-annual differences in yield well, showing only minor variations between simulated years. This is also reflected in corresponding simulated LAI curves and energy fluxes that differ only insignificantly from year to year.

(Line 483-490):"

[Figure]

Figure 3: Simulation results of (a-d) LAI and simulation results averaged for each month of (e-h) NEE, (i-l) LE, and (m-p) H for all winter wheat years (see Table 3) at the sites (from left to right) BE-Lon, DE-RuS, DE-RuM and DE-Kli. Simulation results from the new routine with crop specific parameterization – CLM_WW (blue) are compared to control simulations (orange) and available site observations (grey) of LAI (all available point observations plotted) and fluxes (averaged over all respective years and for each month respectively). Corresponding performance statistics for daily simulation results during the crop growth cycle are listed in Table 4."

**Fig. 2** states that the observations are GLAI, whereas Figs. 3-5 state that the observations are LAI. Are the observations LAI, GLAI, or does this vary by location? If it is different by location and both LAI and GLAI are used, how might this change the ability to evaluate CLM?

For winter wheat, the green leaf area index (one-sided green leaf area per unit ground surface area) is measured in the field and compared to CLM simulated LAI (defined as one-sided leaf area index, no burying by snow). We used the GLAI synonymously for the LAI for winter wheat and changed the wording in the revised manuscript for consistency.

**Fig. 5:** There aren't any LAI observations plotted in panel a, yet the figure legend suggests that there should be site observation data for LAI.

Unfortunately, there are no LAI observations available for that year. Please see Figure 3 (above) or line 483-490 of the revised manuscript.

**Lines 394-5:** As you state, it looks like the LAI peak is indeed too early. However, even more noticeable (and not mentioned) is the fact that the LAI peak looks to be dramatically underestimated.

Thanks, we added this statement.

(Line 452-453): "The default vernalization also resulted in peak LAI occurring too early in the year, leading to significantly lower photosynthesis compared to the observations."

**Lines 413-4:** Table 4 suggests that crops are only harvested _ 1 month too early, but there are higher observed LH fluxes later in the season than just one month. Is this due to cover cropping, which is not included here?

We have rephrased the text to make this clearer.

(Line 526-530): "The high latent heat fluxes measured at BE-Lon and DE-Kli in the later months of the year (from day 220 onwards) reflect the growth of a cover crop. At both the BE-Lon site as well as at the DE-Kli site, cover crops are typically sown after harvest of winter wheat (mustard at BE-Lon, radish and brassica at DE-Kli), and they strongly affect surface energy fluxes later in the year. In contrast, in the control simulations, as well as in CLM_WW, the crop field were simulated as fallow after the harvest of winter wheat (Fig. 3, Table A1)."

**Line 420:** I think the phrasing "overestimated early growing season LAI" is potentially misleading. While it is technically correct, the simulated peak LAI values are actually similar to observed peak LAI values, but happen earlier in the year. I think it might be more informative to state that the peak magnitudes are similar, but that the peaks happen too early in the year.

Thanks for pointing this out. We rephrased accordingly.

(Line 470-471): "In general, CLM_WW yielded LAI peak magnitudes similar to observations at the sites BE-Lon, DE-RuS and DE-RuM (Fig. 3)."

(Line 478-479): "Overall, the LAI peak simulated with CLM_WW occurred about one month earlier than observed, suggesting that maturation was reached too early."

**Lines 422-3:** What does "growing cycle" refer to hear? As you mentioned earlier, LAI peaks too early and planting and harvest start early, suggesting that phenology is not accurate. Therefore, it is unclear what you mean by "generally good correspondence in growing cycle and LAI".

(Line 466-469): "For all study sites and simulation years, CLM_WW simulations resulted in a much better representation of the growth cycle and corresponding seasonal LAI variation and magnitudes compared to control simulations (Fig. 3). Also, the temporal pattern of energy fluxes and NEE were improved with CLM_WW compared to the control run."

**Lines 437-8:** How can you say that CLM_WW resulted in more realistic magnitudes when you stated in the previous sentence that observations aren't available?

We now compare the LAI results to those for another site and we have rephrased the text accordingly.

(Line 471-472): "For DE-Kli, site-specific observations of the LAI were not available, but simulated LAI magnitudes for DE-Kli using CLM_WW are similar to those for BE-Lon."

**Lines 438-9:** This is confusing. Does it refer to only the simulations, or also reference the observations? I get the sense that you are conflating simulated peak LAI with simulated and observed crop yields. It implies that lower LAI causes the lower crop yields, although I don't think you can say that for sure.

We have rephrased the text to make this clearer.

(Line 654-660): "Despite the general improvement of winter wheat growth and yield simulated with the modified CLM_WW, there is still potential in further increasing the flexibility towards simulating different crop varieties and management practices. Due to the phenology algorithm of CLM5, a low simulated LAI can indicate a lower grain yield due to low biomass growth. Accordingly, the higher simulated LAI for the DE-RuS site was associated with a slightly higher simulated grain yield for DE-RuS compared to BE-Lon. However, this relationship is not reflected in the observations, as the measured grain yield is lower for DE-RuS compared to BE-Lon, although the observed LAI is higher for DE-RuS (Figure 3, Table 3)."

**Line 440:** I think this may be backwards. Table 4 suggests that yields are overestimated in 2011 and match really well for 2016.

Thanks, we corrected this.

(Line 494-495): "For DE-Kli, the CLM_WW simulated crop yield matched the recorded yield data very well for the year 2016 and was overestimated for 2011 by approximately 16 %."

**Lines 453-4:** Are all the subsequent mentions (including the metrics in Table 5) calculated using the cumulative monthly sums?

The metrics were calculated using simulation output and observation data at daily time step.

(Line 520-523): "However, CLM_WW was able to better capture seasonal variations of surface energy fluxes during the growing cycle of the crop (Fig. 3). The correlation coefficients for the energy fluxes (LE, H and Rn) calculated over the period from planting to harvest date for daily simulation results and daily observation data improved for all sites (Table 4)."

**Line 459:** You just stated that the BE-Lon sites high some of the highest correlations in the previous sentence, and here single out this site as having high RMSE and biases with low correlations.

Thanks for pointing this out. We corrected this.

(Line 523-534): "Highest correlations were reached for the sites DE-Kli with r values of 0.62 and 0.71 and for BE-Lon with r values of 0.5 and 0.46 for sensible heat and latent heat flux respectively (Table 4). Due to the simulated LAI peak being too early, latent heat flux is underestimated by CLM_WW (Fig. 3, Table 4). The high latent heat fluxes measured at BE-Lon and DE-Kli in the later months of the year (from day 220 onwards) reflect the growth of a cover crop. At both the BE-Lon site as well as at the DE-Kli site, cover crops are typically sown after harvest of winter wheat (mustard at BE-Lon, radish and brassica at DE-Kli), and they strongly affect surface energy fluxes later in the year. In contrast, in the control simulations, as well as in CLM_WW, the crop field were simulated as fallow after the harvest of winter wheat (Fig. 3, Table A1). While the correlation of the latent and sensible heat flux during the growing cycle of the crop is generally increased with the CLM_WW model, the overall annual correlation is still relatively poor due to the influence of cover cropping and poor representation of post-harvest field conditions (annual performance metrics are included in the supplementary material, Table S3)."

**Lines 460-461:** This sentence should be moved to above, where you briefly mention the mismatch in late-season LH. Also, how does this affect the metrics in Table 5 (see above comments as well).

Thanks for pointing this out. The metrics in Table 5 are calculated for the time between recorded planting and harvest of the crop and thus not affected by this. Please see our response to the comment above, line 523-534 in the revised manuscript, as well as the annual performance metrics included in the supplementary material, Table S3.

**Line 464:** Are you referring to CLM_WW? I suggest clarifying here, as you do include simulations that represent cover crops.

Up to this point, all simulations were run with either the default model or the model including the new winter cereal representation. The cover cropping approach is then introduced in the next section:

(Line 527-530) "At both the BE-Lon site as well as at the DE-Kli site, cover crops are typically sown after harvest of winter wheat (mustard at BE-Lon, radish and brassica at DE-Kli), and they strongly affect surface energy fluxes later in the year. In contrast, in the control simulations, as well as in CLM_WW, the crop field were simulated as fallow after the harvest of winter wheat (Fig. 3, Table A1)."

**Lines 471-2:** It is not quite accurate to say that NEE observations match better due to improved LAI. Consider changing to: "in part due to the better representation of LAI".

Thanks for the suggestion.

(Line 534-535): "Furthermore, CLM_WW was generally better able to match NEE observations compared to control runs, partly due to the better representation of the seasonal LAI variations (Fig. 3)."

**Line 473:** Are you actually using cumulative monthly values? Fig. 6 show NEE in unites of umol CO2/m^2/s^1.

We used NEE rates in $\mu mol\ CO_2/m^{-2}/s^{-1}$ averaged for the respective month.

**Line 475:** Both sites? You mentioned three in the previous sentence. If only referring to two sites, please specify which ones.

(Line 539-540): "The resulting correlation for CLM_WW simulations is still relatively low due to an underestimation of the cumulative monthly NEE during seasons with high NEE at BE-Lon and DE-RuS."

**Section 4.2:**

Perhaps this should be titled "New Parameterizations" or "Sugar beet and Potato Parameterizations" to distinguish from the modified winter wheat parameterization in Section 4.1 The evaluation of corn here seems a bit out of place since this section focuses primarily on the new parameterizations. I'm not sure where it goes (perhaps in supplemental?), though. Perhaps this section could be refocused as "Evaluation of other crop types", which includes corn and also the new crop types.

> We appreciate the suggestions and renamed this section accordingly. The evaluation for temperate corn was moved to the supplement.
>
> (Line 547 onwards): "4.2 Crop specific parameterization of sugar beet and potatoes"
>
> (Line 415-416): "Furthermore, an evaluation of the default parameterization for the CFT temperate corn at the site DE-Kli is included in the supplementary material (Fig. S1, Table S1)."

**Lines 489-91:** I suggest rephrasing to add some detail: "The modifications to winter wheat in CLM_WW do not affect other crop types. Therefore, we add new parameterizations for sugar beet and potatoes to this code."

> Thanks for this suggestion. However, we removed this comment from the revised manuscript because this did not add any crucial information for the reader. Also, this does not have an impact on the study (we could also have used the default model version in this step).

**Lines 502-4:** Is this parameter set modified, or new? What is it strongly improved compared to, if these didn't exist in CLM? I assume it was compared to the default CLM crop model (where the crop might be represented by another type of crop), and it would help to know for sure.

> We changed the wording from 'modified' to 'crop-specific' throughout the text for better comprehensibility.
>
> The CFTs for sugar beet and potatoes exist in the infrastructure of CLM5, yet due to the lack of crop-specific parameterizations, these CFTs (and a number of other CFTs) are merged into the spring wheat CFT within the model code. Thus, although sugar beet and potatoes may be assigned on the simulated land units, the default model basically simulates spring wheat using the corresponding parameterization for spring wheat. In the course of this study, we activated these CFTs within the code to prevent them from being merged into the spring wheat CFT. Consequently, we also supplied crop-specific parameterizations for sugar beet and potatoes.
>
> (Line 548-549): "The crop specific parameter sets were tested for several years with sugar beet and potatoes planting at BE-Lon and DE-RuS respectively."

**Line 507:** You reference spring wheat here. Is this the crop type that default CLM uses for these sites? If so, you might want to make this clearer (and mention it earlier). For example: "The default parameterization in CLM uses spring wheat for these crop types and effectively reproduced the growth cycle and seasonal LAI, simulations using the potato and sugar beet parameterizations better captured harvest date and growth cycle. "

> Thanks for your suggestion.
>
> (Line 549-555): "The performance in reproducing seasonal variations and magnitudes of energy fluxes was strongly improved with the crop specific parameterization. Correspondingly, simulations with the crop specific parameter sets for both sugar beet and potatoes were able to reasonably capture seasonal variations and peak values of LAI as well as growth cycle length and harvest time (Fig. 5, Fig. 6). The control run in CLM uses the spring wheat parameterization for these crop types and therefore reproduced the growth cycle and seasonal LAI of spring wheat, while simulations using the crop-specific potato and sugar beet parameterizations better captured harvest date and growth cycle of these crops."

**Line 509:** As in previous comments, I don't think "modified" is the best way to describe this. I suggest using "crop-specific parameters" or "parameterizations for new crop types" or similar. As far as I understand, parameters for new crop types were added, not modified.

> We appreciate the suggestions and changed the wording accordingly. Please see our responses to the previous comments as well as to Section 3.1.

**Lines 510-2:** It looks like the latent heat flux is very similar for the other site, which might be worth mentioning.

> Thanks for pointing this out. However, as the text is already very extensive we did not want to extend this part of the text any further.

**Lines 528-30:** Performed better for NEE? Please clarify.

> We clarified this:

(Line 591-593): "Simulations of the NEE using the crop specific parameter set yielded a slightly better correlation of 0.58 compared to the control simulation that resulted in a correlation of 0.43 (Table 5)."

**Figures 8 & 9:** I suggest updating the use of "default" and "modified" here based on above comments. Please specify that the LAI results are daily (if they indeed are).
In previous figures, NEE is described as "cumulative monthly", but here is described as "monthly averaged". Can these be calculated and referred to in the same way for consistency?

> We calculated the arithmetic mean of the daily NEE rate for the respective month. The LAI observations are single field measurements (point observations).

> (Line 578-582): "Figure 4: Simulation results of (a-b) LAI and monthly averaged simulation results of (c-d) NEE, (e-f) LE, (g-h) H, (i-j) G and (k-l) Rn for all potatoes years (see Table 5) at the sites (left) BE-Lon and (right) DE-RuS. Simulation results for the control run (orange) and the crop specific parameter set (blue) are compared to available site observations (grey) of LAI (all available observations plotted) and fluxes (averaged over all respective years). Corresponding performance statistics for daily simulation results are listed in Table 5."

**Section 4.3** It seems that this section focuses on crop rotation as much as cover cropping. I suggest updating the heading to "Cover cropping and crop rotation" or similar to reflect this.

> Thanks for the suggestion. We changed the title of this section accordingly.

> (Line 601 onwards): "Cover cropping and crop rotation scheme"

**Lines 553-4:** Is the simulation of a second crop growth onset for the same crop or for the cover crop? The current wording suggests that a second onset is for the same crop within one year AND for the cover crop. If this isn't intended, perhaps change to "simulation of a cover crop as a second crop growth onset within a single year"

> The focus is set on the second onset within a single year. Both a second onset of the cash crop, as well as the onset of a cover crop are possible. We rephrased the text for more clarity.

> (Line 602-606): "The cover cropping scheme was tested for two fields of application: (1) simulation of a cover crop as a second crop growth onset within a single year, and (2) a more flexible crop rotation between different cash crops. In this step, simulations were run with the previously tested crop specific parameterizations for sugar beet, potatoes and winter wheat and results were again compared to a control simulation run, where a consecutive growth of spring wheat is simulated."

**Line 556:** "Greening mix" is this the same as cover crop, catch crop? Please be consistent in your terminology throughout.

> Thanks for pointing this out, we now use the term 'cover crop' throughout the revised manuscript to be consistent.

> (Line 608-609): "A greening mix was planted as a cover crop in between the cash crop rotation of barley (simulated using the spring wheat CFT) in 2016 and sugar beet in 2017."

**Lines 556-557:** Perhaps it would be more accurate to say "the cash crop rotation of barley (simulated using the spring wheat CFT)".

> Thanks for the suggestion. Please see our response above.

**Line 557:** Spring wheat in CLM is not considered a perennial. It can simulate multiple years of spring wheat in a row, but that doesn't make it perennial.

> Thank you for pointing this out.

> (Line 609-612): "While only a consecutive growth cycle of spring wheat is simulated in the control run, the new routine was able to represent the crop rotation from barley to sugar beet in the following year as well as a cover crop in between the cash crop cycles."

**Lines 559-561:** Can the effects of planting cover crops and the crop rotation be isolated?

> Here, we wanted to show that not only an easier crop rotation is possible (especially from summer to winter crop) but also the simulation of a crop that is not considered a cash crop. Technically, this follows the same scheme.

**Line 563:** Please change "plantation" to "planting"

> Thanks, we corrected this.

**Line 576:** Similar to above, spring wheat is not a perennial crop in CLM, as it's planted every growing season.

Thank you for pointing this out. Please see our response to the review comment to line 557 above.

**Figures 10-11:** It looks like these are for the same site and continuous. Why not plot the full time series on the same panel, adding lines or shading to show the transitions and associated crop type labels. Also, do you not have observational data for LH for 2017-2019 (Fig. 11)?

We appreciate your suggestions. We added a plot of latent heat flux to Figure 8.

(Line 636-641): "

[Figure]

Figure 5: (Top) Simulated LAI for crop rotation from sugar beet (2017) to winter wheat (2017/2018) and to potatoes (2019) at DE-RuS using the new cover cropping subroutine (blue) in comparison to control simulation results with the default phenology algorithm of CLM5 (orange). (Bottom) Corresponding monthly averaged simulation results for the latent heat flux with respective bias, RMSE and r over the whole time interval (calculated using simulation output and observation data at daily time step). Available observation data are plotted in grey."

**Section 5**

In addition to the benefits and challenges of the new model developments that you include, I was hoping to see further big-picture discussion, for example about how these new developments might improve future large-scale simulations, possible interactions with climate, etc. Consider adding a paragraph to highlight how your improvements can improve our understanding of larger-scale processes. Also, NEE isn't mentioned at all. Why do you think that NEE didn't improve as much as energy fluxes?

Thanks for your suggestions. We added more detail to this part of the discussion.

Field observations indicate that heterotrophic respiration from soil organic matter and litter acts as a carbon source, which is not represented well in CLM5 and is one of the reasons the quality of the NEE simulation is relatively low. Also, a study by Levis et al. (2014) indicated that CLM4.5 underestimated the land use $CO_2$ by neglecting soil disturbance from cultivation. The authors conclude that the representation of atmospheric $CO_2$ and soil carbon could be improved in LSMs by accounting for enhanced decomposition from cultivation (Levis et al., 2014).

(Line 650-653): "With an average annual winter wheat yield of around 20 Mt/a for Germany, an improvement of 87 % in simulated yield with CLM_WW compared to the default model (as observed at the DE-Rus site in 2018) could result in a difference of several tens of millions of tons in total predicted annual yield on a nation-wide scale."

(Line 730-740): "In their approach, Lombardozzi et al. (2018) studied the effects of idealized cover crop scenarios by simulating winter crops in all crop regions throughout North America. They found that the effects of cover crops on winter temperatures is strongly related to plant height and LAI and emphasized the importance of biogeophysical effects and varietal selection when evaluating the climate mitigation potential of cover cropping (Lombardozzi et al., 2018). With our new routine, it is now possible to evaluate the biogeophysical effects of cover crops over longer time scales and in combination with typical cash crop rotations throughout agricultural areas. Also the ecological potential of different cover crop

varieties could be evaluated. We anticipate that this modification will allow a more realistic representation of seasonal LAI in ecosystems where cover cropping and crop rotations are common management practices. The application of this routine is also of interest for areas with several cash crop cycles within a year like multiple annual crop cycles in India and China (Biradar and Xiao, 2011; Li et al., 2014; Sharma et al., 2015)."

**Lines 597-8:** As mentioned in a previous comment, higher LAI does not mean higher grain yield. There are many factors that affect yield, including photosynthetic rate, nutrient availability, etc. Also, the results presented in this sentence further support that LAI does not directly correspond to yield: grain yield was higher at BE-Lon (which had lower LAI) than DE-Rus.

Thanks for pointing this out. Please see also our earlier reply to comment line 438-9.

(Line 656-670): "Due to the phenology algorithm of CLM5, a low simulated LAI can indicate a lower grain yield due to low biomass growth. Accordingly, the higher simulated LAI for the DE-RuS site was associated with a slightly higher simulated grain yield for DE-RuS compared to BE-Lon. However, this relationship is not reflected in the observations, as the measured grain yield is lower for DE-RuS compared to BE-Lon, although the observed LAI is higher for DE-RuS (Figure 3, Table 3).

In CLM, there are several variables that influence the simulated crop yield, such as LAI cycle and peak, length of the leaf emergence phase, harvest date, and water availability from the soil. Except for soil moisture, these variables are strongly correlated to the GDD scheme which suggests that the simulated crop yield profoundly depends on the GDD. The high sensitivity of simulated yield in CLM towards GDD is not reflected in actual field observation, where crop yield depends on a multitude of factor, environmental conditions (weather, nutrient availability, atmospheric CO2) and management decisions. Underestimation of winter wheat yield at BE-Lon may be due to model deficiencies in representing the complex crop management practices, such as timing and type of fertilizer, ploughing crop varieties and the usage of different winter wheat varieties that can show different responses to water or heat stress, frost and have different grain productivities (White and Wilson, 2006; Bergkamp et al., 2018; Ceglar et al., 2019)."

**Line 603:** CLM may not represent different varieties, but the parameters could be changed (as you did here) to represent different varieties, especially when simulated at single points.

Thanks for your suggestions. We added more details:

(Line 670-676): "In order to include different varieties of any crop, the list of CTFs could be extended with suitable plant parameterizations. However, this information is not readily available, due to combination of measurement data scarcity and the complexity of the phenology algorithm and parameter scheme. The introduction of a phenology scheme based on plant physiological trait information in CLM could be a major improvement in this field (see Fisher et al., 2019), as plant trait information becomes more readily available (e.g. TRY Plant Trait Database, Kattge et al., 2011). Whether considering different varieties and cultivars of a crop is important for regional or global scale simulations remains to be evaluated."

**Line 607:** It might be clearer to say "The early leaf onset and harvest for winter wheat simulated by CLM: : :"

Done as suggested by the reviewer.

**Lines 619-22:** Can this be more specific? How would discretizing plant hydraulic properties improve yield prediction? Also, why does the reference include "Daniel"? How could the properties (parameters?) be estimated by inverse modeling or data assimilation?

Thanks for pointing this out. We added more details:

(Line 691-699): "Within the crop module of CLM5, the carbon allocation of crops is limited by soil water available to the plant. Thus, both an improved soil hydrology and an improved representation of plant hydraulics could play a major role in improving the quality of yield prediction by the model (Bassu et al., 2014; Kennedy et al., 2019). These plant hydraulic properties could be estimated by inverse modelling or data assimilation (e.g. by assimilating measurement data like NEE, LAI, soil moisture and/or energy fluxes using an augmented state-vector approach). In addition, data assimilation of e.g. in situ or remotely sensed soil moisture data and/or LAI could play a major role in increasing the accuracy of regional yield predictions (e.g. Guérif and Duke, 2000; Launay and Guerif, 2005; de Wit and van Diepen, 2007; Fang et al., 2008; Vazifedoust et al., 2009; Huang et al., 2015; Jin et al., 2018)."

**Lines 629-31:** Why isn't it applicable to regional simulations? If a simulation is set up to use land use change, the distributions of vegetation, including crop types, will change, even on a point scale, and can be customized by the user if desired.

Thanks for pointing this out. We added more details:

(Line 701-709): "There is a tool available for CLM5 that enables the simulation of transient land use and land cover changes (LULCC) (Lawrence et al., 2018). It was designed to simulate the effects of changing distributions of natural and crop vegetation, e.g. land use change from forest to agricultural fields and also allows for changes in crop type between years (Lawrence et al., 2018), but does not account for intra-annual changes of agricultural management on crop vegetated areas that happen in double and triple cropping scenarios. While this tool is useful to study general land use changes by changing the land cover type of individual land units, we found it lacks flexibility in accounting for changes within land units of the same land cover and does not account for all 64 CFTs. Furthermore, this tool changes the CFT of each column on the 1st of January every year according to prescribed values (customized)."

**Line 634:** Do you mean before fall of 2018? Fall of 2017 would be the same year.

Thanks, corrected as suggested by reviewer.

**Line 635:** I don't see Figure 12.

Thanks, we meant Figure 11, now Figure 8.

**Section 6**
**Line 665**: Is higher flexibility for crop rotations possible beyond your study and beyond single point simulations? Because it isn't clear how cover cropping was incorporated in the methods, the applicability of this beyond your study or single point sites isn't clear.

Thanks for pointing this out. We added more details:

(Line 717-729): "This new routine can be used to study cover cropping scenarios in future large-scale simulations. The effect of a cover crop during winter months on all crop land units where cash crops are grown in summer could be tested. This could also be tested for specific cash crops only. In addition, it is possible to simulate cover crop plantations based on harvest date thresholds. A defined maximum harvest date for any specific cash crop could define whether a cover crop such as winter wheat would be planted or not. For example, for all sugar beet land units with harvest dates before a certain threshold (e.g. day 290 of any given year) winter wheat could be planted as a cover crop during winter. If this harvest threshold were not reached and the summer crop is harvested late in the year, no cover crop would be planted. Alternatively, these harvest thresholds could define the type of cover crop, e.g. early harvest - winter wheat, late harvest – simple greening mix, etc. Also, historical land use information could be used to simulate realistic cover cropping and crop rotation scenarios. The succession of different crops from historical data could also be used to model the succession of crops for the future. In order to study large scale effects of cover cropping and common crop rotations, the CLM5 model would greatly benefit from further crop specific parameter sets for cover crops such as mustard, and further important cash crops."

**Lines 675-8:** I appreciate that there are numerous improvements that will improve CLM.
However, none of these seem strongly related to the work presented here. For example, there is no evidence that lack of management or incorrect plant hydraulic properties are contributing to model biases.

Thanks for your suggestion, we added a comment that is more specific to our study.

[revised manuscript text omitted]